# The impact of agriculture on tropical mountain soils in the western Peruvian Andes: a pedo-geoarchaeological study of terrace agricultural systems in the Laramate region (14.5°S)

Fernando Leceta[1,2], Christoph Binder[3], Christian Mader[4], Bertil Mächtle[1], Erik Marsh[5], Laura Dietrich[6], Markus Reindel[7], Bernhard Eitel[1], Julia Meister[3]

[1]Geomorphology and Soil Geography, Geographical Institute, Heidelberg University, Heidelberg, D-69120, Germany
[2]Historical Mapping Laboratory, Espacio Interdisciplinario, Universidad de la República Uruguay (UdelaR), Montevideo, C.P. 11200, Uruguay
[3]Geoarchaeology and Quaternary Science, Institute of Geography and Geology, University of Würzburg, Würzburg, D-97074, Germany
[4]ArchDepth Research Group, Bonn Center for Dependency and Slavery Studies, University of Bonn, Bonn, D-53113, Germany
[5]National Scientific and Technical Research Council (CONICET), Universidad Nacional de Cuyo, Mendoza, CP. M5502JMA, Argentina
[6]Prehistoric Archaeology, Martin Luther University Halle-Wittenberg, Halle, D-06108, Germany
[7]Commission for Archaeology of Non-European Cultures, German Archaeological Institute, Bonn, D-53173, Germany

*Correspondence to*: Fernando Leceta (leceta@gmail.com); Julia Meister (julia.meister@uni-wuerzburg.de)

**Keywords.** Andes, prehispanic agricultural terraces, agricultural terrace soils, soil science, paleo-pedology, land use dynamics, plant microfossils

**Abstract.**

This integrated pedo-geoarchaeological study focuses on three abandoned prehispanic terrace agricultural systems near Laramate in the southern Andes of Peru (14.5ºS), aiming to unravel the pedological and land-use history of the region, which served as a significant agricultural hub during prehispanic times. The key objectives of the investigation involved contextualizing the former agricultural management system within its geomorphological and paleoecological framework and assessing the impact of agricultural practices on soil development and quality by comparing non-irrigated agricultural terrace soils with their undisturbed paleo-pedological counterparts. The Laramate terrace complex, with its diverse terrace systems and varied geomorphological and geological settings, provided an ideal setting for the investigation. This comprehensive examination integrated a range of methodologies, including field surveys, digital mapping, and geomorphological analysis based on GIS and remote sensing applications, soil analysis (e.g. grain size, bulk chemistry, nutrient budget), plant micro fossils (phytoliths and starch), and radiocarbon dating.

In the Laramate region, the geomorphological setting of terrace agricultural systems promotes their optimal functioning. The terraces are often located in solar-sheltered areas with western exposure on middle and lower slopes or valley bottoms, which

mitigate intense solar radiation, reduce evapotranspiration, increase soil moisture and minimize erosion. The study identifies

three soil groups in the Laramate region: *Phaeozems*, *Andosols*, and *Anthrosols*. Unique characteristics of *Phaeozems* challenge typical descriptions, influenced by factors such as climatic seasonality, vegetation, fauna, lithology, and aeolian inputs. The terrace soils in the Laramate region are classified as *Terric Anthrosols*, showing no significant degradation even after long-term use. Their balanced acidity and nutrient levels support Andean crop cultivation. Traditional non-mechanized tools, such as the *chaquitaclla* and *rucana*, likely minimized soil disruption. The terraces' tillage horizons have high organic matter, indicating intentional organic manuring. Phytolith concentrations suggest intensive agricultural activity, particularly maize

cultivation, with varying patterns suggesting changes in cultivation, fertilization or mulching practices over time. Starch grain identification aligns with phytolith analyses, reinforcing maize's significance in the region. Although the use of animal-origin fertilizers requires further investigation, there is no evidence of nutrient maintenance through seasonal burning. Irrigation was minimal, and the abandonment of the prehispanic cultivation system was unlikely due to soil exhaustion or terrace instability.

Overall, the prehispanic history of terraced agriculture in the Laramate region extends over four development phases, reflecting dynamic interactions between environmental, cultural, and agricultural factors. The initial phase, from the Formative Paracas period to the Early Nasca period (800 BCE–200 CE), witnessed the establishment of agricultural terraces with simple terrace architecture, while the Middle Horizon (600–1000 CE) saw systematic areal expansion influenced by the Wari culture. Adaptations to drier conditions included terraced agriculture on volcanic soils. The Late Intermediate Period (1000–1450 CE)

witnessed hydrological variability and further terrace expansion to lower altitudes and less agriculturally suitable locations. The final phase, marked by the onset of the Hispanic colonial period in 1535 CE, saw the gradual abandonment of terraced agricultural systems due to demographic shifts and reorganization of production systems. Despite this, the historical trajectory underscores the adaptability and resilience of prehispanic communities in the Laramate region, showcasing innovative terrace agriculture as a means of coping with changing environmental conditions across diverse landscape units.

## 1 Introduction

Terrace farming encompasses a set of specialized techniques crafted to facilitate agriculture in mountainous regions, where steep slopes typically pose challenges for cultivation. This method is globally practiced across diverse climatic conditions and entails substantial alterations to the natural landscape for agricultural purposes. This involves creating artificial terraces and forming an artificial tillage 'Ap' horizon as a new topsoil, as can be observed in the tropical Andes or in central Amazonia

(Glaser and Birk, 2012; Sandor, 2006; Sandor and Eash, 1995; Varotto et al., 2019; Zavaleta García, 1992). Other mountainous landscapes where terrace agriculture played an important role in the development of ancient civilizations include Mesoamerica, particularly in central Mexico, the Mixteca Alta, and the Maya area (Pérez Sánchez, 2019). Additionally, the terraced

landscapes of the mid-high Dagestan mountains of the eastern Caucasus (Borisov et al., 2021) and the arid Negev highlands of Israel (Sapir et al., 2023; Stavi et al., 2024) are significant examples. In southern Europe, terrace farming was prominent in regions from the Mediterranean to the Alps (Ažman Momirski, 2019; Stanchi et al., 2012).

Similar to other semi-arid and mountainous regions worldwide, terracing in the Peruvian Andes serves the primary goal of enhancing agricultural productivity by addressing environmental challenges. These challenges include the erratic availability of soil moisture during dry and rainy seasons, the constrained expanse of cultivable land, and the inherent risk of soil erosion. In the Peruvian Andes, the availability of fertile and arable land is a limited resource, constituting only 19% of the national territory (Pulgar Vidal, 1996; Zavaleta García, 1992). The Western Cordillera of the southern Peruvian Andes boasts a longstanding tradition of agricultural terracing, a practice with roots dating back to prehispanic times. These agricultural terraces typically align with a fertile strip characterized by rich natural soils, notable accumulations of organic matter in the mineral topsoil, and elevated levels of base saturation. This fertile zone extends along the slopes at elevations between 2500 and 3700 m asl. Locally referred to as "Serrania Esteparia" (Brack Egg and Mendiola Vargas, 2010), "Region Quechua" (Pulgar Vidal, 1996) or "Pajonales" (Weberbauer, 1944), this semi-arid and cool ecosystem is predominantly marked by seasonal and sparse shrub and grassland vegetation. It shares agro-ecological characteristics with highly fertile steppe landscapes in mid-latitudes. While steppe ecosystems and soils have been extensively studied in the Eurasian steppe belt, the Canadian prairies, and eastern Patagonia (Blume et al., 2010; IUSS Working Group, 2015; Zech et al., 2014), the agricultural soils in the Peruvian Andes and neighbouring regions across the Andean highlands, which share similar characteristics, have only captured the attention of the scientific community in recent decades (Sandor and Eash, 1995; Sandor, 2006; Kemp et al., 2006; Branch et al., 2007; Goodman-Elgar, 2008; Kendall and Rodríguez, 2009; Aguirre-Morales, 2009; Nanavati et al., 2016; Handley et al., 2023; Vattuone et al., 2011; Sandor et al., 2022).

The region surrounding the small village of Laramate (14°17'11" S, 74°50'34" W), situated on the upper slopes of the Rio Grande drainage in the Peruvian southern Andes, features numerous terrace agricultural systems (Fig. 1). This terrace complex serves as a compelling illustration of the sustainable and long-term utilization of land and soil in the region during prehispanic times. Recent archaeological findings (Reindel and Isla, 2013a; Soßna, 2015) underscore the region's significance as an agricultural hub in prehispanic times, contributing to the development of the local Paracas (800–200 BCE) and Nasca (200 BCE–600 CE) cultures. Additionally, it played a pivotal role in the expansion of the Wari (600–1000 CE) and Inka (1450–1535 CE) cultures in the Andes.

The agricultural terrace systems in the Laramate region offer a valuable opportunity to explore the enduring effects of agricultural practices on soils within the mountain shrubland ecosystem. By contextualizing the former agricultural management system within its geomorphological and paleoecological framework and assessing the impact of agricultural

practices on soil development and quality we aim to unravel the pedological and land-use history of the region. The exploration of prehispanic terrace agricultural systems involved pedestrian and soil surveys, detailed mapping, and geomorphological analysis based on GIS and remote sensing applications. In order to assess the morphological and geochemical soil characteristics resulting from prolonged agricultural use, we compared five non-irrigated agricultural terrace soils from three

terrace agricultural systems with two undisturbed reference profiles across various soil groups in diverse lithological and topographical settings (Fig. 1). This evaluation specifically focused on (i) soil hydrological functions, (ii) nutrient availability and organic matter content, and (iii) vegetation, employing a pedo-geoarchaeological approach. To analyse these aspects, selected samples underwent standard procedures for bulk chemistry, texture, nutrient budget, bulk density, phytoliths, and starch. Additionally, the chronostratigraphy of the sequences was established through radiocarbon dating.

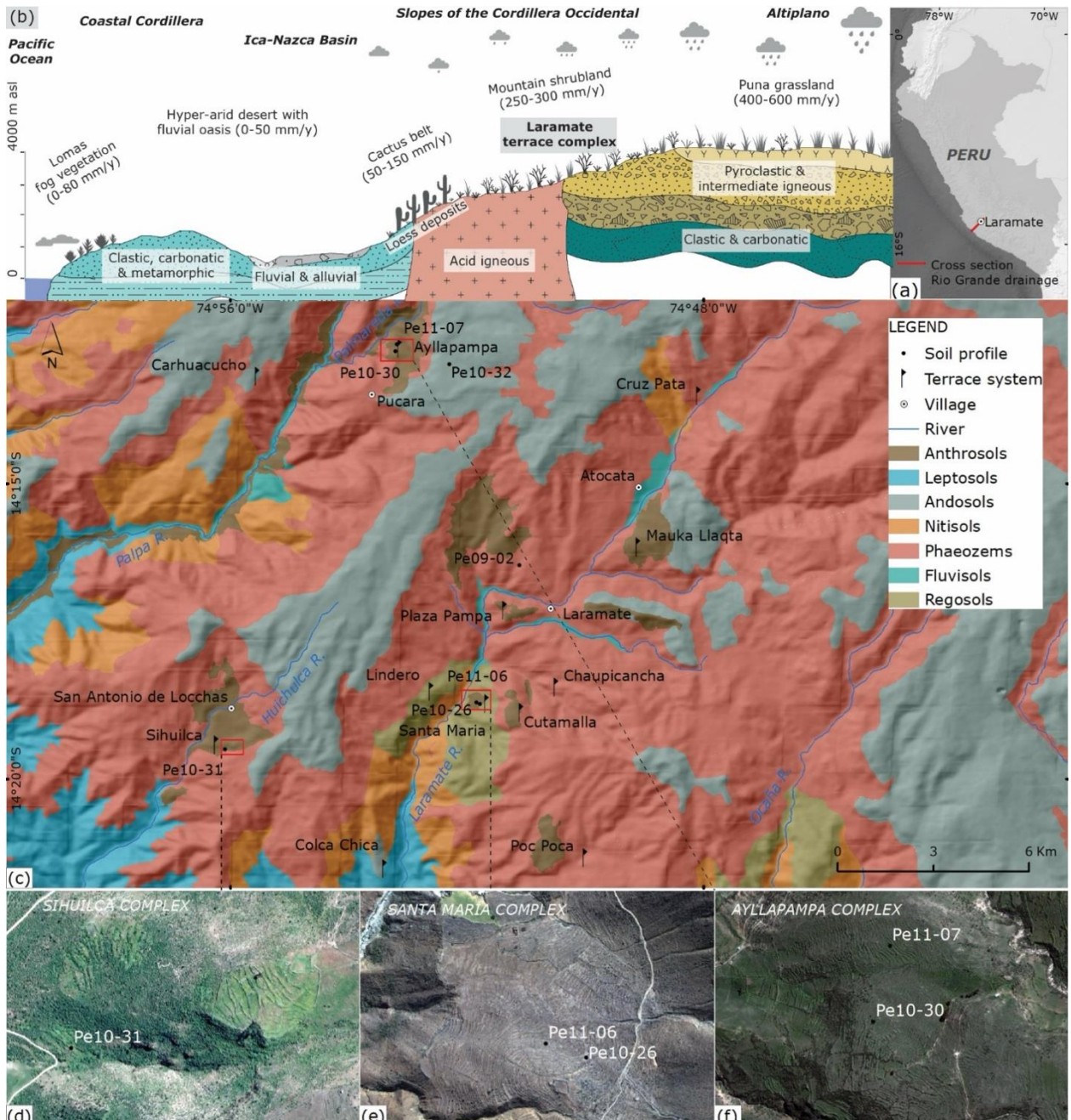

**Figure 1: a:** Location of the Laramate region within Peru. **b:** Schematic cross section of the Rio Grande drainage, including major landscape elements and the Laramate terrace complex. **c:** Soil map of the study area according to the FAO WRB classification and location of major terrace systems, sampling sites and settlements. **d-f:** Satellite images of the Ayllapampa, Santa María and Sihuilca terrace complexes and location of the soil profiles studied. Sources: figure (a): soil data own sampling and survey; lithology, river & village data basis INGEMMET 2001; elevation model ASTER GDEM (e) map (b): NaturalEarth; images (c), (d) © DigitalGlobe; image (e): © Google Maps 2023; figure (b): modified from Mächtle 2007 and INGEMMET 2001, some drawing-vectors from Freepik.

**2 Study site**

**2.1 Location**

The study area is located near the rural village of Laramate (14°17'11"S, 74°50'34"W, 3045 m asl) and covers a total area of 380 km$^2$ in the mountain shrubland landscape of the western Cordillera Occidental in Peru (Fig. 1). In terms of hydrology, this area corresponds to the upper and middle drainage region of the Rio Grande de Nasca. The valley follows a south-westerly course, descending from the continental divide to the Pacific Ocean. The altitude of the mountainous region ranges from 2000 m to 4200 m asl.

**2.2 Geology**

The geology of the Laramate region is complex and can be divided into three main groups. The consolidated sedimentary rocks of the Yura Group, of Jurassic-Lower Cretaceous age, form a thick sedimentary sequence of sandstone, quartzite, shale and limestone, exposed in the valley bottoms and adjacent slopes of the Palpa, Viscas and Ingenio rivers. The location of the Santa María archaeological site corresponds to the Hualhuani Formation subgroup, within which quartzite and quartz rich sandstone beds are arranged in thick beds with some intercalation of laminar stratified silt and claystones (Castillo et al., 1993).

Tonalite and granodiorite are the most widespread lithological group. These acidic intrusive rocks belong to the plutonic bodies of the Peruvian Coastal Batholith, which have intruded since the Lower and Upper Cretaceous through older sedimentary rocks and younger volcanic bodies. While tonalites are mainly exposed in the valley bottoms, granodiorite outcrops are restricted to the summits and slopes. Spheroidal weathering is typical of these rocks, best seen in the sector between the villages of Laramate and Ocaña, where intense weathering disintegrates them into boulders or coarse sand masses. The archaeological site of Sihuilca and the control soil profile Pe09-02 are located on this sub-group (Castillo et al., 1993).

A thick volcanic-clastic sedimentary sequence of Paleogene-Neogene age dominates the north-eastern half of the study area, occupying the upper positions of the Cordillera Occidental as it enters the Altiplano. It gathers a series of conglomerates, breccias, tuffs, lava flows and reworked sandstones. The Ayllapampa archaeological site is located in the Castrovirreyna Formation, an Eocene-Oligocene subunit. At this site, breccias of andesitic composition dominate (Castillo et al., 1993). The Pliocene deposits of the Nasca Group sub-unit are exposed at higher elevations within the Rio Grande watershed. The sequence is composed of a package of lapilli tuffs at the base and is completed in the upper levels by volcanic conglomerates containing clasts of rhyolite, dacite and trachyandesite composition within a sandy tuffaceous matrix (Castillo et al., 1993). The sample site Pe10-32, near Pucará, was developed from the volcanic ejecta of the basal levels.

A major terrain uplift associated with the Quechua 2 and Quechua 3 tectonic phases in the Upper Miocene (Ghosh et al., 2006) has shaped the landscape of the Cordillera Occidental morphological unit (1500-3800 m asl), which is characterised by a highly

dissected topography with pronounced V-shaped symmetrical and asymmetrical transverse valley profiles, shallow soils and steep flanks.

## 2.3 Climate

The study area covers a transitional zone between the semi-arid western slopes of the Cordillera Occidental and the sub-humid western ranges of the Altiplano. Seasonality and semi-aridity are the main features that characterize the climatology of the Laramate region. There are two reasons for the aridity of the Pacific coast and the western slopes of the Andes. First, the orography of the central Andes prevents the transport of moist air from the Amazon basin and the Atlantic. Secondly, the Southeast Pacific Anticyclone generates very dry and stable conditions (Garreaud et al., 2003), inhibiting the development of

local convective activity. The moist air masses that reach the Altiplano and the western flanks of the Cordillera Occidental originate as tropical convective rainfall over the Amazon basin and are subsequently transported to the western Andes by the tropical easterly current during the austral summer months (December to March) (Garreaud et al., 2003).

The climate of the Laramate region is classified as alpine semi-arid and temperate, with a pronounced rainy season from

December to April in which about 90% of the total annual rainfall of 250–300 mm/year is concentrated. This is followed by a long dry season (May to November) characterized by a lack of rainfall, high insolation and greater temperature amplitude due to the lack of cloud cover and stronger easterly winds. Laramate has an average annual temperature of 10°C, and frost can occur in the winter months. Thermal amplitudes increase proportionally with altitude (ONERN, 1971).

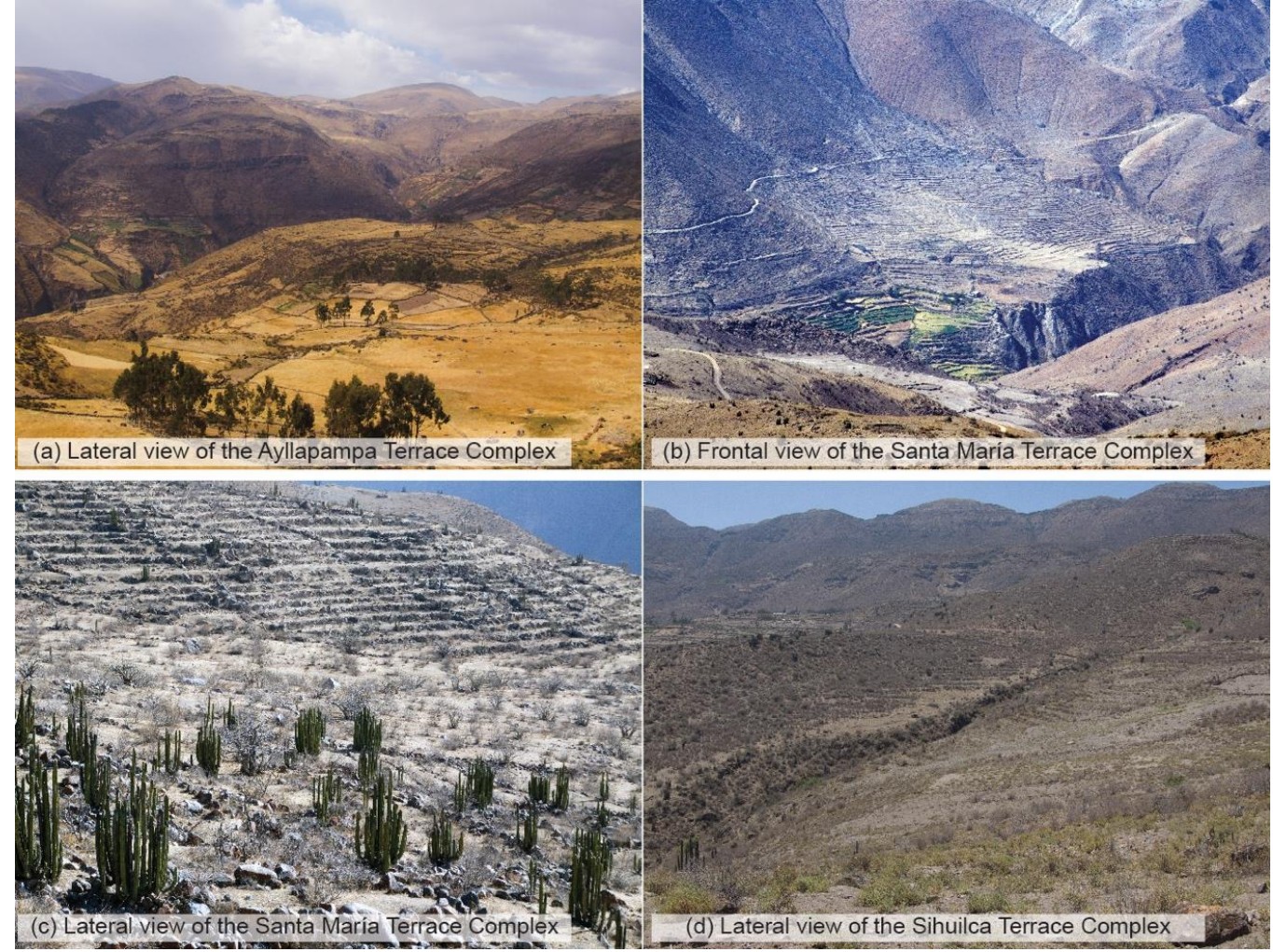


**Figure 2: Panorama photos of the Ayllapampa (a), Santa María (b, c) and Sihuilca terrace complexes (d).**

## 2.4 Vegetation

The mountain shrubland (2500–3800 m asl) is a high-altitude, inner-tropical, low-rainfall vegetation formation that extends
across the steep and highly dissected topography of the Cordillera Occidental. Floristically, it is characterized by the gradual substitution of succulent (*Cereus candelaris* and *Opuntia subulate*) and xerophytic shrub communities (*Viguiera sp, Chinopappus benthamin*, *Mutisia acuminata* and *Dodonea viscosa)*, which predominate in the semi-arid lower levels. The sub-humid upper levels are dominated by deciduous and frost-tolerant woody shrubs (*Schinus molle*, *Escallonia resinosa*, *Agave americana, Dodonea viscosa*, *Tecoma sambucifolia*, *Spartium junceum* and *Caesalpinea spinosa*) and perennial grasses
(*Festuca, Stipa, Deyeuxia* and *Poa)* (Brack Egg and Mendiola Vargas, 2010; MINAM, 2015).

An ecosystem of economic importance since prehispanic times, the mountain shrubland of the Peruvian Andes has suffered more degradation from deforestation, burning and overgrazing than any other vegetation type in the Cordillera Occidental. Non-intensive agriculture, intended for subsistence, is confined to the wet, narrow banks of the major river valleys, as well as
seasonally wet valley flanks and small episodic streams. Barley (*Hordeum vulgare*), potatoes (*Solanum tuberosum*), wheat (*Triticum spp.*), maize (*Zea mays*), avocado (*Persea americana*), ulluco (*Ullucus tuberosus*), broad beans (*Vicia faba*), pea (*Pisum sativum*), oca (*Oxalis tuberosa*) and mashua (*Tropaeolum tuberosum*) are today among the most frequently cultivated species (ONERN, 1971).

**2.5 Soils**

The study area is characterized by two particularly fertile soil groups, *Phaeozems* and *Andosols*, which form on different parent rocks and provide the undisturbed pedological context for the agricultural terrace soils, which are classified as *Anthrosols*.

*Phaeozems* are the dry grassland soils of the semi-arid mountain shrubland of the Peruvian Andes. They typically form on the middle and upper slopes of the Cordillera Occidental at altitudes between 2500 and 3800 m asl. Due to their excellent agro-
ecological quality, the *Phaeozems* of the mountain shrubland have been used since prehispanic times and are considered the most productive soils in Peru (Brack Egg and Mendiola Vargas, 2010; Instituto de Estudios Histórico-Marítimos del Perú., 1996; Zamora Jimeno and Bao Enríquez, 1972). Although *Phaeozems* are not described in detail for the study area, the genetically related soil types *Luvic Kastanozems* and *Calcic Kastanozems* are described. While *Luvic Kastanozems* are characterized by increasing clay content and clay coatings with depth due to clay illuviation, *Calcic Kastanozems* are alkaline
soils with typical soil saturation of more than 50% in the upper 50 cm of the topsoil (Instituto de Estudios Histórico-Marítimos del Perú., 1996; Zamora Jimeno and Bao Enríquez, 1972).

*Andosols* are dark soils typical of volcanic landscapes, developed from glass-rich volcanic ejecta in almost any climate. In the western Andes, they occur in the undulating landscape of the upper slopes of the Cordillera Occidental and the adjacent
Altiplano above 3600 m asl, where semi-aridity prevents soil nutrient leaching and infrequent frosts do not severely restrict root growth (Brack Egg and Mendiola Vargas, 2010; Instituto de Estudios Histórico-Marítimos del Perú., 1996; Zamora Jimeno and Bao Enríquez, 1972). Their development is strongly conditioned by the widespread grassland vegetation of the mountain shrubland of the Peruvian Andes, which provides organic matter (Zamora Jimeno and Bao Enríquez, 1972). Mountain shrubland *Andosols* are moderately acidic and generally shallow; organic carbon levels within the top 50 cm are
typically above 1% (Zamora Jimeno and Bao Enríquez, 1972). Although not as fertile as *Phaeozems*, *Andosols* have dark and humus-rich surface horizons, less base-rich than typical mid-latitude steppe soils (*Chernozems*, *Kastanozems*, *Phaeozems*) but potentially valuable for agriculture if managed appropriately (IUSS Working Group, 2015). Whereas *andic* properties are

related to an advanced state in the weathering sequence of pyroclastic deposits (tephric soil material > vitric properties > andic properties), layers with *vitric* properties are distinguished by a lower degree of weathering (IUSS Working Group, 2015).


*Anthrosols* are soils significantly modified by human activities such as harvesting, irrigation, fertilization, and ploughing. These soils are found worldwide in regions with a long history of agriculture. Local environmental conditions and agricultural practices contribute to the formation of diagnostic horizons (e.g., *irragic*, *terric*, *escalic*, *hortic*), distinguishing different types of Anthrosols (Zech et al., 2014; IUSS Working Group, 2015). The formation of these anthropogenic soils, referred to as

agropedogenesis (Sandor et al., 2005; Kuzyakov and Zamanian, 2019), contrasts with natural pedogenesis; constant soil mixing leads to a homogenization of soil characteristics, causing *Anthrosols* to become similar despite their different origins. Long-term and intensely cultivated soils may eventually reach a stage of agrogenetic degradation, characterized by a loss of fertility. However, if agricultural practices on these soils are abandoned, they can gradually return to their original, undisturbed state over time (Kuzyakov and Zamanian, 2019).

*Anthrosols* on agricultural terraces are a characteristic feature of the Peruvian Andes, occurring in various lithological and vegetation contexts at altitudes between 2200 and 3800 m asl. Prominent terrace systems are found in the large inter-Andean valleys, such as Mantaro, Apurimac, Urubamba, and Santa, as well as in the southern Cordillera Occidental, including the Colca Valley (Brack Egg and Mendiola Vargas, 2010; Instituto de Estudios Histórico-Marítimos del Perú., 1996; Zamora Jimeno and Bao Enríquez, 1972). Although the area in the middle and upper catchment of the Rio Grande showcases extensive

prehispanic agricultural infrastructure (Reindel and Isla, 2013a; Soßna, 2015); cf. chapter 2.7), detailed pedological descriptions of the *Anthrosols* are still lacking.

## 2.6 Holocene environmental and prehispanic settlement history of the Laramate region

The history of human occupation and settlement is well documented in the Laramate region, from the Archaic period (8000–3500 BCE) to Inka sites from the early sixteenth century (Beresford-Jones et al., 2023; Reindel, 2009; Reindel and Isla, 2013a;

Soßna, 2015; Unkel et al., 2012). The Cerro Llamocca sacred mountain complex (4487 m asl) includes the oldest archaeological site hitherto recorded in this study area: a rock shelter with evidence of human occupation in the Early Archaic period ~ 8000 BCE (Mader et al., 2023b). Settlement and land use activities increased significantly during the Formative period from 1700 to 200 BCE, especially toward the end of the Paracas period (800–200 BCE), coinciding with a more humid climate following a pronounced dry period from 2600 to 2200 BCE. Large-scale archaeological excavations at Collanco (1630

m asl) and Cutamalla (3300 m asl) have provided rich data on the chronology, layout, and use of Paracas settlements in this highland region (Mader, 2019; Mader et al., 2023a, 2024; Reindel et al., 2015; Reindel and Isla, 2017, 2018). These settlement activities were associated with cultivation in the extensive terrace farming systems that surrounded the sites. Cutamalla was an economic and agricultural centre that was integrated into a considerable interregional exchange network of various goods

(e.g., obsidian, camelid wool, maize, and other agricultural goods) with llama caravans (Mader, 2019; Mader et al., 2018, 2021, 2023a, 2024; Reindel and Isla, 2013b).

The transitional Initial Nasca period (120 BCE–90 CE) was characterized by intense demographic mobility (Unkel et al., 2012). The fluorescence of the Nasca archaeological culture during the Early Intermediate Period (200 BCE–600 CE) – a time of high cultural development – coincides with a humid and relatively stable interval. The decline of the Nasca culture at the end of the Early Intermediate Period and a strong presence of the Wari culture during the Middle Horizon (600–1000 CE) were largely contemporaneous with an abrupt turn to a sustained dry period between 600–800 CE. The expansive Wari state, which originated in the highlands, occupied ancient Nasca settlements and extended its influence into coastal regions (Reindel and Isla, 2013a). A representative Nasca and Wari site in the Laramate region is Huayuncalla (3100 m asl), which was also excavated on a large scale to obtain reliable including settlement data (Isla and Reindel, 2014, 2022). Dry conditions prevailed until 1250 CE.

More humid but hydrologically highly variable conditions prevailed again after 1250 CE, which apparently had positive effects on settlement activities and an increase in population during the Late Intermediate period (1000–1450 CE), another era of cultural and economic flourishing in the region (Soßna, 2015). These developments were supported by advances in agricultural techniques and a massive expansion of sophisticated and partly irrigated farming terraces, leading to an intensification of agricultural production. The Inka state of the Late Horizon (1450–1535 CE) had a minor presence in the Laramate region; Santa María is one of the few sites with surviving Inka architecture and it was connected to the imperial road network. European contact, beginning in 1535 CE, had severe consequences for Andean communities and agricultural regimes. Introduced diseases led to a demographic decline and thus the available workforce. Moreover, aridization in the eighteenth and nineteenth century led to a profound environmental turnover, marked by dry conditions impacting the Altiplano ranges (Schittek et al., 2018). The current population density in the Laramate region is low and merely a small portion of the terrace agricultural systems are in use today, usually located along rivers.

In sum, important cultural changes seem to have coincided with significant changes in paleoclimate (Eitel and Mächtle, 2009; Mächtle and Eitel, 2013; Reindel and Isla, 2013a; Schittek et al., 2015). The mobility of people, llamas, and commodities was an important component of cultural and economic development (Beresford-Jones et al., 2023; Mader et al., 2021). Archaeological and geoscientific data indicate that a main function of prehispanic highland settlements and their related extensive terrace systems was agricultural production (Mader et al., 2024).

## 2.7 The Laramate terrace complex

The Laramate terrace complex encompasses a collection of predominantly abandoned prehispanic agricultural terrace systems near the village of Laramate (14°17'11" S, 74°50'34" W, 3045 m asl; Fig. 1). These terrace systems extend in a scattered pattern across both flanks of the Laramate-Viscas and Palmadera-Palpa rivers, situated between the semi-arid and sub-humid margins of the mountain shrubland ecosystem. Almost each terrace system is closely associated with, and frequently named after, an adjacent archaeological settlement of varying prehispanic age. Notable examples include Sihuilca, Locchas, Santa María, Pocpoca, Chaupicancha, Cutamalla, Carhuachuco, Ayllapampa, Lindero, Mauka Llaqta, Plaza Pampa, and Colca Chica (Soßna, 2015).

While the primary purpose of the terrace systems in the Laramate area was for the cultivation of agricultural products, discoveries indicate that these terraces served as more than just cultivation sites (Reindel and Isla, 2013a). They also functioned as processing and storage units for specific products, such as maize, wood, or wool, intended for later commercial exchange. Furthermore, these terraces were occasionally utilized for purposes such as animal husbandry, residential use, erosion control, and even ceremonial practices, as exemplified by the *Challupas* building in Santa María (Soßna, 2015).

The dimensions of each individual terrace system vary, influenced by both soil moisture conditions and the area of potentially cultivable land. Additionally, a discernible connection between the size of the system and its geomorphologic setting exists. The most expansive terrace systems, spanning several square kilometres, are typically situated on lower and gently sloping terrains, such as valley bottoms, alluvial fans, and debris cones. In contrast, those confined to narrow and steep valleys and slopes tend to be significantly smaller in size, exemplified by the terraces surrounding the Chaupicancha site (Fig. 1).

The absence of significant hydraulic engineering features in the Laramate area, such as water reservoirs, aqueducts, or irrigation canals described in other regions like the Rio Grande's floodplain and lower drainage area, or the notable *Puquios* filtration galleries and *Khadin*-like water harvesting systems found in the Nasca-Palpa region of the Andean footzone (Hesse, 2008; Mächtle et al., 2009; Soßna, 2015), suggests that irrigation played a minor role in these agricultural terrace systems. In prehispanic times, the functionality of the Laramate terrace complex essentially relied on a rain-fed system, where moisture management centred on efficient drainage and the storage of rainwater within each terrace.

Only a relatively limited network of irrigation canals supplies certain terraces, primarily concentrated on the adjacent slopes of rivers in the upper catchments, such as the Laramate-Viscas river, which ensures a consistent water supply throughout the year. Currently, the terraces located between Laramate and Atocata stand out as the sole group with a constant water supply utilized year-round. Additionally, smaller irrigation systems have been identified in the lower slopes and on alluvial terraces, notably in the northern sector of the Santa María settlement. Overall, pinpointing irrigation infrastructures in the Laramate

area and determining their age poses challenges due to their often poor preservation. However, in the vicinity of Santa María, an irrigation facility is likely associated with the Middle Horizon Cultural Period, given its connection to a nearby small Wari compound (Soßna, 2015).

Currently, the active terraces are utilized for agriculture only during a brief period determined by the rainy season. Permanent water sources are scarce, occurring mainly in favourable locations where they enable an extension of the regular cropping season and, subsequently, facilitate extensive sheep and goat grazing. The Chaupicancha terrace system stand out as an exceptional case in this regard. Previous research suggests that the construction of terraces and their intensive use in the Laramate area in prehispanic periods can be associated with more favourable climatic conditions than those observed today. This correlation is particularly noted during the Early Horizon (800–200 BCE), the Early Intermediate Period (200 BCE–600 CE), and the Late Intermediate Period (1150–1450 CE) in the upper and middle Rio Grande basin (Eitel et al., 2005; Mächtle, 2007; Mächtle and Eitel, 2013; Schittek et al., 2015).

## 3 Material and Methods

GIS-based geomorphological analyses have been carried out in order to understand the regional morphodynamics and the geomorphological setting of the terrace systems. Soil surveys were conducted to generate an accurate and current soil map of the study area at a scale of 1:50,000, adhering to international standards. The investigation into prehispanic agricultural terrace systems involved both pedestrian surveys and aerial mapping, utilizing satellite images. To discern morphological and geochemical features arising from prolonged agricultural activity, detailed pedological examinations and subsequent laboratory analyses were carried out on five key terrace profiles. These profiles were strategically chosen from a cluster of three terrace agricultural systems in the Laramate area, directly associated with the archaeological sites of Ayllapampa (profiles Pe11-07 and Pe10-30), Santa María (profiles Pe10-26 and Pe11-06), and Sihuilca (profile Pe10-31), which exhibit evidence of repeated prehispanic use. Additionally, two reference-sampling sites (Pe10-32, Pe09-02) were selected within the *Phaeozems* and *Andosols* edaphic soil provinces to represent undisturbed pedological contexts (Fig. 1, 2).

### 3.1 GIS-based geomorphological analyses and mapping of agricultural terraces on satellite images

To obtain a detailed understanding of the regional morphodynamics and their impact on the structural stability of the agricultural terrace systems, GIS-based geomorphological analyses were conducted. These analyses focused on (1) the main landforms of the fluvial subsystems of the Ingenio, Viscas, and Palpa rivers, and (2) the geomorphological features of each terrace system within its local catchment. The analyses were based on the ASTER Global Digital Elevation Model (GDEM) Version 3 (NASA, 2019) and the physiographic map of the Rio Grande at a scale of 1:300,000 for the main landforms (ONERN, 1971), guided by the FAO hierarchy of major landforms (Jahn et al., 2006).

For precision in mapping the chosen agricultural terrace systems, a thorough analysis of satellite imagery from Google Earth Pro was conducted, focusing on discernible patterns, such as parallel lines indicative of terrace presence. Subsequently, individual terrace walls and boundaries were manually identified and digitized, distinguishing between abandoned and actively used terraces. Nevertheless, challenges arose in the identification process, especially in regions obscured by vegetation. Distinguishing terrace walls from animal tracks and geological formations proved challenging in certain instances.

Additionally, there were areas where satellite imagery suggested terracing, yet the terrace walls lacked the necessary detail for accurate mapping.

## 3.2 Field methods

Fieldwork was conducted in four campaigns spanning the years 2009 to 2011 and 2021, occurring during the dry season in the austral winter months. In consideration of local relief conditions, soil profiles were systematically arranged in a catenary

manner along hillslopes. The variations observed in genetically related soils were found to be influenced by landscape location, anthropogenic impact, drainage characteristics, and the downstream transport of soil material (Blume et al., 2010; Zavaleta García, 1992). The catena under investigation stretches across a 13 km transect running parallel to the Laramate-Viscas and Palmareda-Palpa rivers, situated between the villages of San Antonio de Locchas (14°19'01"S, 74°55'56"W, 2760 m asl) and Pucará (14°13'32"S, 74°53'36"W, 3520 m asl). Soil field properties were described in accordance with the FAO Guidelines

for soil description (Jahn et al., 2006), and soil classifications were carried out following the IUSS Working Group on the WRB (IUSS Working Group, 2015).

## 3.3 Laboratory methods

### 3.3.1 Soil parameters

A total of twenty-eight samples for soil physico-chemical analyses were collected from seven soil profiles (Supplementary

Table S1). The samples were analysed at the Laboratory of Geomorphology and Geoecology of the Institute of Geography at the University of Heidelberg.

The sample preparation for laboratory analysis was following standard procedures of ISRIC (van Reeuwijk, 2002); the fine-earth fraction ($\leq$ 2.0 mm) of air-dried samples was removed by dry sieving prior to the analyses. All analytical results refer to the fine earth fraction. Grain size analyses were carried out by a combination of the Köhn-pipette method (silt and clay

fractions) and wet sieving (sand fractions) (Schlichting et al., 1995). The measurement of the field-estimated bulk density followed the WRB's recommendations (Jahn et al., 2006). A cylindrical coring instrument was used to obtain the samples volume without disturbing the natural soil structure. The dry mass was determined after 48 hours of drying at 105°C. Bulk density's calculation expresses the quotient of the substrate´s dry weight and the core´s volume. Soil colour was determined using the Munsell Soil Color Chart book (Munsell Color, 2000). The process involved using 5 g of homogenized soil substrate,

maintaining uniform diffuse lighting conditions, and applying a controlled amount of water with a sprayer to achieve a low moisture level in the sediment. The pH value was measured in a 1:2.5 suspension in 1 M KCL solution using a pH-meter (van Reeuwijk, 2002). The cation exchange capacity ($CEC_{pot}$) and base saturation (BS) were detected using the Ammonium Acetated Method (van Reeuwijk, 2002). The concentrations of plant-available cations Magnesium ($Mg^{2+}$), Kalium ($K^+$), Calcium ($Ca^{2+}$), Natrium ($Na^+$) were determined through Flame Atomic Absorption Spectrometry (Schimadzu AA-6300). Additionally, the concentration of exchangeable cation Aluminium ($Al^{3+}$) was determined using Graphite Furnace Atomic Absorption Spectrometry (Analytik Jena-AAS Zenit 60). $H^+$ ion's concentrations were obtained from the determination of Exchange Acidity (H-value) in 1 Mol KCL solution. The soil organic carbon ($C_{org}$) contents were determined by applying the Wet Combustion Method following the DIN procedures (DIN 19684-2, 1977). To obtain the organic matter (OM) content indirectly, we followed the procedures described by ISRIC (van Reeuwijk, 2002) and Barsch et al. (2000). This involved converting Corg content (%) to OM content (%) by multiplying by the empirical factor of 2. Total carbon, nitrogen, and sulphur contents were measured on milled samples using the CNS analyzer vario MAX (Fa. Elementar). The carbon:nitrogen (C:N) ratios were calculated to evaluate the mineralization rates in the soil. Total phosphorus contents were photometrical detected in the form of a molybdenum-blue complex at 880 nm (Analytik Jena Specord 200 Plus) using the Ammonium molybdate spectrometric method according to Olsen (1954) described by van Reeuwijk (2002). The plant available phosphorus fraction was extracted with a 1% citric acid solution; phosphate in the extract was determined colorimetrically with the blue ammonium molybdate method with ascorbic acid as reducing agent (van Reeuwijk, 2002).

### 3.3.2 Radiocarbon dating

To place the terraces systems in time and to compare chronologies to nearby residential contexts, six charcoal samples and one bulk sediment sample from the terraces at Sihuilca, Santa María and Ayllapampa were radiocarbon dated at the Curt-Engelhorn-Centre of Archaeometry in Mannheim, Germany (Table 1). Samples were pretreated using the acid-alkali-acid method to remove contamination by carbonates and humic acids. The results were calibrated in OxCal 4.4 (Bronk Ramsey, 2009), using a site-specific mixed curve of SHCal20 (Hogg et al., 2020) and IntCal20 (Reimer et al., 2020) that reflects modern mixtures of air from the Northern and Southern Hemispheres at Cutamalla (Mader et al., 2024), following recently published methods (Ancapichún et al., 2022; Marsh et al., 2023).

### 3.3.3 Phytolith analyses

Phytolith analyses were conducted on twenty-five samples derived from five terrace profiles (Pe10-31, Pe10-26, Pe11-06, Pe10-30, Pe11-07), along with one reference profile (Pe10-32) at the University of Würzburg, following established extraction protocols (Albert et al., 1999). Approximately 1 g of air-dried sample (< 2 mm) was treated with 3 N HCl, 3 N $HNO_3$, and $H_2O_2$ (30 %) to remove carbonates, phosphates, and organic material. The insoluble mineral components were then fractionated

into three density-based fractions using a ~2.4 g/ml sodium polytungstate solution [$Na_6 (H_2W_{12}O_{40}) \times H_2O$]. The third fraction, anticipated to contain the majority of phytoliths due to its lower density, was used for analysis. Approximately 1 mg of this fraction was weighed, mounted on a microscope slide with Entellan New (Merck), and examined under a KERN OBE-114 microscope at 400x magnification. Image capture of selected phytoliths was achieved using a KERN ODC 825 microscope camera. To establish a suitable basis for statistical evaluation, a minimum of 200 identifiable phytoliths per sample were counted (Albert et al., 1999; Meister et al., 2017; Piperno, 2006).

Morphological identification of phytoliths relied on subject-specific literature (Bremond et al., 2008; Pearsall et al., 2003; Piperno, 1988, 2006; Twiss, 1992), with adherence to the International Code for Phytolith Nomenclature where applicable (International Committee for Phytolith Taxonomy (ICPT), 2019). Phytoliths lacking a clear description were categorized as weathered morphotypes. Phytolith concentrations were calculated as described by Albert et al. (1999).

### 3.3.4 Starch analysis

Starch analyses were conducted at the University of Halle on five samples derived from two terrace profiles (Pe10-31, Pe10-26), along with one reference profile (Pe10-32; Supplementary Table S2). Microfossil extraction followed the procedure developed by Liu et al. (2018). For each sample, 40 mg of sediment was meticulously ground in a mortar and transferred to individual tubes (1.5 ml). To these tubes, 50 µl of EDTA was added, followed by 450 µl of a 2.4 g/ml sodium polytungstate solution [$Na_6 (H_2W_{12}O_{40}) \times H_2O$]. The tubes were vortexed to disperse the sediment and then subjected to an ultrasonic bath for 10 minutes. Afterward, the tubes underwent centrifugation at 5000 rpm for 10 minutes to concentrate the starch, and the resulting supernatant was transferred to a new tube and vortexed. Subsequently, 50 µl of the supernatant from each sample was mounted on glass slides and analysed for starch using polarisation filters at 400x magnification with a Bresser polarisation microscope. Photographs were captured using a 12 MP Bresser Microcam.

## 4 Results

### 4.1 Geomorphology and landforms of the Laramate terrace complex

The Laramate terrace complex roughly corresponds to a morphological subunit that extends along the NE-SW courses of the main tributaries: Palpa, Viscas, Ocaña, and Otoca (Fig. 1). With an average slope of 19° and an elevation gradient of 344 m/km, this subunit is a high-gradient mountain range (Jahn et al., 2006), also known as the Transverse Valleys of the Western Slopes (Castillo et al., 1993). The structures were shaped by morphodynamic activity within the drainage system rather than by major geological structures such as faults, anticlines, or synclines. The erosive nature of the Rio Grande drainage system has resulted in a heterogeneous lithological sequence, featuring landforms with diverse topography at different positions and altitudes within the incised valley, ranging from scarps on the upper slope (3800 m asl) to younger fluvial terraces on the valley floor (2000 m asl). The largest agricultural terrace systems are found in areas where the terrain is more gently sloping, such as on debris cones, alluvial fans, colluvial deposits, or fluvial terraces.

The **Sihuilca terrace system** is situated within the Huichulca catchment area, covering 21.8 km², which is a tributary of the Viscas River system (Fig. 1d, 3). Terrace cultivation remnants are evident across the entire area, ranging from steep scarps at 3100 m asl near the catchment boundaries to 2300 m asl at the lowermost fluvial terraces. The most prominent cultivable area, however, lies on a southwest-facing dissected plain consisting of consolidated colluvial deposits with an average slope of 4°. This area extends over a length of 2150 m between altitudes of 2650 and 2800 m asl.

The **Santa Maria terrace system** is located within a minor water catchment area spanning 1.61 km², a tributary of the Laramate-Viscas River, characterized by a network of short-range and intermittent streams. Throughout the Santa Maria catchment area, which ranges from 2555 to 3240 m asl, the slopes vary greatly and can be classified as moderately steep, with an average gradient of 22°. The core terraced area of Santa Maria is situated on a debris cone covering 0.28 km² at the lower slope of the catchment (Fig. 1e, 8), with slopes averaging 11° and altitudes ranging from 2660 to 2760 m asl.

The **Ayllapampa terrace system** lies within a diverse landscape marked by small-scale, parallel, and intermittent streams that feed into the Palmadera-Palpa River. Spanning an area of 7.28 km², the catchment of the Ayllapampa system is situated at elevations ranging from 3160 to 4280 m asl. The upper catchment area features a hilly relief, transitioning to a flatter terrain in the middle and lower sections. The agricultural terraces are located in these middle and lower reaches, at altitudes between 3170 and 3530 m asl, primarily oriented northwest along the left edge of the Palmadera River. The slopes in this area are moderately steep with gradients up to 13°.

## 4.2 Studied terrace systems and pedo-sedimentary characteristics

### 4.2.1 Sihuilca terrace system: overview and soil profile Pe10-31

The archaeological site Sihuilca (PAP-888 and PAP-897) was recorded in 2008 by surveys of the Nasca-Palpa Archaeological Project. According to surface ceramics and architectural remnants, PAP-888 is a domestic settlement dating to the Nasca Early Intermediate period (200 BCE–600 CE) and Wari Middle Horizon (600–1000 CE), while PAP-897 probably dates to the Middle Horizon (600–1000 CE) (Reindel and Isla, 2013a; Soßna, 2015).

The Sihuilca terrace system encompasses an area of approximately 650 ha (6.50 km²), with around 318 ha (3.18 km²) clearly identified as terraced agricultural land (Fig. 3). Mapping efforts have identified a total of 2739 terrace walls, amounting to a collective length of 107.49 km. These identified terraces predominantly face west and exhibit a generally heterogeneous pattern. Larger terraces are observed in flatter areas, while smaller terraces are found on steeper slopes. The width of the terraces varies considerably, ranging from 3 to 30 m.

Soil profile Pe10-31 (14°19'30.22"S, 74°56'5.01"W, 2750 m asl) is situated approximately 1 km southwest of the village of Locchas within the catchment area (21.8 km²) of a tributary of the Huichulca-Viscas river system. The profile is positioned in the centre of a 650 m long slope that faces west, featuring an inclination of 25°.

Table 1: Sample information and results of radiocarbon dating in the Laramate terrace complex. $\delta^{13}C$ values are from AMS stable isotope measurements and are not comparable to $\delta^{13}C$ values measured with IRMS.

| Lab code MAMS | Profile and sample ID | ¹⁴C age | ± | $\delta^{13}C$ AMS (‰) | Median | 95% range (rounded by 10 years) | Soil horizon | Horizon depth (cm) | Sample material |
|---|---|---|---|---|---|---|---|---|---|
| 13597 | Pe10/30-4 | 1291 | 22 | -25.2 | 730 CE | 670–860 CE | AC | 65–100 | charcoal |
| 13598 | Pe10/30-3 | 1183 | 22 | -22.6 | 930 CE | 770–990 CE | Ah | 35–65 | charcoal |
| 13599 | Pe10/31-3 | 1169 | 13 | -24.1 | 940 CE | 880–990 CE | Ah | 20–50 | charcoal |
| 20089 | Pe10/26-4 | 212 | 22 | -26.8 | 1760 CE | 1650–1810 CE | C1 | 60–100 | charcoal |
| 20090 | Pe11/06-3 | 611 | 19 | -27.8 | 1350 CE | 1310–1410 CE | AC | 30–45 | charcoal |
| 20091 | Pe11/07-2 | 2920 | 21 | -27.3 | 1080 BCE | 1210–1000 BCE | Ah | 25–50 | charcoal |
| 18749 | Pe10/32-3 | 3754 | 19 | -28.9 | 2120 BCE | 2210–2030 BCE | 2Ahb | 20–30 | bulk sediment (≤ 2 mm) |

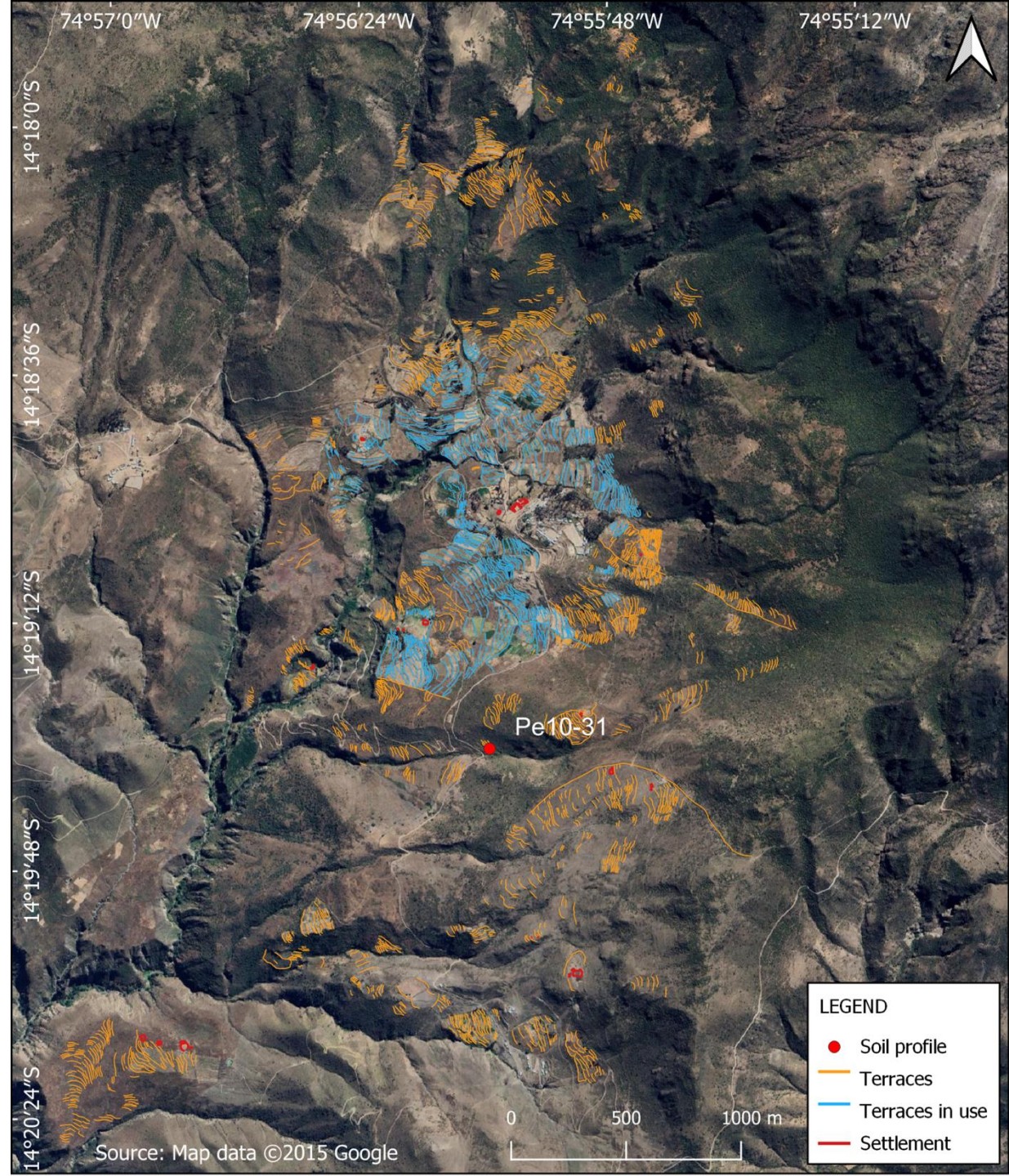

**Figure 3: Detailed map of the Sihuilca terrace system.**

The sample site exhibits no signs of recent soil erosion. Isolated stones (6–20 cm) and boulders (20–60 cm) likely originate
from collapsed terrace walls. The topsoil can be divided into two sections based on its structure. The first section (0–20 cm) is
stratified and moderately cohesive, while the second section (20–50 cm) is characterized by coarse, blocky, sub-angular peds
and displays moderate to high cohesion. The upper 50 cm of the soil matrix is enriched with roots ranging from 2 to 5 mm in
diameter (Fig. 4).

The soil type is a *Terric Anthrosol (Loamic, Escalic)* with two edaphic cycles and six soil horizons (Ap1-Ap2-Ah-Bwt-C1-
C2). **Ap1** (0–10 cm) is a slightly compacted, poorly developed, dark brownish-black (10 YR 3/3) tillage horizon. The
underlying **Ap2** (10–20 cm) tillage horizon is slightly more compact, less developed and dark brownish-black (10 YR 3/3),
containing human artefacts such as obsidian point fragments. The following **Ah** (20–50 cm) is brownish-black (10 YR 2/3),
with slightly higher humus content and structural development. It represents the former soil surface before terracing. The
horizon contains particles of burnt charcoal. Its lower boundary is gradual. **Bwt** (50–70 cm) is a transitional horizon with a
brownish-black colour (10 YR 2/3), higher skeletal content and an increased clay content. It marks the lower limit of human
influence on the soil. **C1** (70–85 cm) is a brownish-black (10 YR 2/3) horizon. **C2** (> 85 cm) represents the parent material of
the soil, slightly weathered granodiorite; the horizon contains a few boulders, systematically situated at the lower terrace base.
A [14]C-dated charcoal taken from the **Ah** horizon (20–50 cm) returned an age of ~940 CE (880–990, 95% probability range;
Table 1, Fig. 5).

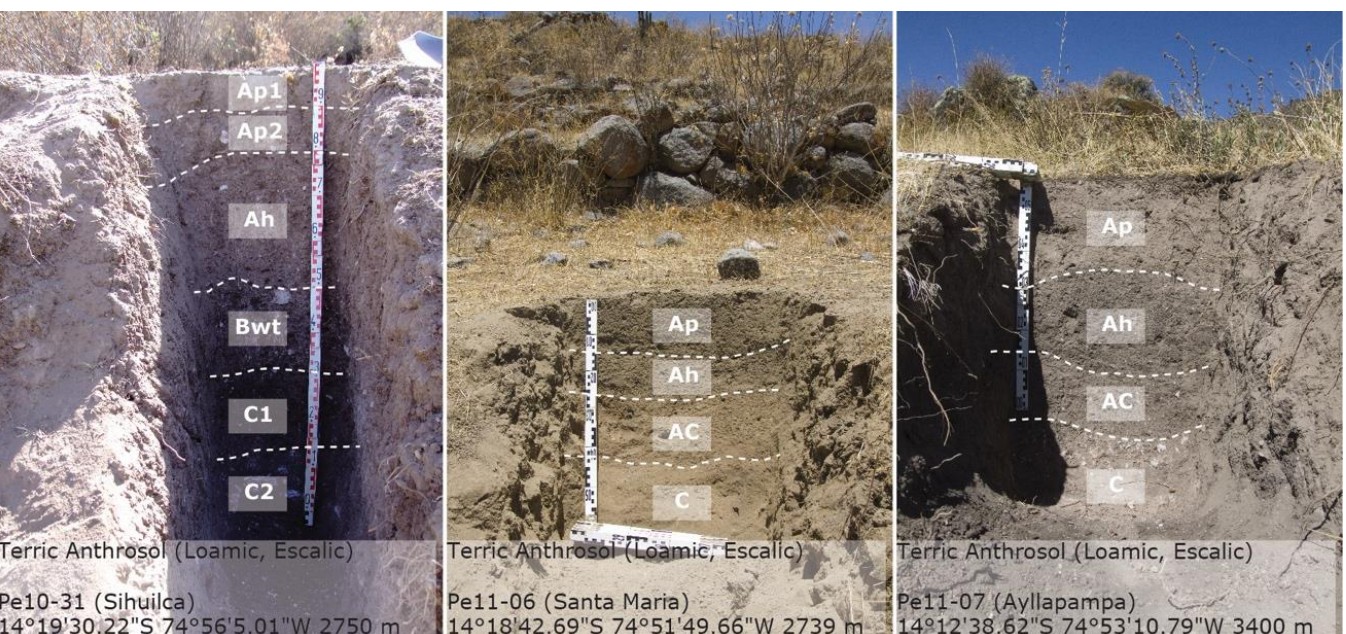

**Figure 4: Photographs of the terrace soils Pe10-31, Pe11-06 & Pe11-07.**

While the highest **CEC$_{pot}$** values are found in the surface horizons Ap1 and Ap2 (16.3 and 16.9 cmol/kg), the underlying Ah
has the lowest values (13.9 cmol/kg; Fig. 5). BS reaches very high values in all sampled horizons, starting from 99.4%. **Soil
pH** ranges from 6.1 in Ap1 to 5.4 in the Ah, where soil acidity is moderate to weak. **Organic matter** tends to decrease from
3.5% in Ap1 to 1.6% in C1. The lowest value (1.5%) was found in the buried Ah. The **C:N ratio** decreases gradually towards
the parent material and varies between 7.5 in the uppermost Ap1 and 6.4 in the C1. The **soil texture** of the uppermost Ap1 is
loamy. An increase in the relative proportion of clay fractions in all underlying horizons results in a change in texture class to
clay loam. **The skeleton fraction** of the topsoil horizons differs from the subsoil horizons by a higher content of coarse
fractions. While the proportions in the upper 60 cm range between 24.7 and 25.4 %, this proportion decreases to 19.3–14.2 %
in the lower part of the profile. The **plant available P fraction** remains consistently high (0.4 mg/g) at the cultivable Ap
horizon, and its concentration decreases drastically (0.1–0.0 mg/g) in the Ah and with increasing depth.

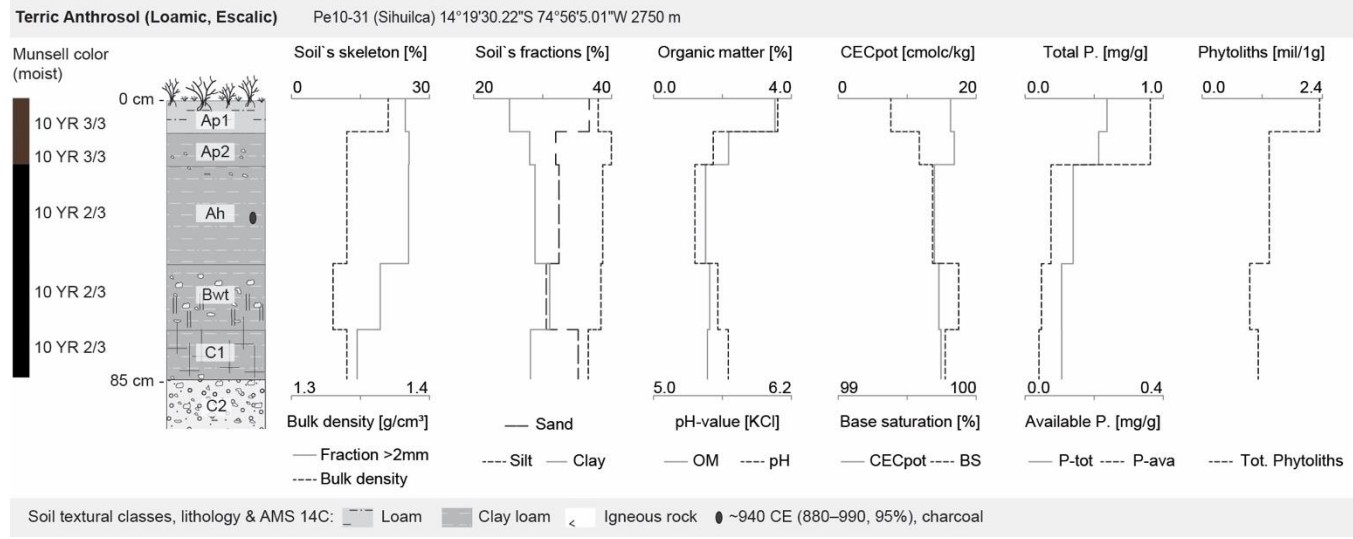

**Figure 5: Profile description and laboratory results for terrace soil Pe10-31: macro-morphology, pedo-chemical analytical data,
AMS $^{14}$C dating and total phytolith amounts.**

In the context of **phytolith analysis**, a total of 46 phytolith morphotypes were identified across all sites (Supplementary Tables
S3 & S4, Fig. 6). For simplicity, the following descriptions are based on the 20 most common morphotypes (Fig. 7). Ten out
of the 20 morphotypes can be attributed to the monocotyledonous class (Poaceae), while the remaining morphotypes may also
originate from dicotyledons. Four of the monocotyledonous (Poaceae) morphotypes can serve as diagnostic indicators for *Zea
mays* (i.e., RONDEL ELONGATED/ WAVY TOP, RONDEL TALL, BILOBATE LONG, CROSS). The frequency of these morphotypes
exhibits variation across sites.

In soil profile Pe10-31, **phytolith concentrations** varied from 2,343,877 to 950,881 phytoliths per 1 g of sediment (Fig. 5). The highest phytolith concentrations were found in Ap1 (2,343,877 phytoliths per gram of sediment). The lowest concentrations were found in Bwt (950,881 phytoliths per gram of sediment). ELONGATED SUM (mean = 22.6%, σ = 2.9, n = 5) are the most abundant **phytolith morphotypes**, followed by RONDEL (mean = 19.2%, σ = 0.6, n = 5) and RONDEL ELONGATED/WAVY TOP morphotypes (mean = 14.1%, σ = 2.7, n = 5). Another *Zea mays* diagnostic morphotype, RONDEL TALL (mean = 1.9%, σ = 1.1, n = 5), shows a significant occurrence, as well as POLYHEDRAL IRREGULAR SUM (mean = 3.9%, σ = 2.3, n = 5) and PERFORATED PLATELETS (mean = 0.4%, σ = 0.2, n = 5). BULLIFORM FLABELLATE (mean = 1.1 %, σ = 1.5, n = 5) occur significantly in the Ah horizon while PRICKLES (mean = 3.6 %, σ = 2.5, n = 5) show a significant increase in Bwt. In the tillage horizon Ap2, two **starch grains** were identified, while the Bwt horizon revealed the presence of one starch grain, likely corresponding to *Zea mays* starch grains (Fig. 6f; Gismondi et al., 2019).

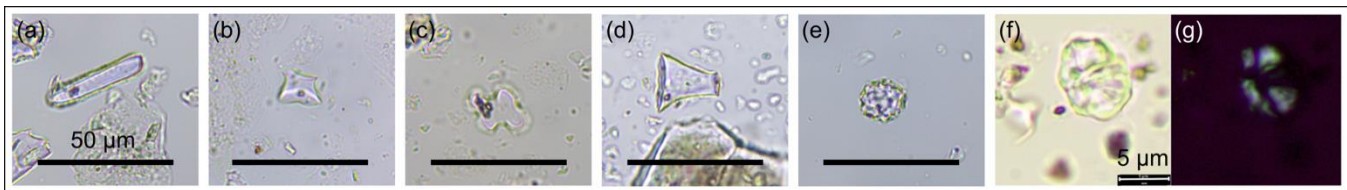

**Figure 6: Photomicrographs of selected phytolith morphotypes identified in the samples. The photographs were taken at 400× magnification. (a) RONDEL ELONGATED; (b) RONDEL WAVY TOP; (c) CROSS; (d) RONDEL TALL; (e) SPHEROID ECHINATE; (f) Starch grain from *Zea mays* under plane-polarized light; (g) same starch grain under cross-polarized light and on the same scale.**

### 4.2.2 Santa María terrace system: overview and soil profiles Pe10-26 & Pe11-06

The archaeological site of Santa María (PAP-788) was catalogued in 2006 through surveys carried out by the Nasca-Palpa Archaeological Project. Based on surface ceramics and architecture, PAP-788 is a Middle Horizon (600–1000 CE) and Late Intermediate period (1000–1450 CE) settlement. The site also has an Inka Late Horizon (1450–1535 CE) component (Reindel and Isla, 2013a; Soßna, 2015).

The Santa María terrace system spans an area of approximately 42 ha (0.42 km²), with around 39 ha (0.39 km²) clearly identified as terraced agricultural land (Fig. 8). Mapping efforts have revealed a total of 692 terraces, collectively measuring 33.80 km in length. These identified terraces predominantly face west and exhibit a generally heterogeneous pattern. Larger terraces are observed in flatter areas, while smaller terraces are found on steeper slopes. The width of the terraces varies from about 2 m to over 10 m.

Soil profiles Pe10-26 (14°18'43.83"S, 74°51'46.35"W, 2750 m asl) and Pe11-06 (14°18'42.69"S, 74°51'49.66"W, 2739 m asl) are positioned approximately 3.5 km southwest of the village of Laramate, situated along a northwest-exposed debris cone

(0.37 km², 2.55 km long, average slope: 6°) adjacent to the Laramate-Viscas River (Fig. 8). These profiles are located at upper and middle slope positions along a 100 m long, northwest-facing slope with an inclination of 19°.

The sample sites exhibit no signs of erosion since the abandonment of the terraces. Individual stones (6–20 cm) and boulders (20–60 cm) likely originate from collapsed terrace walls. The soil structure in both profiles is uniform in the upper 30–40 cm, characterized by coarse and blocky sub-angular peds with moderate to strong cohesion. Abundant roots with a diameter of 2–5 mm are present in the upper 40 cm of the profile.

In both cases, the soil type is classified as a *Terric Anthrosol* (*Loamic, Escalic*). Within a recurring profile sequence (Ap-Ah-
555 AC-C), up to two edaphic cycles could be identified. Minor differences in morphology between the two profiles primarily pertain to horizon thickness and the formation of horizon boundaries (Fig. 4, 9).

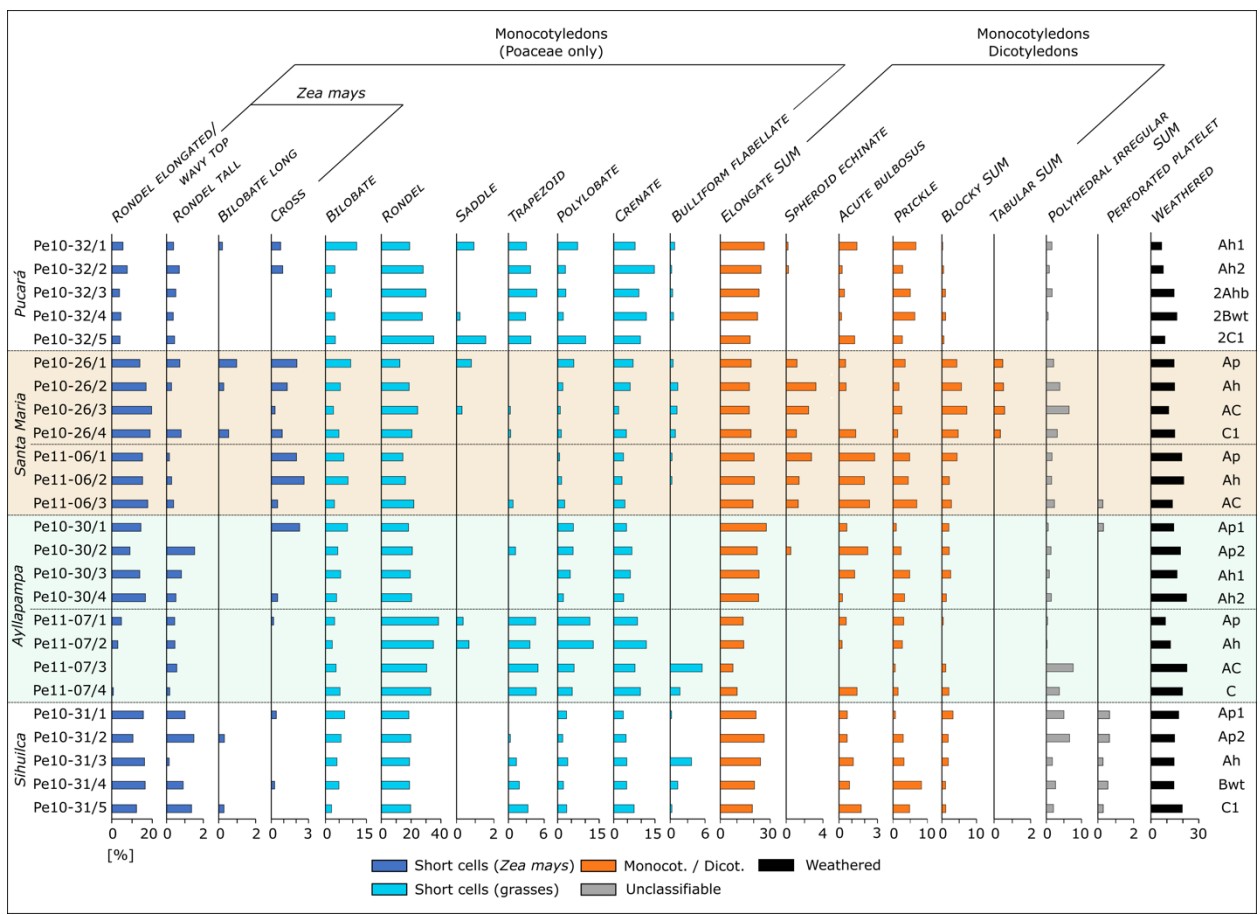

**Figure 7: Distribution of the most abundant phytolith morphotypes within the studied soil profiles in the study area.**

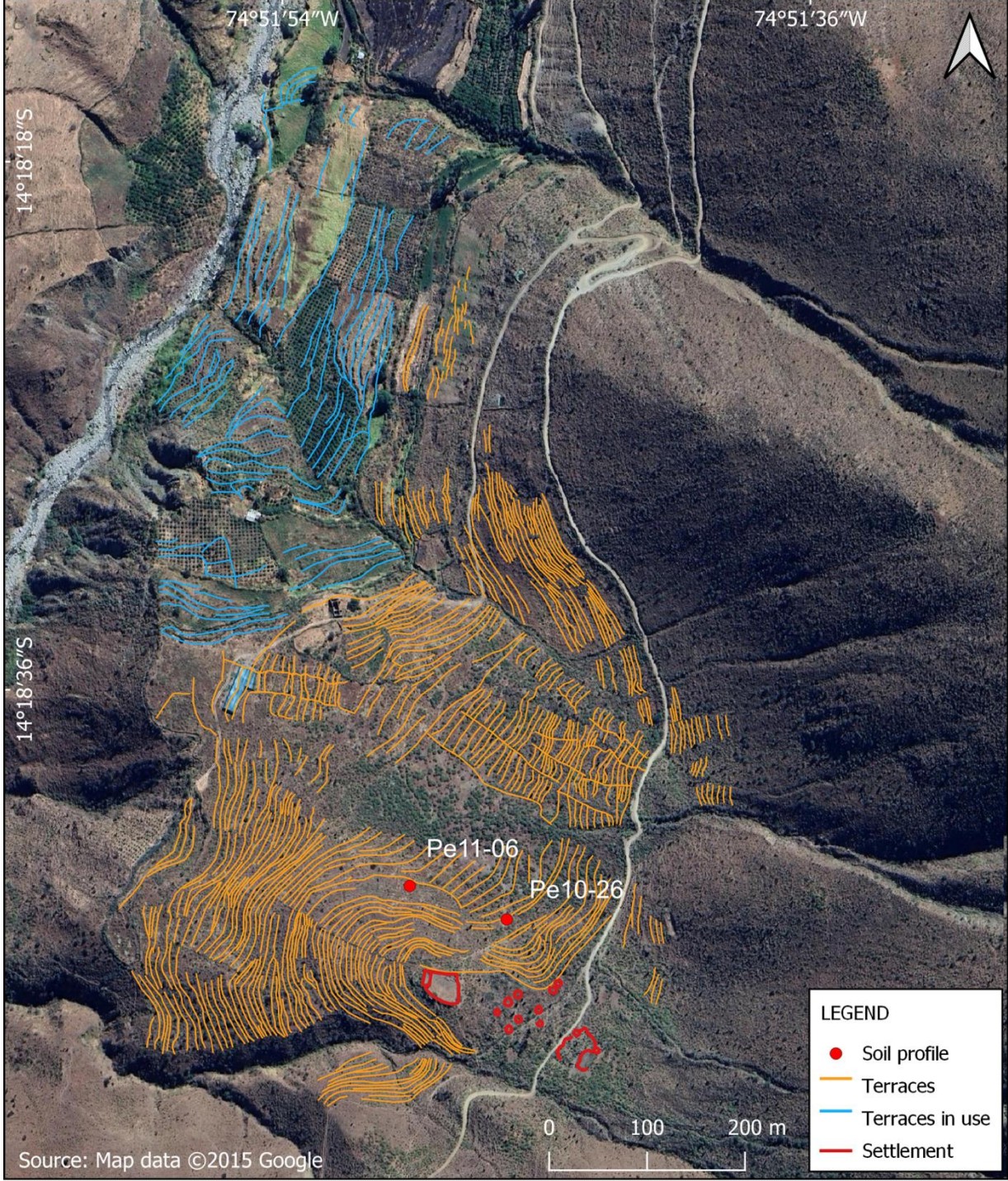

**Figure 8: Detailed map of the Santa María terrace system.**

Profile Pe10-26 (Supplementary Fig. 1) has a slightly compact, poorly developed and brownish-black 10 YR 2/3 **Ap** (0–15 cm) tillage horizon on its surface that contains human artefacts (Fig. 9). The underlying **Ah** (15–40 cm) is brownish-black in colour (10 YR 3/2) and has a higher humus content and a more developed soil structure. The horizon contains fragments of non-diagnostic pottery. It represents the former soil surface before terracing. The lower horizon boundary is very diffuse. The **AC** (40–60 cm) is a transitional horizon to the parent material, with a dark brownish-black (10 YR 2/3) colour and a very diffuse lower boundary. The horizon contains fragments of non-diagnostic pottery and charcoal particles and marks the lower limit of human influence on the soil. **C1** and **C2** (> 60 cm) consist of the parent material of the soil, which is slightly weathered quartzite. A [14]C-dated charcoal taken from the **C1** horizon (60–100 cm) returned an age of ~1760 CE (1650–1810, 95% probability range; Table 1, Fig. 9).

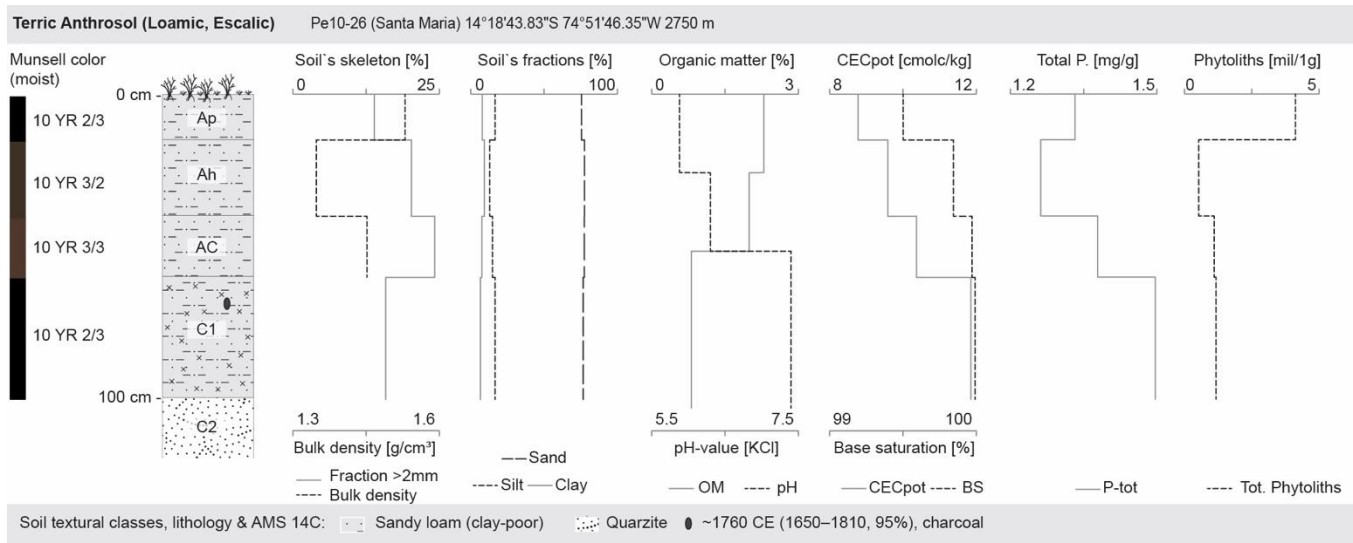

**Figure 9: Profile description and laboratory results for terrace soil Pe10-26: macro-morphology, pedo-chemical analytical data, AMS [14]C dating and total phytolith amounts.**

Profile Pe11-06 has a slightly more compacted, less developed and brownish-black (10 YR 2/3) **Ap** (0–15 cm) tillage horizon on its surface, which contains human artefacts (Fig. 4, 10). The underlying **Ah** (15–30 cm) is brownish-black in colour (10 YR 3/2) and has a higher humus content and a more developed soil structure. It represents the former soil surface before terracing. The horizon contains large quantities of domestic waste including charcoal particles, bone remains in a poor state of preservation and undiagnostic pottery fragments. **AC** (30–45 cm) is a transitional horizon to the parent material. It is brownish-black brown (10 YR 3/4) in colour and has a very diffuse lower boundary. It contains a layer of charcoal particles at 40 cm depth and non-diagnostic ceramic fragments. The horizon marks the lower limit of human influence on the soil. **C** (> 45 cm) consists of the parent material of the soil, which is composed of slightly weathered quartzite. A [14]C-dated charcoal taken from the **AC** horizon (30–45 cm) returned an age of ~1350 CE (1310–1410, 95% probability range; Table 1, Fig. 10).

The **CECpot** values differ between sampling sites, both in terms of absolute amounts and behaviour in the profiles (Fig. 9,
10). While the values in Pe10-26 are lowest in Ap and Ah (8.8 and 9.6 cmol/kg) and increase slightly towards the bottom (up
to 11.9 cmol/kg), the CECpot values in profile Pe11-06 are much higher and remain constant between 17.1 and 23.3 cmol/kg
throughout the profile. BS values are very high in all horizons at both sites, with an absolute minimum of 99.5%. **Soil pH**
values are very similar in both profiles and increase with depth. Values range from moderately acidic at the surface to slightly
alkaline in the soil parent material (pH 5.5 to 7.3 in Pe10-26; pH 5.9 to 7.4 in Pe11-06). **Organic matter** decreases gradually
in both profiles, with values ranging from 1.9% to 0.5% in profile Pe10-26 and from 2.3% to 0.8% in profile Pe11-06. **C:N**
**ratios** vary between the two sampling sites. The ratios in profile Pe10-26 are lower, with values ranging from 4.9 (Ah) to 7.3
(Ap). The ratios in profile Pe11-06 are significantly higher, with values between 12.8 (Ah) and 11.0 (AC). The Ap surface
horizons of both profiles are significantly more compact than the underlying Ah horizons, as indicated by the **bulk densities**
(Ap: 1.53 and 1.53 g/cm³ at Pe10-26 and Pe11-06 respectively; Ah: 1.35 and 1.31 g/cm³ at Pe10-26 and Pe11-06 respectively).
The **packing density** coefficients of the topsoil horizons reach a value of 1.6 (Ap, Ah) in profile Pe10-26, while in profile
Pe11-06 they show values of 1.7 (Ap) and 1.5 (Ah). The **soil texture** of both profiles is sandy loam with slightly varying clay
and sand contents. In profile Pe-11 the clay content is generally slightly higher and the sand content slightly lower than in
profile Pe10-26. The **skeleton fraction** marks a clear boundary between the Ap and the underlying Ah in both profiles. The
percentage of the coarse fraction (> 2 mm) increases from 13.9 % (Ap) to 20.2 % (Ah) in Pe10-26 and from 13.3 % (Ap) to
14.7 % (Ah) in Pe11-06. In soil profile Pe11-06, the **plant available P** content remains consistently high at 1.7 mg/g in the Ap
and Ah horizons and decreases to 0.9 mg/g in the AC.

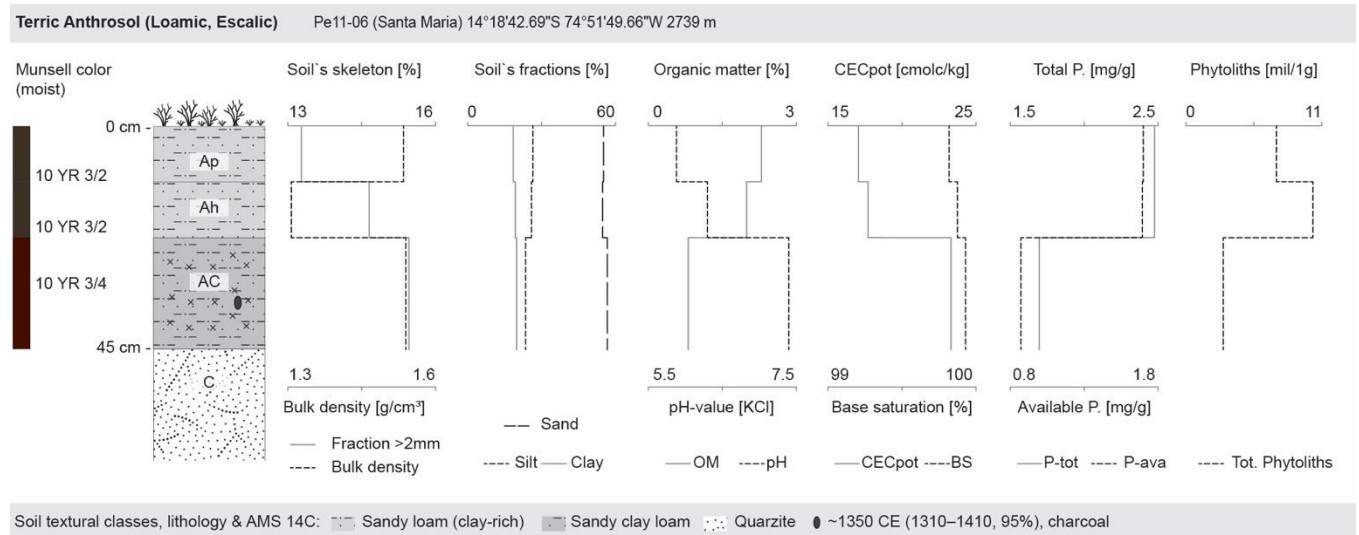

**Figure 10: Profile description and laboratory results for terrace soil Pe11-06: macro-morphology, pedo-chemical analytical data,**
**AMS [14]C dating and total phytolith amounts.**

**Phytolith concentrations** in Pe10-26 vary considerably (Fig. 9). The Ap shows an amount of 4,119,018 phytoliths per 1 g of sediment while the lowest concentration is found in Ah (532,325 phytoliths per 1 g of sediment). AC (1,106,241 phytoliths per 1 g of sediment) and C1 (1,175,349 phytoliths per 1 g of sediment) show a similar concentration. With respect to **phytolith morphotypes**, RONDELS (mean = 19.0 %, σ = 5.1, n = 4) are most abundant, followed by ELONGATE SUM (mean = 18.1 %, σ = 0.6, n = 4) and *Zea mays* diagnostic RONDEL ELONGATED/WAVY TOP (mean = 17.3 %, σ = 2.6, n = 4) morphotypes. Other diagnostic *Zea mays* morphotypes such as RONDEL TALL (mean = 0.9 %, σ = 0.7, n = 4), BILOBATE LONG (mean = 0.4 %, σ = 0.4, n = 4) and CROSS (mean = 1.1 %, σ = 0.7, n = 4), show a detectable signal. SPHEROID ECHINATES (mean = 1.9 %, σ = 1.0, n = 4), ACUTE BULBOSUS (mean = 2.2 %, σ = 0.5, n = 4), PRICKLE (mean = 4.8 %, σ = 0.9, n = 4) and BLOCKY SUM (mean = 5.5 %, σ = 1.3, n = 4) also occur in significant numbers. In the tillage horizon Ap, a single **starch grain** was detected, likely corresponding to *Zea mays* (Gismondi et al., 2019). Conversely, no starch was observed in horizon C1.

Profile Pe11-06 shows the highest phytolith concentration in Ah (10,277,195 phytoliths per 1 g of sediment) and the lowest in AC (3,017,753 phytoliths per 1 g of sediment; Fig. 10). In the Ap horizon (0–15 cm depth) an abundance of 7,326,396 phytoliths per 1 g of sediment was observed. Phytolith assemblages from profile Pe11-06 show similar trends to profile Pe10/26. ELONGATE SUM (mean = 20.4 %, σ = 0.4, n = 3) are the most common morphotypes, followed by RONDEL (mean = 17.5 %, σ = 3.9, n = 3) and RONDEL ELONGATED/WAVY TOP (mean = 16.0 %, σ = 1.5, n = 3). In addition, the *Zea mays* diagnostic CROSS (mean = 1.7 %, σ = 1.1, n = 3) and RONDEL TALL (mean = 0.5 %, σ = 0.2, n = 3) morphotypes occur significantly. SPHEROID ECHINATES (mean = 1.7 %, σ = 0.8, n = 3) are observed with similar frequency. ACUTE BULBOSUS (mean = 2.2 %, σ = 0.4, n = 3) and PRICKLES (mean = 4.8 %, σ = 1.2, n = 3) occur much more frequently than in profile Pe10/26.

### 4.2.3 Ayllapampa terrace system: overview and soil profiles Pe10-30 & Pe11-07

The archaeological site of Ayllapampa (PAP-967) was recorded in 2009 during surveys carried out by the Nasca-Palpa Archaeological Project. Based on surface ceramics and architectural remains, PAP-967 primarily dates to the Middle Horizon (600–1000 CE), but also has a Late Intermediate period (1000–1450 CE) component. PAP-967 shows the typical spatial layout of a Middle Horizon domestic settlement (Soßna, 2015).

The Ayllapampa terrace system encompasses an area of approximately 373 ha (3.73 km²), with around 191 ha (1.91 km²) clearly identified as terraced agricultural land (Fig. 11). A total of 2504 terrace walls have been revealed, collectively measuring 97.66 km in length. These identified terraces predominantly face west and exhibit a generally heterogeneous pattern. Larger terraces are observed in flatter areas, while smaller terraces are found on steeper slopes. The width of the terraces varies widely, ranging from about 2 m to over 10 m.

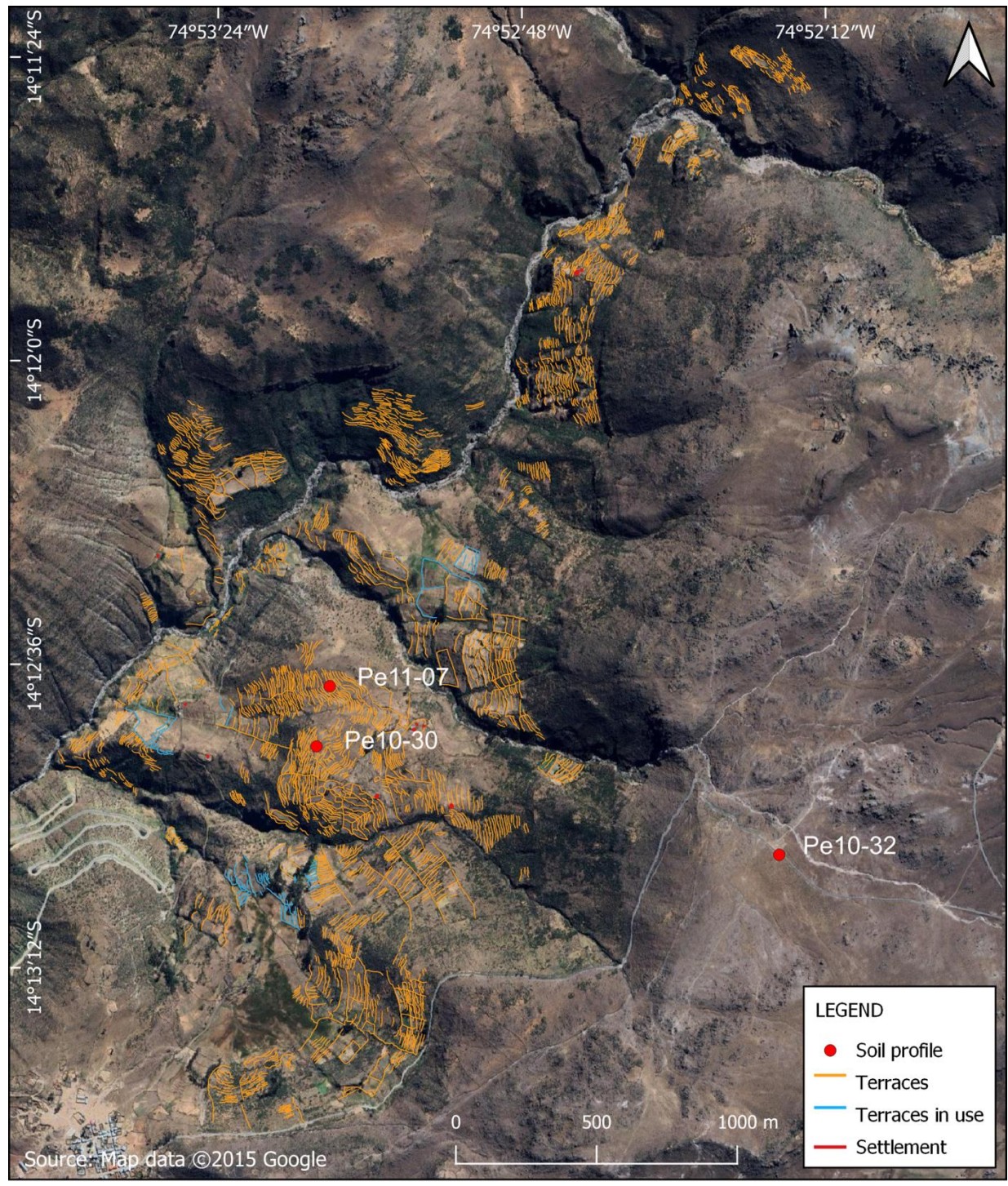

**Figure 11: Detailed map of the Ayllapampa terrace system.**

Profiles Pe10-30 (14°12'45.75"S, 74°53'12.35"W, 3410 m asl) and Pe11-07 (14°12'38.62"S, 74°53'10.79"W, 3400 m asl) are situated approximately 1.7 km northeast of the village of Pucará, on the central section of a west-facing debris cone (1.0 km², 4.5 km long, average slope: 17°) adjacent to the Palmadera-Palpa River (Fig. 11). Sampling was conducted at a mid-slope position on a 200 m long slope that faces northwest with an average slope of 18°.

The sample sites exhibit no signs of erosion since the terraces were abandoned. Isolated stones (6–20 cm) and boulders (20–60 cm) likely originate from collapsed terrace walls. The soil structure of profile Pe10-30 is uniform in the upper 65 cm, displaying a coherent massive structure with moderate cohesion. In contrast, the soil structure of Pe11-07 exhibits a coherent massive structure in the upper 25 cm and blocky subangular peds with moderate to strong cohesion at a depth of 25–50 cm. Roots with a diameter of 2–5 mm are present in the upper 70 cm of the Pe10-30 profile, whereas slightly more roots are found
in the upper 50 cm of profile Pe11-07.

In both cases, the soil type is classified as *Terric Anthrosol* (*Loamic, Escalic*). However, despite their proximity and the shared parent material, Pe10-30 and Pe11-07 exhibit notable morphological differences in terms of profile thickness, horizon boundaries, and the extent of intermixed soil material (Fig. 4, 12, 13).
The Pe10-30 (Supplementary Fig. 1) soil sequence consists of a 100 cm thick brownish-black topsoil (10 YR 2/3) composed of four horizons with diffuse boundaries (Ap1-Ap2-Ah1-Ah2) formed on the parent rock (Fig. 12). The fine-grained soil matrix is very homogeneous and mixed at least to a depth of 65 cm. The two tillage horizons **Ap1** (0–8 cm) and **Ap2** (8–35 cm) are distinguished by a slightly higher skeletal content in Ap1. Ap2 also contains non-diagnostic pottery fragments. Its lower
boundary is very diffuse. **Ah1** (35–65 cm) contains large amounts of domestic debris: charcoal particles, bone remains in a poor state of preservation, obsidian point fragments and undiagnostic pottery fragments. It probably represents the pre-terraced ground surface, which also served as a settlement area. Its lower boundary is very diffuse. **Ah2** (65–100 cm) is a transitional horizon to the parent material, which has a higher content of coarse fragments, but at the base of which terrace building material has also been deposited. It marks the lower limit of human influence on the soil. **C** (> 100 cm) consists of the parent material
of the soil, which is volcanic tuff. Two $^{14}$C-dated charcoals taken from the **Ah1** (35–65 cm) and **Ah2** (65–100 cm) horizons returned ages of ~930 CE and ~730 CE respectively (770-990 and 670-860 respectively, both with 95% probability range; Table 1, Fig. 12).

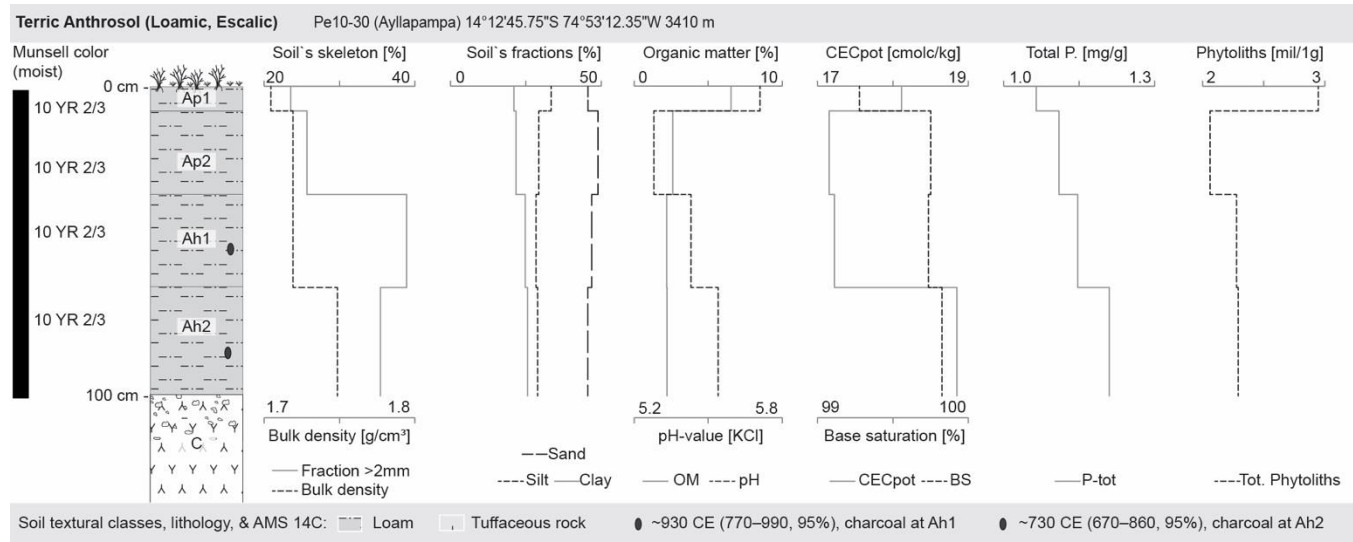

**Figure 12: Profile description and laboratory results for terrace soil profile Pe10-30: macro-morphology, pedo-chemical analytical data, AMS [14]C dating's and total phytolith amounts.**

The soil sequence of Pe11-07 begins with a fine-grained, slightly compacted and brownish-black (10 YR 3/1) tillage horizon **Ap** (0–25 cm). The underlying buried **Ah** (25–50 cm) is more developed and black (10 YR 2/1). Among other human artefacts (e.g., obsidian point fragments and non-diagnostic pottery) it contains a layer of burnt charcoal particles at a depth of 35–40 cm. The horizon represents the former soil surface before terracing. **AC** (50–60 cm) is a brownish-black (10 YR 2/2) transitional horizon to the parent material. It contains fragments of obsidian points and marks the lower limit of human influence in the soil. **C** (> 60 cm) consists of the soil parent material, volcanic tuff. Unlike profile Pe10-30, the terrace has no construction or fill material at its base. The colour of the substrate is brownish-black (7.5 YR 3/2). A [14]C-dated charcoal taken from the **Ah** horizon (25–50 cm) returned an age of ~1080 CE (1210–1000, 95% probability range; Table 1, Fig. 13).

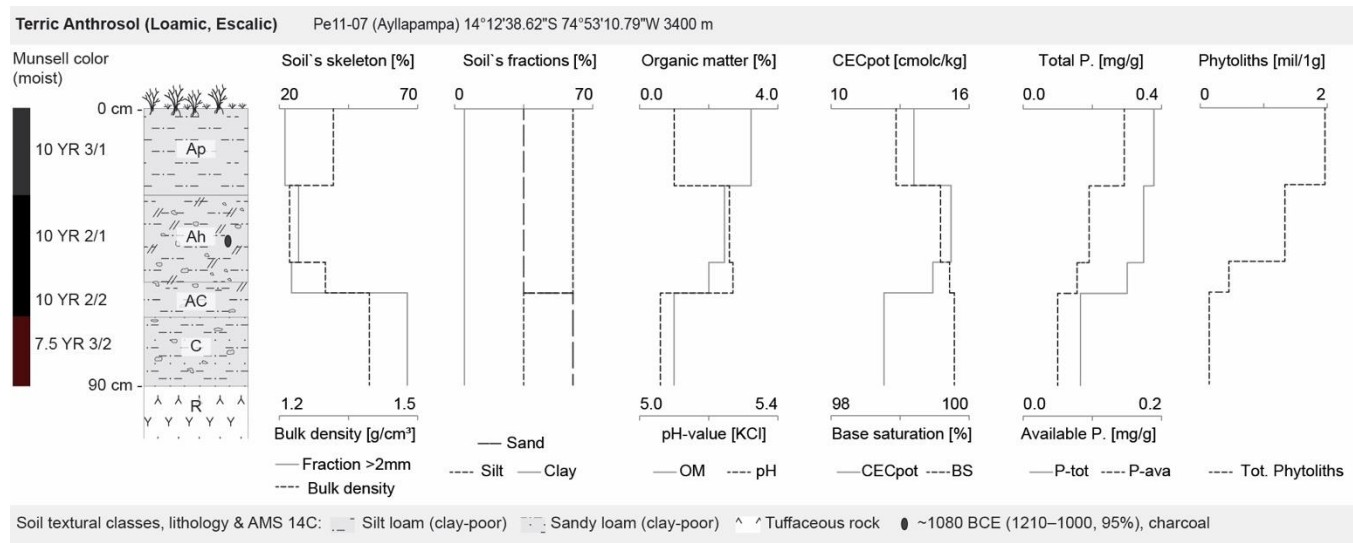

**Figure 13: Profile description and laboratory results for terrace soil profile Pe11-07: macro-morphology, pedo-chemical analytical data, AMS ¹⁴C dating and total phytolith amounts.**


The **CECpot** values differ at both sampling sites (Fig. 12, 13). While the values in profile Pe10-30 are slightly higher and the variations are small (from 18.1 in Ap1 to 18.8 cmol/kg in Ah2, but 17.2 cmol/kg in Ap2 and Ah), the CECpot values in profile Pe11-07 show a clear peak in the Ah-horizon (15.2 cmol/kg); the values are lower both in the parent material (12.3 cmol/kg in C) and at the surface (13.6 cmol/kg in Ap). BS reaches very high values in all horizons at both sites, with an absolute minimum

of 98.9% (Ap) in profile Pe11-07. The **soil pH** is moderately acidic in both profiles. It varies between 5.1 and 5.7. **Organic matter** decreases significantly with depth in both profiles, from 6.5% (Ap1) to 2.2% (Ah2) in profile Pe10-30 and from 3.2% (Ap) to 1.0% (C) in profile Pe11-07. The **C:N ratios** are different at the two sampling locations. While the highest value in profile Pe11-07 is found in the Ah (12.3), the Ah in the Pe10-30 profile has one of the lowest values (8.9) within the profile. In general, the ratios in Pe11-07 are slightly higher than in the corresponding horizons in Pe10-30. In both profiles there is a

tendency towards lower values in the Ap horizons compared to the underlying Ah horizons (from 8.1 to 8.9 in Pe10-30 and from 11.0 to 12.3 in Pe11-07). Soil compaction tends to increase slightly with depth in both profiles. Profile Pe10-30 has relatively constant **bulk densities** between 1.22 g/cm³ (Ap1) and 1.25 g/cm³ (Ah2). Profile Pe11-07, on the other hand, shows a much more compact Ap (1.32 g/cm³), while the underlying Ah horizon is less compact (1.22 g/cm³). The highest density (1.40g/cm³) was recorded in the C horizon. **Packing density coefficients** vary between 1.3 and 1.5 in both profiles. The **soil**

**texture** is loam in all Pe10-30 horizons, with relative proportions of sand ranging from 45.5 % to 48.9 %. The soil texture in profile Pe11-07 varies between silty loam in the Ap, Ah and AC horizons and sandy loam with up to 60% sand in the underlying C horizon. The **skeleton fraction** of the soil marks a clear boundary between the tillage horizons and the underlying horizons in profile Pe10-30. While the skeletal fraction (> 2 mm) varies between 23.5 and 25.7 % in Ap1 and Ap2, it increases significantly to 38.9 and 35.4 % in Ah1 and Ah2. Although not as pronounced as in profile Pe10-30, this pattern was also

observed in profile Pe11-07, where the proportion of the coarse fraction increases from 22.0 % in the Ap to 26.9 % in the Ah horizon. The **carbonate content** is 0% in both profiles. The plant available **P** fraction remains low (0.1 mg/g) from Ap to AC and does not have measurable concentrations in the C horizon.

**Phytolith concentrations** vary considerably across the Ayllapampa site (Fig. 12, 13). In Pe10-30, the highest amount of phytoliths was found in Ap1 (2,446,706 phytoliths per 1 g of sediment), followed by the lowest amount in Ap2 (1,560,504 phytoliths per 1 g of sediment), while Ah1 (1,778,108 phytoliths per 1 g of sediment) and Ah2 (1,791,331 phytoliths per 1 g of sediment) show similar concentrations. In terms of **phytolith morphotypes**, ELONGATE SUM (mean = 24.3 %, σ = 2.5, n = 4) are the most common morphotypes, followed by RONDEL (mean = 19.7 %, σ = 1.1, n = 4) and the *Zea mays* diagnostic morphotype RONDEL ELONGATED/WAVY TOP (mean = 13.4 %, σ = 3.2, n = 4). Another *Zea mays* diagnostic morphotype, CROSS (mean = 0.7 %, σ = 1.1, n = 4), also shows a notable occurrence in Ap1 (Fig. 7).

Phytolith concentrations in profile Pe11-07 decrease slightly exponentially with depth (Fig. 13). The highest number of phytoliths was reached in Ap (2,357,810 phytoliths per 1 g of sediment), while the C horizon had the lowest concentrations (168,678 phytoliths per 1 g of sediment). In profile Pe11-07, RONDEL ELONGATED/WAVY TOP (mean = 2.1 %, σ = 2.1, n = 4) morphotypes are rare (Fig. 7). RONDELS (mean = 34.4 %, σ = 3.4, n = 4) are the most common, followed by ELONGATE SUM (mean = 11.5 %, σ = 3.1, n = 4). Compared to profile Pe10-30, other Poaceae short cells, such as TRAPEZOID (mean = 4.4 %, σ = 0.6, n = 4), POLYLOBATE (mean = 9.1 %, σ = 4.0, n = 4) and CRENATE (mean = 9.4 %, σ = 1.8, n = 4) can be observed more frequently, while CROSS (mean = 0.0 %, σ = 0.1, n = 4) can be found in very low numbers in Ap. In AC, a significant occurrence of BULLIFORM FLABELLATE (mean = 1.8 %, σ = 2.6, n = 4) and POLYHEDRAL IRREGULAR SUM (mean = 3.2 %, σ = 3.8, n = 4) can be observed. WEATHERED MORPHOTYPES (mean = 16.2 %, σ = 6.4, n = 4) show an increasing concentration with depth.

### 4.2.4 Soil reference profiles Pe09-02 on plutonic acid & Pe10-32 on volcanic tuff

Profile Pe09-02 (14°16'22.98"S, 74°51'6.96"W, 3184 m asl) is located approximately 1.5 km northwest of the village of Laramate, situated within a modest south-facing catchment (1.8 km² and 6.2 km long) of a tributary to the Laramate-Viscas river system. Positioned in the middle of an 85 m long west-facing slope, the profile location features an inclination of 30°.

No signs of natural or anthropogenic soil erosion were observed at the profile location. Stones (6–20 cm) and cobbles (20–60 cm) are present in limited quantities on the surface. The soil structure is uniform in the upper 50 cm of the topsoil, but surface dry cracks (1–2 cm wide and 0.2–0.5 m apart) are commonly observed. Additionally, the topsoil exhibits coarse and prismatic peds with moderate to strong cohesion. The prevalence of roots with diameters ranging from 2 to 5 mm in the upper 60 cm of

the soil indicates recent biological activity. Bioturbation is evident through krotovinas, insect nests and burrows, as well as the presence of worms and larger animals.

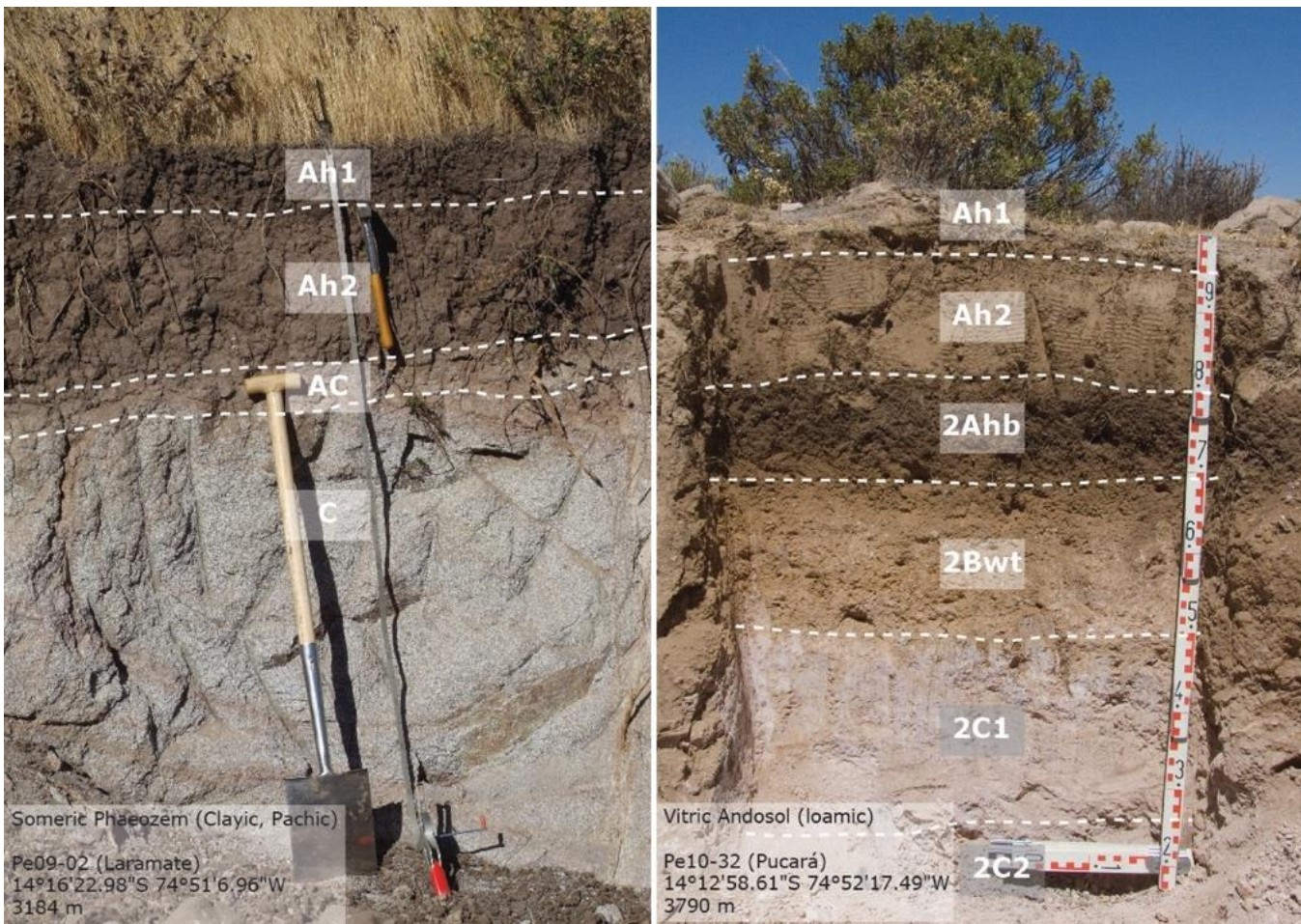

 **Figure 14: Photographs of soil reference profiles Pe09-2 (Laramate) and Pe10-32 (Pucará).**

The soil type is a *Someric Phaeozem (Clayic, Pachic)*. Four horizons (Ah1-Ah2-AC-Cw) can be distinguished (Fig 14, 15). **Ah1** (0–15 cm) is the first topsoil horizon, rich in humus and clay. Its brownish-black colour (7.5 YR 2.5/2) is due to fine-grained organic material. The lower boundary of the horizon is gradual and diffuse. The underlying **Ah2** (15–50 cm) differs slightly in its lighter brown colour (7.5 YR 3/3). Its lower boundary is diffuse and gradually merges into the transitional horizon **AC** (50–60 cm). The **C** (> 60 cm) represents the parent material of the soil, a granodiorite in a highly weathered state. The abrupt change in colour (10 YR 5/1) reflects the vertical boundary of pedogenesis.

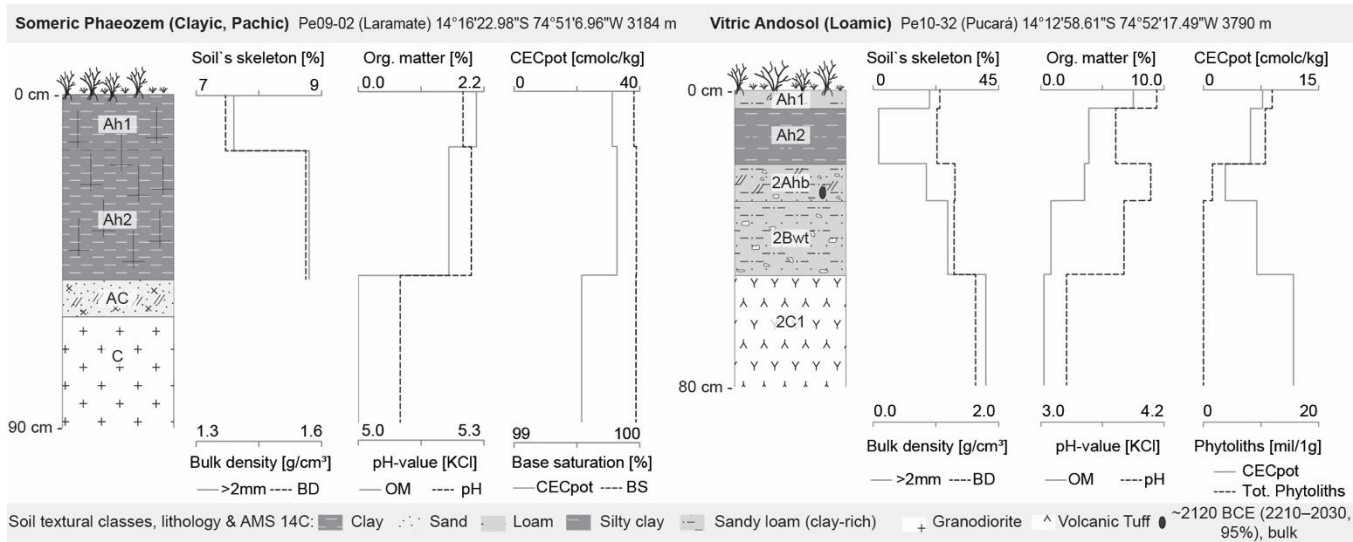

Figure 15: Profile description and laboratory results for the soil control profiles Pe09-02 and Pe10-32: macro-morphology, pedo-chemical analytical data, AMS $^{14}$C dating and total phytolith amounts.

The **CECpot** values are elevated throughout the soil profile (32.8–21.6 cmol/kg), with the highest values being found in the topsoil horizons Ah1 and Ah2 (Fig. 15). Soil saturation reaches a maximum of 100% in all horizons. The **pH values** are in the moderately acidic range and vary minimally between 5.3 (Ah) and 5.1 (C). The **organic matter content** in the soil profile decreases from 2.1% in Ah1 to 1.6% in Ah2. The **C:N ratio** shows values between 9.7 and 10.7 in the topsoil horizons Ah1 and Ah2. As the **bulk density** increases from 1.37 to 1.56 g/cm³, horizon Ah1 becomes slightly looser than the underlying horizon Ah2. The compaction in Ah2 shows an incipient restriction of plant root growth. The **soil texture** is silty clay in both the uppermost Ah1 and Ah2 horizons, with clay contents of 42.8% and 42.1% respectively. The proportions of sand and silt fractions alternate as their total contents reach 32.1% and 25.1% in Ah1 and 28.3% and 29.6% in Ah2, respectively. The **skeleton content** of the topsoil remains constant with values of 7.6 % in Ah1 and 8.8 % in Ah2. The profile has a **carbonate content** of 0%. The plant available **P** fraction increases slightly from 0.1 mg/g in Ah1 and Ah2 to 0.2 mg/g in C.

Profile Pe10-32 (14°12'58.61"S, 74°52'17.49"W, 3790 m asl) is situated approximately 2.8 km northeast of the village of Pucará in an elongated and west-facing catchment (4.9 km² and 9.2 km long) of a tributary to the Palmareda-Palpa river system. Sampling was conducted in the lower part of an isolated, northeast-facing 5 m slope with an inclination of 3°.

No evidence of natural or man-made soil erosion was observed at the profile location. Stones (6–20 cm) and cobbles (20–60 cm) are present in limited quantities on the surface. The topsoil (0–20 cm) exhibits a single grain structure with very weak cohesion. The underlying horizons (20–50 cm) are characterized by granular and blocky subangular clods with moderate

cohesion. Roots with a diameter of 2–5 mm are abundant in the upper 30 cm, while a limited number of insect nests indicate low animal activity.

The soil type is a *Vitric Andosol (loamic)* with a sequence of five horizons (Ah1-Ah2-2Ahb-2Bwt-2C) containing a buried
paleosol (Fig 14, 15). **Ah1** (0–5 cm) has a loose structure, a brownish black brown colour (10YR 3/4) and a high content of undecomposed plant material. **Ah2** (5–20 cm) is a fine-grained, loose and poorly developed topsoil, brown in colour (7.5 YR 4/4). A well-defined lower boundary with colour change marks a clear transition to the horizon below. The **2Ahb** horizon (20–30 cm) is a well preserved paleosol. It differs significantly from the topsoil horizons in its more developed and coherent soil structure, higher humus content and very dark brown colour (7.5 YR 2/3). The following subsoil horizon **2Bwt** (30–50 cm)
shows signs of alteration as indicated by its colour (7.5 YR 5/6). Its lower boundary is gradual and diffuse. **2C1** and **2C2** (> 50 cm) consists of the soil parent material, slightly weathered volcanic tuff which marks the lower limit of pedogenesis. A $^{14}$C-dated bulk sediment sample taken from the **2Ahb** horizon (20–30 cm) returned an age of ~2120 BCE (2210–2030, 95% probability range; Table 1, Fig. 15).

**CECpot** values in surface horizons Ah1 (7.7 cmol/kg) and Ah2 (6.1 cmol/kg) are higher than in the underlying 2Ahb (2.8 cmol/kg, Fig. 15). The values increase with depth to 11.7 cmol/kg in 2C. **BS** is elevated throughout the soil profile (96.8–99.7 %) and shows a similar behaviour to the CECpot values. The **pH values** are in the strong to very strong acidic range, decreasing from 4.1 in the Ah1 and 2Ahb horizons to 3.3 in 2C. **Organic matter** decreases with depth from 7.5 % (Ah1) to 0.3 % (2C1), with the highest values found both in the topsoil (Ah1 and Ah2) and in the buried horizon (2Ahb). The **C:N ratio** decreases
with depth, ranging from 9.5 (Ah1) at the soil surface to 3.4 (2C1) in the parent material. The surface horizons Ah1 and Ah2, with **bulk densities** of 1.06 and 1.01 g/cm$^3$, are significantly looser than the underlying horizons 2Ahb and 2Bwt, with densities of 1.30 and 1.29 g/cm$^3$. **Packing density coefficients** vary between 1.3 and 1.4.
The **soil texture** is loam and silty clay in Ah 1 and Ah 2, with high silt contents between 39.1 % and 44.5 %. A marked decrease in silt content and a simultaneous increase in sand content to 69.6–60.2 % in the underlying 2Ahb, 2Bwt and 2C1 horizons
characterize their texture as sandy loam. The **skeleton fraction** of the soil marks a clear boundary between topsoil and subsoil, as shown by the low percentage of only 1.9 % coarse fraction in Ah2, followed by an abrupt increase to 19.0 % in 2Ahb and 40.4% in 2C1. The profile has a **carbonate content** of 0%. Plant available **P** concentrations show very low peaks in Ah1 and 2Ahb and are not measurable in the rest of the profile.

**Phytolith concentrations** in profile Pe10-32 varied significantly from 11,925,773 to 18,690 phytoliths per 1 g of sediment, following an exponential decrease with depth (Fig. 15). The highest numbers of phytoliths were found in the upper Ah1 horizon (11,925,773 phytoliths per 1 g of sediment), while the lowest concentrations were found in the 2Bwt horizon (18,690 phytoliths per 1 g of sediment). In terms of **phytolith morphotypes**, RONDELS (mean = 28.0 %, σ = 5.9, n = 5) are the most abundant

morphotype and show a slight increase with depth, followed by ELONGATE SUM (mean = 23.2 %, σ = 3.2, n = 5), which also
decrease slightly with depth. In general, Poaceae short cells such as TRAPEZOID (mean = 3.6 %, σ = 3.6, n = 5), POLYLOBATE
(mean = 5.0 %, σ = 3.6, n = 5) or CRENATE (mean = 10.5 %, σ = 2.7, n = 5) are common, whereas diagnostic short cells of *Zea*
*mays*, such as RONDEL ELONGATED/WAVY TOP (mean = 5.0 %, σ = 1.5, n = 5), RONDEL TALL (mean = 0.9 %, σ = 0.3, n = 5)
or CROSS (mean = 0.3 %, σ = 0.5, n = 5), are less frequent. WEATHERED MORPHOTYPES (mean = 10.9 %, σ = 4.4, n = 5) increase
with depth. In horizon 2Ahb, no **starch grain** was detected.

## 5 Discussion

### 5.1 Geomorphology of the terrace agricultural systems

Regarding the geomorphology of the terrace agricultural systems in the Laramate region, we identified patterns that enhance
their optimal functioning. Their topographical locations often act as solar shelters, with a dominant western exposure and
positions on the middle and lower slopes, as well as at the bottom of valleys. These locations mitigate the intense solar radiation
of the inner tropics, favouring vegetative development by reducing evapotranspiration and increasing soil moisture availability.
The largest systems are situated in geomorphological settings characterized by reduced morphodynamics, such as lower slopes
and valley bottoms. Here, the gentle slopes minimize erosion potential. Additionally, these terrace systems are typically built
on colluvial deposits, debris cones, and fluvial terraces. Agriculture in these geomorphological contexts is profitable due to
the generally increased moisture retention capacity and the reduced energy required for ploughing on the less compacted soil
substrates.

### 5.2 Reference Soil Groups in the Laramate region

In the following discussion, our focus will be on the WRB Reference Soil Groups, namely *Phaeozems*, *Andosols*, and
*Anthrosols*. These soil groups are characteristic of the Laramate area and align with the soil profiles under examination.

### 5.2.1 Phaeozems

Mid-latitude *Phaeozems* are typically described as soils found in subhumid steppe margins within regions featuring vegetation
ecosystems ranging from long grass steppe to forested steppe (IUSS Working Group, 2015; Zech et al., 2014). In the Laramate
region, *Phaeozems* exhibit a distinctive Ah-(AC)-C profile sequence, marked by a dark and humus-rich surface horizon,
significant organic matter accumulation in the mineral topsoil, and a notably high BS in the upper meter. While Laramate
*Phaeozems* share similarities with well-described mid-latitude grassland steppe soils in their exceptional fertility, they deviate
from the typical descriptions by not developing under subhumid conditions. Laramate *Phaeozems* are entirely devoid of
carbonates throughout the profile, displaying no leaching features or accumulation of secondary carbonates. These soils

originate from intensely weathered granodiorites and tonalites, as well as various unconsolidated parent materials such as reworked volcanic conglomerates and alluvial deposits within the mountain shrubland of the Peruvian Andes. The absence of secondary carbonates precludes categorization as *Chernozems* or *Kastanozems*, related steppe soils. The diagnostic presence of subsoil secondary carbonates (*protocalcic* properties or *calcic* horizons) characterizes *Chernozems*, while *Kastanozems* are distinguished by greater amounts of secondary carbonates (IUSS Working Group, 2015).

A characteristic shared by all fertile steppe soils is the development of a *mollic* horizon in the advanced stages of pedogenesis (Blume et al., 2010; Eitel, 1999; IUSS Working Group, 2015; Zech et al., 2014). In profile Pe09-02, the presence of a *mollic* horizon serves as the diagnostic feature for classifying it as a *Phaeozem*. This near-surface mineral horizon is defined by a coherent and homogeneous substrate structure, an organic carbon content ranging from 0.8 to 1.0%, a dark-coloured substrate with Munsell colour and chroma values of 3 or less, a total horizon thickness of 50 cm, and a BS of ≥ 50%. In the tropical Andes, the development of *mollic* horizons is influenced by seasonality, which dictates the alternation between two phases responsible for organic matter accumulation in the topsoil within an annual cycle. The first phase involves the production of organic matter and the accumulation of undecomposed litter. The second phase is linked to bioturbation and the intensive enrichment of humified organic matter (Eitel, 1999; Zech et al., 2014). In the Laramate Region, the first phase coincides with a peak in soil moisture related to monsoonal rainfall activity during the summer months. Humid conditions lead to rapid vegetation growth, including short and medium grasses, deciduous, and xeromorphic shrubs, promoting the production of substantial biomass. The accumulation of undecomposed organic material in the topsoil and upper subsoil layers predominates, although partial decomposition of the litter is stimulated by increased microbial activity during the warm and humid summer periods (Blume et al., 2010; Eitel, 1999; Zech et al., 2014). The onset of the dry season in April and May initiates significant changes in the physical, chemical, and biological dynamics of the soil substrate. The colder temperatures during early winter months inhibit microbial activity and decomposition. Conversely, grazing mammals, rodents, and invertebrates intensify their burrowing activity as a means to escape extreme environmental conditions, including increased solar radiation in the absence of cloud cover, rapid nocturnal temperature drops, and frequent frosts. This activity results in the displacement and mixing of undecomposed organic material from the topsoil into the subsoil up to a depth of 1 m, where humification and some remineralization processes occur (Eitel, 1999). The presence of krotovinas, insect nests, worm remains, and burrows of larger animals in the topsoil of the soils analysed in the Laramate region attests to the intensive soil mixing by the local fauna. Facilitated by the loose soil structure, Andean mice (*Phyllotis spp.*), Viscacha (*Lagidium peruanum*), Zarigüeya (*Common opossum*), Andean fox (*Pseudalopex culpaeus*) and taruca (*Hippocamelus antisensis*) are particularly effective in this process (Brack Egg and Mendiola Vargas, 2010). (Eash and Sandor, 1995) identified analogous high-altitude soils in the Colca Valley, classified as *Mollisols* according to the US classification of soil taxonomy. These soils exhibit similar characteristics of *mollic* horizons as observed in the Laramate area. The authors also propose that slower decomposition of organic matter at lower

temperatures at high altitudes and the production of grass biomass play pivotal roles in pedogenesis and the development of high organic matter levels.

In the context of the prevailing aridity in the Laramate region, processes such as leaching and upward movement of water play a minimal role in pedogenesis. Additionally, further profile differentiation of Laramate *Phaeozems* is infrequent due to the
heightened water-holding capacity of the topsoil horizons, facilitated by humic substances, which restricts percolation and nutrient leaching. Furthermore, the presence of soil fauna contributes to the upcycling of non-weathered parent material into the upper soil layers (Eitel, 1999).

While $\delta^{13}C$ (AMS) numeric values lack standardization, rendering them incomparable with other published data, their relative
variations appear to reflect hydrological changes during prehispanic times. The drought during the Middle Horizon is suggested by less δ13C depleted values (Table 1), while the lowest $\delta^{13}C$ values correspond to more humid periods, possibly due to the adapted metabolism of CAM plants like *Prosopis juliflora*, which could have served as wooden fuel sourced from local vegetation. However, this hypothesis warrants further investigation through IRMS measurements in future studies.

**5.2.2 Andosols**

Different types of *Andosols* can be found in the southern Peruvian Andes, characterized by the volcanic source material (Zamora Jimeno and Bao Enríquez, 1972). In the Laramate region, *Andosols* typically develop on weathered volcanic material such as ashes, scoria, and vitric pyroclastics. Younger volcanic soils, identified as *Vitric Leptosols* or *Vitric Andosols*, are dominated by volcanic glass, glassy aggregates, and other glass-coated primary minerals. The presence of secondary minerals with higher concentrations of allophane clays and/or ferrihydrite indicates advanced stages of volcanic soil maturity,
classifying them as *Andic Andosols* (Zamora Jimeno and Bao Enríquez, 1972; Zech et al., 2014). In the Laramate region, the occurrence of *Andic Andosols* is less frequent due to prevailing semi-arid conditions. In this area, the most common and extensive soil association comprises initial *Leptosols* over volcanic material, along with more developed *Vitric Andosols*. The Pe10-32 soil profile in this region can be categorized as a *Vitric Andosol*, as confirmed by $Al_{ox} + \frac{1}{2}Fe_{ox}$ values exceeding the minimum threshold of 0.4% and phosphate retention values reaching the minimum of 25%.


The *Andosol* in profile Pe10-32, situated on a gently sloping plain, exhibits a well-preserved soil-paleosol sequence associated with two edaphic cycles. A fine-grained and relatively poorly developed 'Ah' topsoil overlays a well-preserved '2Ahb-2Bwt-2C1-2C2' paleosol, offering insights into the paleoenvironmental evolution of the Laramate region. Drawing on earlier investigations and the examination of geoarchives in the Palpa region (Mächtle, 2007), peat sediments from Cerro Llamocca
(Schittek et al., 2015), and paleosols in the Laramate area, it is postulated that there were approximately three pedogenesis phases in the study region during the Holocene. These phases were punctuated by two distinct dry periods between 2650–2250

BCE and 750–1200 CE (Fig. 16). The initial soil formation phase, characterized by heightened environmental humidity and geomorphodynamic stability, spanned from approximately 9050 to 2650 BCE and is deemed the most significant in terms of both duration and extent. It was during this phase that many of the *Phaeozems* and *Andosols* observed in the Laramate region formed. Subsequent soil formation phases were of shorter duration and lower intensity, making them difficult to detect.

The [14]C age of the fossil topsoil horizon '2Ahb' in Pe10-32 returned an age of ~2120 BCE, likely to represent the final manifestation of in-situ pedogenesis in autochthonous material before the horizon was buried by finer-grained sediments. However, the exact initiation time of pedogenesis remains uncertain. Studies of paleosols in the Colca Valley suggest that pedogenesis commenced in the middle Pleistocene, coinciding with the stabilization of land surfaces and glaciers not extending below 4000 m asl (Eash and Sandor, 1995). (Kappl, 2012) investigated soil development along a Pleistocene ground moraine sequence in the nearby Puquio region, approximately 150 km east of Laramate. The author postulates that the formation of soils with *andic* characteristics in the study area spanned approximately 40,000 years with the youngest *Vitric Andosols* developing over surfaces that became stable around 30 and 23.7 ka BP.

In the case of profile Pe10-32, sedimentological data reveals a distinct boundary between the uppermost Ah horizons (Ah1, Ah2) and the underlying paleosol sequence. The topsoil horizons exhibit a notably loose structure, low bulk density values, minimal skeleton fraction proportions, and elevated levels of silt and clay. A comparison of the observed grain size composition with aeolian sedimentary archives from the Palpa region indicates similar granulometric properties. This suggests that the fine-grained sediments in Ah1 and Ah2 primarily consist of allochthonous material transported aeolian, likely originating from the loess archives of the western Andean foothills referred to as 'desert margin loess' (Eitel et al., 2005; Mächtle, 2007). It appears plausible that during pronounced dry periods in the Late Holocene, desert loess at the base of the Andes was carried by the wind and redeposited in the highlands. However, mineralogical analyses of sediment origins in the Rio Grande basin have indicated that aeolian sediments in the Laramate region could also originate from the Eastern Cordillera of Peru (Brenner, 2011).

### 5.2.3 Anthrosols

In the terrace profiles, the presence of tillage horizons overlaying buried paleosols resulted in specific profile sequences: 'Ap-Ah-Bwt-C1-C2' in profile Pe10-31, 'Ap-Ah-C' in profile Pe10-30 and 'Ap-Ah-AC-C' in profiles Pe10-26, Pe11-06, and Pe11-07, classifying them as *Anthrosols*. Prolonged cultivation or the addition of materials can alter the properties of the topsoil horizons, but typically only to a certain depth. Consequently, the horizon differentiation of a buried soil may remain intact (IUSS Working Group, 2015). Morphological descriptions and analytical data indicate that all the investigated terrace soils are *Terric Anthrosols*, characterized by *terric* horizons with a total thickness of 50 cm. These *terric* horizons exhibit no clear

stratification, possess a colour resembling the parent material, and demonstrate a BS of ≥ 50%. No other diagnostic horizons for *Anthrosols*, such as *irragric*, *anthraquic*, *hydragric* or *hortic pretic* were identified.

## 5.3 Characteristics and effects of terrace soil management

### 5.3.1 Soil texture, skeleton fraction and bulk densities

The terrace soils in the Ayllapampa and Santa María systems consistently exhibit a lower proportion of the soil skeleton fraction (up to 8% less) in the arable Ap horizon compared to the underlying buried paleosol. This reduction in the soil fraction is likely attributed to the deliberate removal of gravel and stones during terrace construction. This practice can offer economic advantages by enhancing the terrace's water holding capacity and reducing tillage resistance. This strategy aligns with the findings of (Sandor and Eash, 1995) in the Colca Valley, where farmers placing gravel masses at the forefront of agricultural terraces, possibly to promote water drainage.

The soil texture of the investigated terrace soils is defined by various loam types, creating optimal conditions for soil moisture retention (Blume et al., 2010; USDA-NRCS, 1998b; Zavaleta García, 1992), as documented in various regional studies (Branch et al., 2007; Goodman-Elgar, 2008; Nanavati et al., 2016; Sandor and Eash, 1995). Inherent site conditions, notably the parent rock, contribute to minor distinctions between terraces. Although the water storage capacity of the Santa María terrace soils are comparatively lower, the Sihuilca and Ayllapampa terrace systems exhibit a slightly higher water storage capacity owing to elevated silt contents in the terrace fill deposits.

Concerning bulk densities, the terrace profiles show a distinct boundary between the tillage horizons (Ap) and the underlying topsoil horizons (Ah) of the buried soils, with higher bulk density values in the tillage horizons, as expected for long-term cultivated soils (Blume et al., 2010; USDA-NRCS, 1996b; Zavaleta García, 1992). Among all the studied terraces, the Santa María terraces display the most compact soils. The bulk densities of the topsoil horizons in profile Pe11-06 are even marginally higher than those in profile Pe10-26, potentially adversely affecting root growth. In contrast, the terrace soils in Ayllapampa show minimal compaction, creating favourable conditions for root growth (USDA-NRCS, 1996b). Overall, the slight variations in bulk density do not suggest that the individual terraces served purposes beyond agriculture.

### 5.3.2 Soil acidity, nutrient availability and soil quality

Soil acidity and CEC/BS are well-known indicators of soil quality (Blume et al., 2010; Zavaleta García, 1992). They are effective for assessing the soil's response to intensive agricultural practices and determining its suitability for certain crops, such as maize or potatoes. The terrace soils analyzed showed varied responses to historical agricultural practices.

The Sihuilca system stands out for its high soil quality. Notably, the pH of the tillage horizon (Ap) in profile Pe10-31 is markedly higher than the underlying Ah of the former soil surface and the pH values observed in the soil control profile Pe09-02. Additionally, the tillage horizon shows a substantial increase in available nutrients ($Ca^{2+}$, $K^+$, and $Mg^{2+}$), a contrast that is significant compared to the underlying Ah horizon. When combined with other soil parameters such as elevated total phosphorus and organic matter contents, these findings suggest that the Sihuilca terrace underwent intensive cultivation, directly meliorating the soil properties of the terrace.

Terrace profiles Pe10-30 and Pe11-07 in the Ayllapampa system differ significantly from their undisturbed reference profile Pe10-32. The observed variations in soil properties appear primarily attributed to local differences in the parent substrate on which the terrace soils and the reference soil developed (andesitic breccias and volcanic tuffs), rather than substantial alterations resulting from agricultural use. This assessment is grounded in the similarity between the buried paleosol sequences (Ah-AC-C) of the terrace soils and the characteristics of the cultivated tillage horizons (Ap). The soil quality of the tillage horizons is only marginally superior to that of the paleosols. In profile Pe10-30, the soil properties support agricultural use, as the tillage horizon (Ap) is less acidic and concurrently offers a higher quantity of nutrients compared to the underlying former soil surface (Ah1). The significant increase in the base cation $K^+$ is particularly noteworthy. In contrast, the tillage horizon (Ap) of profile Pe11-07 demonstrates incipient soil acidification and reduced nutrient availability, marked by a clear loss of basic cations $Ca^{2+}$ and $Mg^{2+}$.

The tillage horizons (Ap) within the Santa María terraces exhibit signs of incipient soil acidification and nutrient leaching, notably marked by a significant loss of alkaline $Ca^+$ cations. Soil acidification and nutrient leaching are common in various soil types under prolonged increases in soil moisture, such as those resulting from irrigation or more humid conditions (Zavaleta García, 1992). Lower pH values in the terraced topsoil horizons are also common in the Colca Valley and are probably related to higher levels of organic acids, carbonic acid or greater nitrification (Sandor and Eash, 1995). Given the prevailing arid conditions in the Laramate area, where soil acidification and nutrient losses are infrequently observed, the deficiencies noted in Santa María likely stem from the cultivation of the terrace system. It is plausible that livestock production in the region increased following the final abandonment of the agricultural terrace system, influencing soil pH conditions. Presently, the Santa María area is extensively utilized for livestock farming, with its gently sloping riverbanks and proximity to permanent water sources making it particularly appealing for grazing, especially during the dry season.

Overall, the studied terrace soils provide a conducive environment for agriculture. The soil's acidity and nutrient availability support the development of microorganisms, the establishment of robust roots, and the successful germination of seeds. This favourable environment facilitates the cultivation of staple Andean crops (Zavaleta García, 1992), including those identified in the region and associated with prehispanic agricultural systems, such as maize, potatoes, and alfalfa (Goodman-Elgar, 2008; Handley et al., 2023; Kendall and Rodríguez, 2009; Nanavati et al., 2016; Sandor and Eash, 1995). Since CEC/BS is determined by clay content, clay mineralogy, soil pH, and the amount of organic matter in the soil matrix (Blume et al., 2010; Zavaleta García, 1992), this parameter likely reflects efforts to counteract the potential negative effects of intensive cultivation, such as nutrient depletion or soil acidification, and to optimize crop production. The comparatively sandier terrace soils of Santa Maria benefited from the incorporation of organic matter into the topsoil and the mixing of the soil during terrace use; despite long-term use, soil acidification and nutrient losses are at non-critical levels. The significant increase in available nutrients in the terrace soils of Sihuilca supports this assumption.

In general, the terrace soils may have partially recovered since the abandonment of agricultural use, but assessing the extent to which current soil properties are due to post-agrogenetic processes—i.e., natural soil formation—is challenging. It is also unclear what level of soil degradation was present before abandonment. However, because terrace abandonment typically spans only a few centuries and natural soil formation is much slower than agropedogenesis (Kuzyakov and Zamanian, 2019), the influence of post-agrogenesis is likely still minimal. A valuable site to understanding the dynamics of agrogenetic and post-agrogenetic processes described by Kuzyakov and Zamanian (2019) is offered by the Patachana system (14°15'24 "S, 74°49'37 "W, 3425 m a.s.l.), a group of terraces not included in this study. This system has typological similarities with Sihuilca, Ayllapampa, and Santa María but features unique prehispanic irrigation characteristics. Unlike other systems in the Laramate region, which were largely abandoned after 1535 CE, the Patachana system continued to be regularly used during the colonial period and modern times. Therefore, the Patachana system is a valuable site for evaluating ancient agrogenetic soils without post-agrogenetic influences and should be considered in future research on terrace soil development.

### 5.3.3 C:N ratios, organic matter and organic phosphorus contents

C:N ratios in the Ap tillage horizons across all terraces reveal rapid mineralization rates, which differ from those observed in the underlying buried Ah horizons and their undisturbed soil context. These elevated mineralization levels are comparable to C:N levels found in terrace soils in the region (Nanavati et al., 2016; Sandor and Eash, 1995) and can be attributed to fertilization and selective cultivation.

In the Sihuilca terrace soils, both total carbon and total nitrogen content surpass the values of the *Phaeozem* reference profile, accompanied by relatively lower C:N ratios (7.5). The Santa María terraces, situated on crystalline parent material, exhibit

similarly low ratios (7.3-11.7) in the Ap horizons, yet possess higher total carbon and total nitrogen contents than the undisturbed *Phaeozem* reference profile. The increase in organic carbon and the greater amounts of nitrate in cultivated horizons can be attributed to long-term management practices involving the addition of organic matter and the cultivation of nitrogen-fixing crops (Sandor and Eash, 1995). Terraces located on volcanic soils within the Ayllapampa system display slightly higher C:N ratios (8.1-11.0) compared to the *Andosol* reference sequence and the terrace soils of the other systems. However, they have comparatively higher total carbon and total nitrogen contents. This difference may arise from the organic matter in these terraces not having fully decomposed, potentially associated with the spread of natural vegetation on less intensively cultivated land (Sandor and Eash, 1995).

The organic matter contents (1.9-3.5%) in the tillage horizons of all terraces are notably elevated, surpassing levels observed in both the undisturbed soil contexts and the underlying Ah horizons of the buried paleosols. Coupled with the absence of acidification and nutrient leaching, as well as the moderate to high C:N ratios observed in the studied terrace systems, these findings indicate generally favourable soil quality. They suggest that activities aimed at promoting the accumulation and maintenance of high organic matter content in the terrace topsoil were integral to prehispanic terrace management practices in the Laramate region. This aligns with findings from other studies on prehispanic agricultural terraces in neighbouring regions, such as the Colca Valley in Arequipa (Sandor and Eash, 1995) the Chicha Soras Valley in Apurimac (Branch et al., 2007; Handley et al., 2023) and the Viejo Sangayaico area in the upper Ica drainage (Nanavati et al., 2016). These studies propose that intensive agricultural use was accompanied by the systematic implementation of diverse fertilizing practices, including organic manuring, seasonal field burning, or manuring through grazing animals. While the terrace soils of the Colca Valley have undergone significant changes due to long-term fertilization since the Middle Horizon, leading to thickening of the soil and modification of its pedo-chemical properties, the terrace soils of Chicha Soras and Viejo Sangayaico seem to have been managed without heavy fertilization but rather through repeated additions of organic material. A study on Formative prehispanic agricultural soils in the Tafí Valley in the Argentinean region of Tucumán provides insights into the effects of long-term tillage and the use of fertilizers to compensate for soil deficits (Vattuone et al., 2011). Organic fertilization appears to have been a common prehispanic practice in the semi-arid Andes. This practice aimed to counteract the potential decline in organic matter and consequent deterioration in soil quality resulting from continuous cultivation and harvesting. The increased aeration of soils through tillage, coupled with the disruption of aggregates, exposes organic matter to aerobic conditions after tillage, stimulating microbial activity and accelerating its decomposition (Stevenson, 1982). Over time, tillage and the depletion of organic matter result in nutrient loss, increased compaction, diminished water infiltration capacity of the soil, and further erosion of soil cover (USDA-NRCS, 1996a).

The decomposition of organic matter in the soil releases nitrogen, carbon and organic phosphorus ($P_{org}$). Under optimal moisture conditions, carbon and nitrogen become rapidly available for plant uptake, while $P_{org}$ remains stable in an unavailable

fraction (Buckman and Brady, 1977). The accumulation of $P_{org}$ in the soil substrate may thus indicate practices involving the continuous and long-term incorporation of biomass (organic fertilization) into the soil to sustain consistently high levels of organic matter content (USDA-NRCS, 1996a, 1998a; Vattuone et al., 2011). The substantial buildup of $P_{org}$ in the Santa María and Ayllapampa terraces suggests prolonged agricultural use with ongoing application of organic manure. This interpretation is reinforced by increased bulk densities in the tillage horizons (Ap) of the terraces, particularly those with high $P_{org}$ contents (e.g., terrace soils Pe11-06 and Pe11-07). The observation that the younger terrace system of Santa María, dating to the Late Intermediate period, exhibits a stronger $P_{org}$ signal than the older terrace system of Sihuilca, dating to the Early Horizon, could have multiple causes. On one hand, the Sihuilca system is situated in an area where terrace soils formed on *Phaeozems* with excellent agroecological characteristics, potentially requiring little artificial improvement of soil properties through fertilization. On the other hand, the Santa Maria system likely saw more intensive terrace cultivation during a cultural period when such practices were more prevalent.

Moreover, the replenishment of soil nutrients through seasonal burning of fields is highlighted as a prehispanic agricultural practice in the region (Handley et al., 2023; Nanavati et al., 2016). However, the limited presence of charcoal particles within the Laramate terraces and the absence of burnt soil layers hinder the reliable attribution of charred material to a specific soil horizon or cultural period. Consequently, there is no evidence supporting the notion that nutrient levels were sustained through seasonal burning in the Laramate region.

### 5.3.4 Phytolith concentrations and assemblages

The phytolith concentrations indicate clear soil mixing in profiles Pe10-26 and Pe11-06 from Santa María, Pe10-30 from Ayllapampa, and Pe10-31 from Sihuilca. This is evident as the concentrations are more uniformly distributed within the profile and are notably high at depth. Consequently, it can be inferred that the sampled terraces have been utilized for agricultural activities over an extended period, as the highest phytolith concentrations in natural soils typically occur in the upper humic horizon and decrease with depth (Qader et al., 2023; Strömberg et al., 2018). Interestingly, the terrace profiles from Santa María exhibit distinct differences. The concentrations in profile Pe11-06 are markedly higher than those in Pe10-26, despite both profiles being in close proximity to each other and at a similar distance from a settlement. The concentrations in both profiles suggest intensive terrace use, hinting at longer use for Pe11-06. However, it is worth noting that the C1 horizon (60-100 cm depth) of profile Pe10-26 appears to be approximately 400 years younger than the AC horizon (30-45 cm depth) of Pe11-6, introducing a potential contradiction to this theory (Table 1).

The situation differs in terrace profile Pe11-07 from Ayllapampa, where concentrations are high in the Ap horizon but sharply decrease with depth (Fig. 13). This mirrors the soil control profile Pe10-32 (Fig. 15), devoid of terracing and likely agriculture, suggesting that the Pe11-07 terrace might not have been used extensively for agriculture over an extended period (Qader et al.,

2023; Strömberg et al., 2018). In contrast, the phytolith concentrations of terrace profile Pe10-30 exhibit clear signs of soil mixing, indicating agricultural use (Fig. 12). This interpretation finds support in the higher abundance of C3 Pooideae diagnostic Grass Silica Short-Cell Phytoliths (GSSCP) such as RONDEL, TRAPEZOID, POLYLOBATE and CRENATE in profiles Pe10-32 and Pe11-07 (Fig. 7). Compared to other sites, the abundance of C3 Pooideae GSSCP is significantly higher. This grassland community is typical in environments of high elevations, increased moisture, and lower temperatures (Aleman et al., 2014; Bremond et al., 2008; Twiss, 1992). Given the proximity of Pe10-32 to the Ayllapampa system, the similarity of phytolith assemblages could be explained, among other factors, by their close location. Assuming that the terrace of profile Pe11-07 has not been used extensively for a long time, the natural vegetation of the surrounding area might have reestablished itself on the site. This interpretation is supported by the radiocarbon dating of the Ah horizons of both terrace soils. The Ah horizon of profile Pe11-07, situated beneath the tillage horizon, dates back to the Initial Period (1210–1000 BCE). Given that the subsequent Paracas period signifies the inception of the first agricultural terrace infrastructure, it is reasonable to assume that the terrace was constructed and utilized during this period, a hypothesis substantiated by the terrace architecture. In contrast, the Ah1 horizon of profile Pe10-30 is dated to ~930 CE, toward the end of the Middle Horizon (600–1000 CE). This Horizon is regionally known for a systematic expansion of terrace agriculture. The dating of terrace soil Pe10-31 from Sihuilca also falls within this period (Fig. 16). The similarity between these two profiles is evident not only in the phytolith concentrations but also in the phytolith assemblages, suggesting a similar use of the two terraces, e.g., maize cultivation. In contrast, the terrace of profile Pe11-07 may have been unused since its potential initial construction and use in the Paracas period. It is plausible that, for various reasons, some terraces were no longer used for agricultural purposes, while others continued to be cultivated or were recultivated.

A notably higher occurrence of the diagnostic morphotype RONDEL ELONGATED/WAVY TOP associated with *Zea mays* is evident at the other sites (Fig. 7). This may be related to maize cultivation and agricultural use of the terraces (Bozarth, 1998; Handley et al., 2023; Logan et al., 2012; Pearsall et al., 2003; Piperno, 2006), as well as to fertilization and mulching practices (Ranjan et al., 2017). Generally, there is a greater presence of phytoliths from maize cobs (RONDEL ELONGATED/WAVY TOP) compared to those from leaves and husks (CROSSES) in the profiles.

Interestingly, both Santa María profiles (Pe10-26 and Pe11-06) show a distinct signal of CROSS-shaped phytoliths, possibly originating from maize (Handley et al., 2023; Piperno, 2006). Andean maize typically produces medium-sized CROSSES and LONG BILOBATES (Logan et al., 2012), both of which are present in profile Pe10-26 from Santa María. The radiocarbon dates from Santa María suggest a prolonged period of terrace use or recultivation, with potential changes in cultivation or fertilization practices over time. In contrast to Sihuilca or Ayllapampa, where leaves and glumes seem to have been mostly removed, there is a clear presence of CROSS morphotypes indicative of maize leaves or glumes in the Santa María terrace profiles (Fig. 7).

Elevated phosphorus values in Pe10-26 and Pe11-06 also stand out compared to other profiles, potentially indicating fertilization with organic material like maize leaves or corncobs.

Another possibility is fertilization with camelid dung or the grazing of camelids on the terraces. Since maize could have been a component of the camelid diet, the presence of leaf phytoliths, especially CROSS-shaped morphotypes, might suggest such a

practice (Cadwallader et al., 2012; Finucane et al., 2006; Handley et al., 2023). However, it's essential to note that the CROSS-shaped morphotype alone cannot definitively identify maize, as other grasses of the Panicoideae family also produce such phytoliths (Logan et al., 2012). Therefore, a more precise method, such as discriminant function analysis, would be needed to distinguish CROSS-shaped phytoliths from *Zea mays* and determine their specific origin (Piperno, 2006). CROSS-shaped phytoliths from other Panicoideae could also have been part of the camelid diet or used as fertilizer.

Surprisingly, SPHEROID ECHINATE phytoliths, considered diagnostic for Arecaceae (palms) (Huisman et al., 2018; International Committee for Phytolith Taxonomy (ICPT), 2019), were discovered in both profiles from Santa María (Pe10-26, Pe11-06), while they were present in very low abundance in the soil control profile Pe10-32 and terrace profile Pe10-30 from Ayllapampa (Fig. 7). Palms typically thrive in lower, warmer, and moister forests east of the Andes in Peru (Henderson et al., 2019;

Huisman et al., 2018). However, (Handley et al., 2023), working in the nearby Chicha-Soras Valley, also found SPHEROID ECHINATE phytoliths in similar concentrations in their samples. They suggest that these palm remnants may result from human cultivation or the transport of plant parts such as fruit, leaves, or wooden tools. In comparison to the Arecaceae phytoliths described by (Huisman et al., 2018), the SPHEROID ECHINATE phytoliths in our study (Fig. 6e) resemble the forms of Ceroxylon alpinum (seed and wood). Nonetheless, there is the potential for confusion with other plant families producing SPHEROID

ECHINATE morphotypes, such as Bromeliaceae. Common genera such as Tillandsias belong to the Bromeliaceae. They are found in the nearby Lomas ecosystem but are also seasonally and locally established in the adjacent Cactus Belt (e.g. supported by various cacti species on the slopes or by *Orthopterygium huaucui* in lower valleys) (Beresford-Jones et al., 2015; Brack Egg and Mendiola Vargas, 2010; Mächtle, 2007; MINAM, 2015). Due to a possible overlap in size, SPHEROID ECHINATES cannot be conclusively assigned to either Arecaceae or Bromeliaceae, warranting further research (International Committee

for Phytolith Taxonomy (ICPT), 2019; Piperno, 1988).

### 5.3.5 Starch analyses

The identification of *Zea maize* starch grains (Gismondi et al., 2019) in the Ap tillage horizons of terrace profiles Pe10-31 in Sihuilca and Pe10-26 in Santa María aligns with the findings from phytolith analyses. This suggests that maize held significant agricultural importance in the region during prehispanic times, a fact supported by the findings on the agricultural terraces

around the nearby archaeological site of Cutamalla (Mader et al., 2024). The starch grain in the Bwt horizon (50-70 cm) of profile Pe10-31 could originate from the former soil surface (Ah 20-50 cm) during an early phase of cultivation prior to terrace

construction. This is more likely than a later incorporation from the uppermost Ap horizons. The mixing of the soil substrate in the 30 cm thick Ah horizon, and its subsequent translocation into the underlying Bwt horizon, is plausible given the combined action of the available tillage tools of the time (cf. Chapter 5.5) and the influence of local burrowing fauna. Therefore, we assume that the Sihuilca site experienced early agricultural use during the Late Formative/Early Horizon periods or possibly even earlier. The lack of starch in the C1 horizon of terrace profile Pe10-26 in Santa María indicates that the soil mixing process did not extend to the parent material. Overall, the relatively low starch content in the terrace soils is likely a consequence of unfavourable conservation conditions stemming from starch degradation and damage inflicted by soil bacteria (Hutschenreuther et al., 2017). The lack of maize starch in the control profile Pe10-32 corroborates the presumption that maize cultivation did not take place in this location.

### 5.3.6 Prehispanic agricultural practices in the Laramate region

Ethnohistorical studies on prehispanic agrarian technologies in the Laramate area (e.g., crops, tillage tools, irrigation techniques, fertilization techniques, crop rotation, surface burning) are unfortunately almost non-existent. Drawing from various ethnohistorical sources and their own research findings on prehispanic terraces in the nearby upper Ica and Paca valleys, Goodman-Elgar (2008) and Nanavati et al. (2016) highlight the advantages of non-mechanized tillage tools. Tools such as the *chaquitaclla*, which have been employed in the Peruvian Andes since ancient times and remain part of the current rural landscape, typically till the soil to a depth of 30 cm with their curved metal or wooden tips. In conjunction with another traditional hand-tillage tool, the *ranccana* or *rucana*, these tools facilitate soil cultivation with less destructiveness compared to industrial ploughing. It is reasonable to infer their utilization for the Laramate terraces. Future studies should aim to test this hypothesis and investigate its actual effects on the terrace soils of the Laramate region.

Maintaining consistent levels of organic matter through intentional organic manuring appeared crucial for preserving soil health. The presence of diverse plant elements in phytolith assemblages (e.g., maize leaves, glumes, and particularly corncobs) across all terrace systems suggests the deliberate application of plant material to the soil surface, most likely for fertilization and/or mulching purposes (Ranjan et al., 2017). The latter provides several benefits, including reduced evaporation, increased water infiltration, improved water retention, enhanced organic matter availability, temperature regulation, and surface run-off control (Blume et al., 2010). Typically, a combined cultivation and fertilization strategy benefits from in-situ biomass, which is produced in greater quantities within agricultural terraces compared to natural sites, and allochthonous organic matter, which is added as needed to meet the demands of intensive cultivation.

The potential role of organic fertilizers of animal origin (e.g., camelid dung) in the Laramate region warrants systematic investigation in future studies. Handley et al. (2023) found that in the Chicha Soras terrace system in the neighbouring Apurimac region, the nutrient content of the terraces was likely maintained in prehispanic times by periodic burning of the

terrace vegetation combined with the regular addition of camelid dung. In the Laramate terraces, the addition of organic

nitrogen sources (e.g., urea) was difficult to detect, as evidenced by the almost negligible total nitrogen contents. However, the total phosphorus contents are elevated in the profiles of the Santa Maria complex, particularly in profile Pe11-06 (Supplementary Table 1), which is not lithologically related to the outcropping quartzites or nearby plutonites. This suggests that the use of livestock manure is plausible and cannot be completely excluded, especially since a substantial number of prehispanic circular structures (Soßna, 2015), now utilized seasonally as corrals, remain well-preserved in the study area.

Quantifying these structures using GIS techniques could be a starting point for studies on domestic animal densities in an archaeological context. Moreover, using oribatid mite concentrations as an indirect indicator of manuring, similar to studies conducted in the Patacancha Valley in Cuzco (Chepstow-Lusty et al., 2007), could be considered in future work to better understand this hypothesis.

**5.4 Prehispanic history of terraced agriculture in the Laramate region**

Drawing on previous research and the findings of our study, the prehispanic history of terraced agriculture in the Laramate region can be broadly delineated into four phases, encompassing the initial establishment, systematic expansion, and eventual abandonment of the agricultural infrastructure at the beginning of the Hispanic colonial period (Fig. 16).

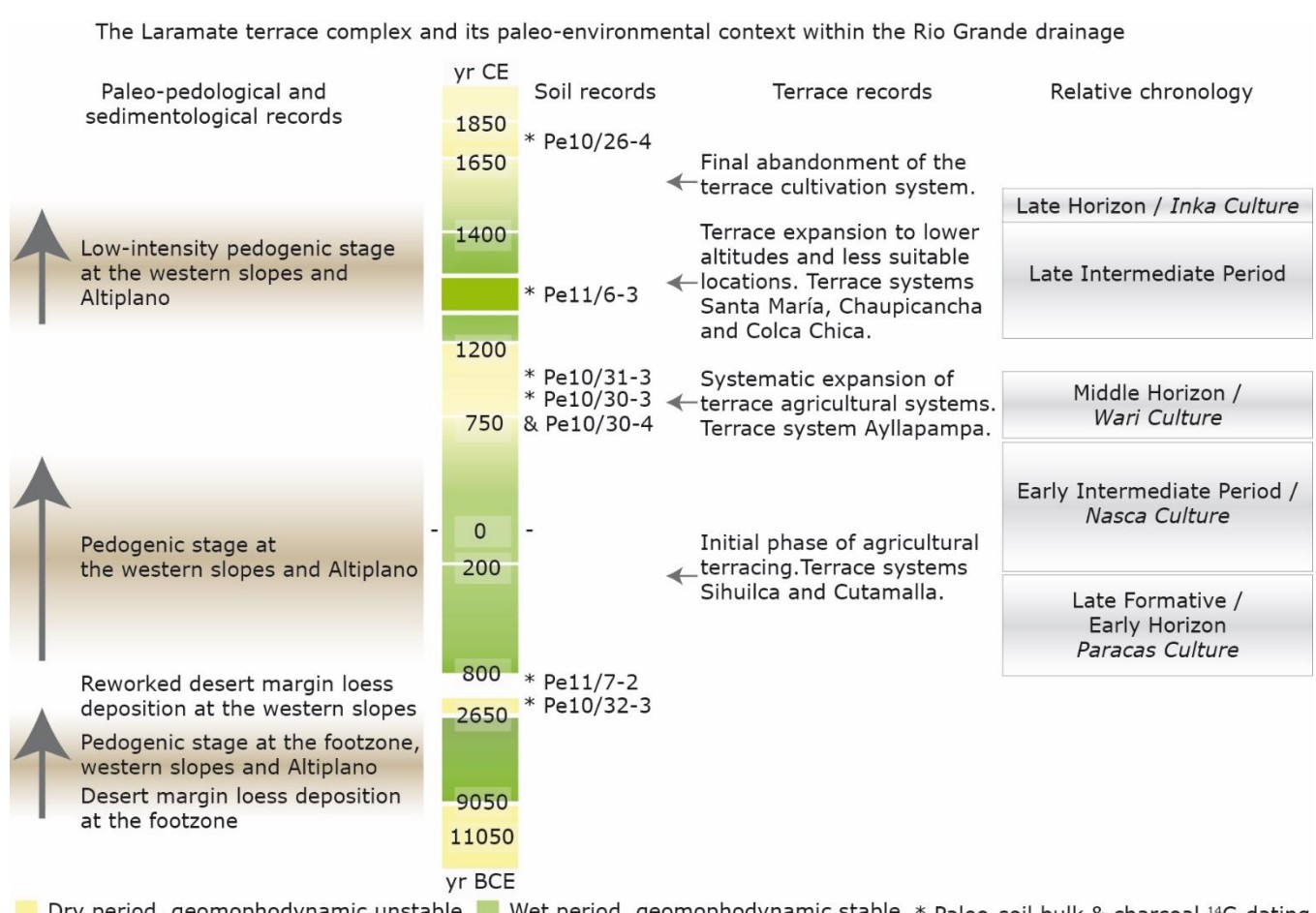

The Laramate terrace complex and its paleo-environmental context within the Rio Grande drainage

**Figure 16: Chronological model for the emplacement and use of the Laramate terrace cultivation system and its paleo-environmental context. Sources: relative chronology modified from Reindel and Isla, 2013a; Soßna, 2015; paleo-environmental and paleo-pedological records modified from Eitel et al., 2005, Mächtle 2007, Schittek et al. 2015 and own sampling; terrace records modified from Soßna, 2015, Mader et al. 2024 and soil records from the current paper.**

*(1) Initial phase of agricultural terracing in the Formative Late Paracas and transitional Early Nasca periods*

Archaeological evidence from the Laramate region indicates that the initial construction and use of agricultural terraces took place during the Formative Late Paracas period (Reindel and Isla, 2013a; Soßna, 2015). Recent geoarchaeological investigations at the archaeological site of Cutamalla and the surrounding extensive agricultural terrace systems, spanning approximately 220 hectares, revealed a contemporaneous occupation of Cutamalla alongside the utilization of the terrace systems. This agricultural terrace-settlement system was intensively employed for a relatively brief period of about 200 years (~250–40 BCE) during the Formative Late Paracas and transitional Early Nasca periods, with no evidence of reoccupation or subsequent reuse of the agricultural systems. The absence of identified irrigation canals suggests rainfed agriculture. The

architecture of the agricultural terraces in Cutamalla is notably similar and simple: the terrace walls with medium height were constructed using fairly large stones and boulders (up to 0.5 m in diameter), representing a typical architectural pattern of Formative agricultural terraces in the region (Mader et al., 2024).

Most of the terraces near the archaeological site of Sihuilca share a similar architectural style with those of Cutamalla. This resemblance implies their initial construction during the Late Paracas and/or transitional Early Nasca periods, a premise bolstered by the dating of the architectural remains of the PAP-888 settlement to the Nasca Early Intermediate period. However, unlike Cutamalla, modifications involving the use of smaller stones on some terrace walls were noted at Sihuilca, particularly in the upper slope areas. This observation suggests that parts of the terrace system were subject to reuse and modification during later settlement periods, a hypothesis supported by the [14]C dating of terrace soil Pe10-31, which yielded an age of ~940 CE, corresponding to the Middle Horizon.

Typical sites for these early terrace systems are usually those that maintained constant and high soil moisture during construction and use, particularly those exhibiting exceptional agro-ecological soil quality, as found in the soil province of *Phaeozems*. The Late Paracas and Initial Nasca periods were not only highly dynamic with significant population growth but also experienced more humid environmental conditions (Mächtle, 2007; Schittek et al., 2015), facilitating the widespread adoption of these agricultural practices in the region (Mader et al., 2024). Unfortunately, the conservation status of these old terraces is often poor due to extended periods of disuse.

*(2) Systematic expansion of terrace agricultural systems in the Middle Horizon*

The widespread and systematic expansion of terrace agriculture in the southern Peruvian Andes during subsequent prehispanic periods, especially in the Middle Horizon (600–1000 CE) and influenced by the Wari culture, has been extensively documented. Parallel developments have been observed in various regions, such as the Colca Valley in Arequipa (Sandor and Eash, 1995), the Chicha Soras Valley in Apurimac (Branch et al., 2007; Handley et al., 2023), and the Viejo-Sangayaico area in the upper Ica catchment (Nanavati et al., 2016).

In the upper Rio Grande basin, the expansion of agricultural systems linked to the Wari culture has been documented, focusing on the nearby Andamarca terraces (Aguirre-Morales, 2009). This trend is also noticeable in the Laramate area, especially around the Ayllapampa site. With the transition to drier environmental conditions and the gradual shift of the desert margin to the east, the importance of soil moisture for the construction and utilisation of terrace systems became crucial. Preferred sites, including some under irrigation, included the adjacent banks of major watercourses and areas with reduced solar radiation. Rather than solely depending on areas with *Phaeozems* (nutrient-rich soils), terraced agriculture on volcanic soils (*Andosols*)

at higher elevations became more widespread. This is supported by radiocarbon samples from the Ayllapampa Pe10-30 terrace profile, which have calibrated 95% probability ranges of 670–860 CE and 770–990 CE. The construction pattern of Ayllapampa terraces is distinct, featuring large stones at the base that, through subsequent modifications with smaller stones (as seen in Pe10-30), can reach medium to high wall heights of up to 1.3 meters. This suggests a later utilization of the terraces, likely extending into the Late Intermediate Period. Despite technological advances in terracing and irrigation techniques during the Middle Horizon, these innovations did not massively transform agricultural systems in the Laramate region. A few new irrigated terrace systems were established, but they represented only a minor proportion of the cultivated area.

*(3) Terrace expansion to lower altitudes and less agriculturally suitable locations during the Late Intermediate Period*

The Late Intermediate period (1000–1450 CE) was characterized hydrologically highly variable conditions, featuring phases of both dry and humid conditions (Eitel et al., 2005; Mächtle, 2007; Schittek et al., 2015). Particularly noteworthy were the positive effects of the humid conditions after 1250 CE on settlement activities and population growth in the Laramate region (Soßna, 2015). This period witnessed advancements in agrarian techniques and a substantial expansion of sophisticated farming terraces, as observed also in neighbouring regions (Branch et al., 2007; Handley et al., 2023; Nanavati et al., 2016; Sandor and Eash, 1995). As the desert fringe receded westward across the Cordillera Occidental (Eitel et al., 2005; Mächtle, 2007; Schittek et al., 2015), the adoption of terracing extended to lower elevations. Additionally, the heightened humidity facilitated the utilization of areas that were previously deemed less suitable for agriculture.

A striking illustration of the effective expansion of cropland in regions with less favourable agro-ecological conditions through terracing is the Chaupicancha system (Fig. 1). Presently, the system relies on seasonal water supply from two springs activated during the rainy season, though the total number of active springs was likely higher during the Late Intermediate period. The consistent availability of soil moisture compensated for deficiencies in soil quality (shallow topsoil, lower nutrient availability, and organic matter, low water retention capacity) and the somewhat unfavourable geomorphology of the site, marked by steep slopes and high solar radiation (Supplementary Fig. 2; Supplementary Table S1). The distinctive architecture of the terraces during this period, characterized by small terrace wall stones, and the absence of buried tillage horizons, indicate that the Chaupicancha terraces were initially built during this period and do not represent a re-use of older terraces. The Colca Chica terrace group, situated at the westernmost edge of the study area at lower altitudes (Fig. 1), exhibits similar characteristics. In this case as well, an area with unfavourable agro-ecological conditions was initially built and utilized during the Late Intermediate period.

Furthermore, some older agricultural terrace systems saw intensive agricultural activity during the Late Intermediate period. The Santa María terrace system, for example, displays indications of repeated use and abandonment spanning multiple cultural

periods. This is substantiated by observed modifications to the terrace structure, the archaeological settlement remains representing various ages, and the radiocarbon ages of the studied terrace soils, ranging from ~1350 CE in terrace profile Pe11-06 to ~1760 CE in profile Pe10-26. The Inka state of the Late Horizon (1450–1535 CE) had a minor presence in the Laramate region, however, Santa María is one of the few sites with remaining Inka architecture (Soßna, 2015).

*(4) Final abandonment of the terrace agricultural systems at the onset of the colonial period*

Terraces gradually fell into disuse and lacked systematic maintenance after the Late Intermediate period contemporaneous to a rapid decrease in rainfall (Schittek et al., 2015). With the advent of the Hispanic colonial period in 1535 CE, the local population was relocated, and the production system underwent reorganization (Aguirre-Morales, 2009; Nanavati et al., 2016).

Moreover, introduced diseases led to a demographic decline. This marked the beginning of the ultimate abandonment of the Laramate terrace complex. During the Spanish colonial administration, the terraced farming system lost its prominence, resulting in the terraces no longer being systematically used for agriculture.

## 6 Conclusion

In conclusion, our integrated pedo-geoarchaeological study has centered on three abandoned terrace agricultural systems and

their paleo-pedological context in the vicinity of Laramate, situated in the southern Andes of Peru (14.5ºS). The principal aim of this investigation was to untangle the pedological and land-use history of the Laramate area, which served as a significant agricultural hub during prehispanic times. This encompassed two key objectives: 1) contextualizing the former agricultural management systems within its geomorphological and paleoecological framework and 2) assessing the impact of agricultural practices on soil development and quality. The Laramate terrace complex emerged as an optimal setting for our inquiry, given

its diverse array of terrace installations and exploitation ages, coupled with varied geomorphological and geological settings. To evaluate the enduring effects on mountain soils resulting from ancient agricultural practices, a comparative analysis was conducted by comparing non-irrigated agricultural terrace soils with their undisturbed paleo-pedological counterparts. This comprehensive examination integrated a range of methodologies, including surveys, soil science techniques, GIS and remote sensing applications, paleobotany, and radiocarbon dating.

The Laramate region is characterized by three soil groups, specifically *Phaeozems*, *Andosols*, and *Anthrosols*. Notably, Phaeozems in this region exhibit a deviation from typical descriptions as they do not develop under subhumid conditions. The formation of their *mollic* horizon is strongly influenced by various factors, including climatic seasonality, dominant herbaceous vegetation, burrowing fauna, local lithology, and inputs of allochthonous aeolian material. In areas with volcanic material, the

prevalent soil association consists of *Leptosols*, along with more developed *Vitric Andosols*. The fossil topsoil in Pe10-32,

dating back to ~2120 BCE, represents the final stage of in-situ pedogenesis. The overlying fine-grained sediments consist of allochthonous material that was deposited after aeolian transport as a loess cover.

Detailed descriptions and analytical data suggest that all the terrace soils investigated in the area can be classified as *Terric Anthrosols*. Examining soil quality indicators of terrace soils reveals no significant signs of severe degradation, even with long-term use, though it is currently unclear how much the soils have recovered since the terrace systems were abandoned. The soil's balanced acidity and nutrient availability  support the successful cultivation of staple Andean crops. The terraces themselves are usually well preserved. Therefore, the final abandonment of the cultivation system is most likely not attributed to soil exhaustion or terrace structural instability, despite facing phases of landscape instability during dry periods.

The Sihuilca terrace system stands out for its notably high soil quality, evident in improvements such as increased water storage capacity and optimal soil acidity levels. The soil quality of the Ayllapampa terrace system is relatively neutral, with tillage horizons exhibiting a marginal superiority over underlying paleosols, as demonstrated by minimal compaction, lower acidification, and a higher quantity of available nutrients. In contrast, terrace soils at Santa María show signs of incipient deterioration; slight acidification, nutrient leaching, reduced water storage capacity, and compaction of cultivable horizons are evident in this area.

Ethnohistorical studies on prehispanic agricultural practices in the Laramate region are scarce. Drawing from research in neighbouring areas, it is inferred that traditional non-mechanized tillage tools, such as the *chaquitaclla* and *rucana*, were likely used for soil cultivation with minimal disruption. The tillage horizons of all terraces exhibit significantly elevated organic matter contents, surpassing both undisturbed soil contexts and buried paleosols. This suggests that the maintenance of soil health and sustainability of prehispanic agricultural practices in the Laramate region was closely tied to  intentional organic manuring of the terrace topsoil, as also evidencend by the substantial accumulation of $P_{org.}$ .

Phytolith concentrations in all terrace profiles strongly indicate soil mixing, signifying intensive agricultural activities over an extended period on the terraces. The presence of the diagnostic morphotypes RONDEL ELONGATED/WAVY TOP and CROSSES, associated with *Zea mays*, is more pronounced in terrace systems with presumed long-term agricultural use, emphasizing a connection to maize cultivation, fertilization with plant material and targeted mulching.  The identification of *Zea maize* starch grains in the tillage horizons at Sihuilca and Santa María aligns with phytolith analyses, reinforcing the significance of maize in the region during prehispanic times. The potential role of animal-origin organic fertilizers, like camelid dung, needs further investigation, as their use in nearby regions suggests a plausible but still poorly proven practice in Laramate. Notably, there is no evidence supporting the idea that nutrient levels were maintained through seasonal burning in the Laramate region. Future studies should focus on testing these hypotheses and exploring their effects on terrace soils. Irrigation played a minor role in these agricultural terrace systems.

In conclusion, the prehispanic history of terraced agriculture in the Laramate region unfolds across four distinctive phases, each reflecting the dynamic interactions between environmental, cultural, and agricultural factors. The initial phase, during the Formative Late Paracas and transitional Early Nasca periods, witnessed the establishment of agricultural terraces exemplified by sites like Cutamalla and Sihuilca. This humid period of intensive agricultural use is characterized by rainfed agriculture and simple terrace architecture. The systematic expansion of terrace agricultural systems marked the Middle Horizon (600–1000 CE), influenced by the Wari culture. The transition to drier environmental conditions prompted adaptations, such as terraced agriculture on volcanic soils as observed in the Ayllapampa terrace system. During the Late Intermediate Period (1000–1450 CE), characterized by hydrological variability, terraced agriculture experienced further expansion to lower altitudes and less agriculturally suitable locations. The Chaupicancha system serves as a notable example of effective cropland expansion in challenging conditions, reflecting more humid conditions and advancements in agrarian techniques. The final phase marked the gradual abandonment of terraced agricultural systems after the Late Intermediate period, particularly with the onset of the Hispanic colonial period in 1535 CE. Demographic shifts and reorganization of production systems led to the loss of prominence for the terraced farming system in the Laramate region, ultimately resulting in the cessation of systematic agricultural activities on the terraces. This historical trajectory underscores the adaptability and resilience of prehispanic communities in the Laramate region, who innovatively employed terraced agriculture to navigate changing environmental conditions and sustain agricultural practices across diverse landscape units.

**Data availability statement**

Data used in the study are available in the article and Supplement. Inquiries can be directed to the corresponding authors.

**Author contributions**

FL: conceptualization, fieldwork, methodology, data acquisition, data curation, formal analysis, data interpretation, visualization, funding, writing and editing – original draft, review and editing; CB: formal analysis, data interpretation, visualization, writing and editing – original draft; CM: methodology, funding, writing and editing – original draft; BM: conceptualization, fieldwork, methodology, data interpretation, writing – review and editing; EM: methodology, formal analysis, statistical analysis, writing – review and editing; LD: methodology, formal analysis, data interpretation; MR: funding, resources; BE: conceptualization, fieldwork, methodology, data interpretation, funding, resources, writing – review and editing; JM: conceptualisation, methodology, data interpretation, funding, resources, writing and editing – original draft, review and editing.

**Competing interests**

The contact author has declared that none of the authors has any competing interests.

**Disclaimer**

Publisher's note: Copernicus Publications remains neutral with regard to jurisdictional claims made in the text, published
maps, institutional affiliations, or any other geographical representation in this paper. While Copernicus Publications makes
every effort to include appropriate place names, the final responsibility lies with the authors.

**Financial support**

Fernando Leceta was financially supported by the German Academic Exchange Service (DAAD; Ref. Number A/08/98226)
and Heidelberg University. Moreover, this research was funded by the Groundcheck Program of the German Archaeological
Institute (DAI). This publication was supported by the Open Access Publication Fund of the University of Würzburg.
Additional support was provided by a Research Award (Julia Meister) of the Frithjof Voss Foundation.

**Acknowledgments**

Fernando Leceta would like to thank the German Academic Exchange Service (DAAD) for financial support (DAAD Ref.
Number A/08/98226), the soil laboratory staff at the University of Heidelberg, led by Gerd Schukraft (†), Nicola Manke and
Adnan Al Karghuli, as well as Dr. Volker Soßna for the exchange of ideas and support during the field campaigns. We would
also like to thank Santiago Ancapichún for calculating the air mixture at Cutamalla.

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
