# Peer review of "The impact of agriculture on tropical mountain soils in the western Peruvian Andes: a pedo-geoarchaeological study of terrace agricultural systems in the Laramate region (14.5°S)"

_EGUsphere, 2024_

## Referee Comment (RC1)

Title: The impact of agriculture on tropical mountain soils in the western Peruvian Andes: a pedo-geoarchaeological study of terrace agricultural systems in the Laramate region (14.5°S)
Author(s): Fernando Leceta et al.
MS No.: egusphere-2024-637
MS type: Original research article

The Manuscript contains new information; the Title corresponds to the content. The introduction reflects the content of the problem in the substantiation of the work, the literature reflects the world level of the study of the problem. The methods are described quite fully.

The Manuscript can be recommended for publication after some revision.

**GENERAL COMMENTS**

**The authors obtained important conclusions**: Examining soil quality indicators of terrace soils reveals no significant signs of severe degradation, even with long-term use. The final abandonment of the cultivation system is not attributed to soil exhaustion or terrace structural instability.

The reviewer believes that in conclusion, the Authors could make a breakthrough if they separated relict, agrogenically determined responses, and recent postagrogenic properties.

In contrast to the numerous processes that are united by the phrase "agricultural soil degradation", and for which there is a Mont Blanc of scientific facts, progressive Agropedogenesis, called progradation, is sometimes noted.

A comparison of Agrosoils that differ in the duration of agricultural load with the supposed alternation of land use practices (including those that, to one degree or another, contained agrotechnical components of a soil-saving orientation, and not always consciously applied), makes it possible to establish inherited signs of progradation. Moreover, their contradictory nature may have features of pseudo-progress [Lisetskii F.N. Agrogenic transformation of soils in the dry steppe zone under the impact of antique and recent land management practices // Eurasian Soil Science. 2008. Vol. 41. No. 8. P. 805–817.]

 *Specific Comments:*

**Keywords**. phytolith analysis. The reviewer draws the attention of the Authors to the fact that in the corporate community of scientists of this profile, the term phytolith is considered obsolete (due to the fact that it is narrowed), and the normative term is biomorphs.

**Abstract**. The reviewer does not believe that the logic of this important component of the Article is ideal. For example, should I write in detail about three WRB Reference Soil Groups?

Abstract must be formatted according to international standards and include the following points.

Introductory speech about the research topic.

Purpose of scientific research.

Description of the scientific and practical significance of the work.

Description of the research methodology.

Main results, conclusions of the research work.

The value of the research conducted (what contribution this work made to the relevant field of knowledge).

The final part of the Introduction section also does not formulate the purpose of the study, but rather a list of what will be done. **[But in the Conclusion section the reader finally learns about the purpose of the study].**

**1 Introduction.**

**a)** Given the fact that terrace agriculture was widely practiced in ancient times in the foothills and mountains of various regions of the world, it would be good to provide a global context in the introductory paragraph without limiting it to the Andes. As a hint, you can indicate specialized studies of ancient terraces lands (The Negev Highlands, Israel) or (Eastern Caucasus), etc.

Stavi, I., Eldad, S., Xu, C., Xu, Z., Gusarov, Y., Haiman, M., & Argaman, E. (2024). Ancient agricultural terrace walls control floods and regulate the distribution of Asphodelus ramosus geophytes in the Israeli arid Negev. *Catena*, *234*, 107588.

Sapir, T., Mor-Mussery, A., Abu-Glion, H., Sariy, G., & Zaady, E. (2023). Reclamation of ancient agricultural terraces in the Negev Highlands; soil, archeological, hydrological, and topographical perspectives. *Land Degradation & Development*, *34*(5), 1337-1351.

Borisov, A. V., Kashirskaya, N. N., El'tsov, M. V., Pinskoy, V. N., Plekhanova, L. N., & Idrisov, I. A. (2021). Soils of ancient agricultural terraces of the Eastern Caucasus. Eurasian Soil Science, 54(5), 665-679.

**b)**The authors of the text Article actively (14 times) use the works of Sandor, J. A. and Eash, N. S. This is scientifically correct, because the pioneering contribution of one of these scientists to the development of this topic is significant. **Jonathan A. Sandor (Department of Agronomy, Iowa State University, Ames, Iowa, USA)**

However, the Reviewer would like to point out that both works in References are from 1995, and the key author has more recent Articles.

Eash, N. S. and Sandor, J. A.: Soil chronosequence and geomorphology in a semi-arid valley in the Andes of southern Peru, Geoderma, 65, 59–79, https://doi.org/10.1016/0016-7061(94)00025-6, 1995.

Sandor, J. A. and Eash, N. S.: Ancient Agricultural Soils in the Andes of Southern Peru, Soil Science Society of America Journal, 59, 170–179, https://doi.org/10.2136/sssaj1995.03615995005900010026x, 1995.

In this regard, the Reviewer believes that in addition to priority works, one cannot ignore the most generalizing previous research of the Author (J. A. Sandor) and the large Chapter 2006, as well as the most recent Article on ancient terraces lands in the Chile region.

Sandor, J. A. (2006). Ancient agricultural terraces and soils. In (Ed.) Warkentin, B. P., Footprints in the soil: People and ideas in soil history (pp. 505–534). Elsevier

Sandor, J. A., Huckleberry, G., Hayashida, F. M., Parcero-Oubiña, C., Salazar, D., Troncoso, A., & Ferro-Vázquez, C. (2022). Soils in ancient irrigated agricultural terraces in the Atacama Desert, Chile. Geoarchaeology, 37(1), 96-119.

L 105-106: «The altitude of the mountainous region ranges from 2000 m to 4200 m asl.». This amplitude is of a "background" informational nature, while the reader, if he looks at the three lower insets of Figure 1, then they all reflect valley-river landscapes, which obviously have a significantly lower altitude, which, if indicated, are more useful for understanding geomorphology of study sites.

It is strange that the Authors, speaking about landscapes where there were terraces, limited themselves to section 2.2 Geology. The reviewer believes that this block needs a Relief (Topography) section, where a full-fledged geomorphological analysis is very important: with a range of heights, where there are terraces, slopes, exposure, shape of slopes, etc.

The historical-agrarian section is missing; what agricultural technologies were used in the past? (depth of processing, crops, etc.). For example, Table 2: 50-70 cm = 1 grain Zea Mays. Why at such a depth? Is this the result of formation turnover? [L 856: «terric horizons with a total thickness of 50 cm»].

With the indicated phase of aridization and the emergence of river valleys, did the contribution of irrigation manifest itself in the transformation of the agricultural system?

Figure 5. Between 0 and 85 cm I would like to see the depth values at the boundaries of the horizons.

L 195. Anthrosols. This most important component of the study is described very sparingly. The reviewer believes that it is important to show that there is a dual nature of Soils of ancient agricultural terraces, on the one hand, as cultural soils that are formed as a result of Agropedogenesis [Kuzyakov, Y., & Zamanian, K. (2019), and on the other hand, these are postagrogenic soils with inherited characteristics from their prehistory.

Kuzyakov, Y., & Zamanian, K. (2019). Reviews and syntheses: Agropedogenesis—Humankind as the sixth soil-forming factor and attractors of agricultural soil degradation. Biogeosciences, 16(24), 4783–4803. https://doi.org/10.5194/bg-16-4783-2019

L 265. Above The authors have used WRB many times and a reference to it earlier would have been more appropriate. And so this is a repetition of what has already been used.

L 284: Mg2+), Kalium (K+), Calcium (Ca2+). All valences must be given in uppercase $Mg^{2+}$…

5.2.2 Soil acidity, nutrient availability and soil quality. The application in the section on soil quality characteristics was not implemented (an integral assessment was not obtained based on the available indicators of potential fertility).

**References.**

When comparing 1260 and 1265 onwards: why is the Title Article given either in capital or in lowercase letters?

**Supplementary.** SuppFig1 (b). The reviewer stubbornly does not see the boundary of the transition to the AC horizon.

| Pe10-30/3 | Ah | 35-65 | 10 YR 2/3 |
|---|---|---|---|
| Pe10-30/4 | AC | 65-100 | 10 YR 2/3 |

Look, you have an OM of 2.2% and below 2.2%, Munsell color = the same, so the photo objectively shows that there was a clear error in determining the boundary. Perhaps this is a buried humus layer. (Very bad photo? Crooked ruler, half of the profile in the shadow).

**Supplementary Table S1: Pedochemical analysis.**

Why are commas used and not periods as separators for numbers?

[cmolc Ca2+/kg]. Hereinafter, valences must be in upper case (Excel will allow you to do this). $Ca^{2+}$

The authors show Munsell color (moist). This is "field" humidity, which will change color at different sites and at different Depths. In this regard, comparability can be maintained by giving Munsell color (dry). The reviewer recommends that the authors, if such data are available, provide a replacement.

The authors create confusion with the designation of carbon: C [%]; Corg, Ctot/N (Compare to L 290: «The carbon/nitrogen (C/N) ratios)».

What is the difference between C and Corg, and where does Ctot suddenly appear? (the C and Corg data differ slightly, is this due to different determination methods?). The text indicates DIN 19684-2, 1977 and CNS analyzer vario MAX (and if the values turned out to be close, what does this add scientifically?)

Typically, the ratio Ctot/N is the same as Corg/N, and more elegant (by default, they write C:N, rounded to whole numbers (under a well-known rating scale).

If in the 1st line OM = 2.1*0.579=1.216 Corg, but not 1.0. It is necessary to clarify how the transition from OM to Corg was made (and in principle, Corg alone (without OM) would be enough).

---

## Author Comment (AC3)

**Remarks Anonymous Referee #3**

**R3.1:** This paper provides extensive research involving multiple methods on the anthrosols formed in the terraced agricultural systems in the Andes. The paper provides extensive data covering soil survey results, phytolith analysis, radiocarbon dating, and more. I believe that this paper would provide a firm base for future studies to understand the past human activities and soil in the region. The paper is also well-written and interesting to read, and I believe that it ultimately deserves to be published.

**A3.1**: Thank you for your kind words and for recognizing the importance of our work and suggesting it for publication. We hope our study will contribute to a better understanding of past human activities and soil conditions in the region.

**R3.2:** One thing that I would like to specifically comment on is the smart way that the authors dealt with the chronology of these soils. Dating soils can be very technically challenging, and this is especially the case when applying radiocarbon dating. The authors chose not to use the radiocarbon dates as the main source of establishing the chronology but as a supplement to existing archaeological interpretations and climatic data, which I think would be the most reasonable way with the current data. There have been some recent advances in dating the soil formations using ramped pyrolysis or luminescence dating, so the authors may consider applying these methods and see if there is any new information that can be obtained in future research.

**A3.2**: We value your positive feedback. Your words motivate us to continue our research and to maintain a close interdisciplinary collaboration with our colleagues in archaeology, palaeobotany and archaeometry. To better understand the chronological development of terrace soils in the southern Peruvian Andes, we plan to incorporate additional dating methods, such as OSL, in future work.

**R3.3:** There is also one thing that I would like to know what the authors think. The authors conclude that the anthrosols do not show indicators of severe soil degradation despite the elongated agricultural practices. The authors mention the consistent application of organic manure in the terraces (Section 5.2.3.), which I think the authors point out as the main strategy to prevent soil degradation. If so, what was the source of organic manure and how was it acquired? I am also curious about what the authors think about the impact on soils by the acquisition of manure on a landscape scale. It is a frequent case in the Eurasian context that the acquisition of organic manure, usually originating from the dung of livestock, may

cause soil degradation within a wider landscape, so I am curious about this case in the Andes.

**A3.3**: Thank you for these intriguing questions, which are of great interest to us as well. We will delve into them further in the revised version of the manuscript. We conclude that maintaining a consistent level of organic matter through intentional organic manuring played a crucial role in preserving soil health. Based on the results, where tracing the addition of organic nitrogen sources (e.g., urea or compost) is challenging and phytolith assemblages indicate a broad range of plant elements (e.g., maize leaves or glumes, particularly cobs) from typical crops, we can attribute the fertilizer source to a plant component at this stage of investigation. The broader range of maize plant components, which could only be distinguished in the Santa María system, suggests a deliberate application of mulch material to the soil surface. A combined strategy benefits from in situ biomass, available at higher rates than in natural vegetation, as well as allochthonous organic matter applied as needed by intensive cultivation requirements.

To extend this conclusion to a broader landscape scale, the potential role of organic fertilizers of animal origin (e.g., camelid dung) needs systematic exploration in future research; their use is plausible. As noted, Handley et al. (2023) suggest that the Chicha Sora terrace system likely received regular maintenance through the addition of camelid dung, which is known to be less prone to accelerated soil erosion than later-introduced cattle, thanks to its adaptation to topography and native vegetation. The Laramate area provides an opportunity to approach this phenomenon from various angles. A substantial number of circular structures (Soßna, 2015), now utilized seasonally as corrals, remain well-preserved; their quantification using GIS techniques forms a starting point for model development. Studies on domestic animal densities in an archaeological context, utilizing oribatid mite concentrations as indicators, such as those conducted in the Patacancha Valley in Cuzco by Chepstow-Lusty et al. (2007), could offer valuable insights.

**R3.4:** Line 270: "Twenty-eight samples were collected …" to "Twenty-eight samples for physico-chemical soil analyses". I think it would be better to address what samples are they for firsthand. The following sentence can be modified accordingly.

**A3.4**: Thank you for your pertinent observation. We agree with your suggestion. An alternative sentence could read as follows: "A total of twenty-eight samples for soil physico-chemical analyses were collected from seven soil profiles (Supplementary Table S1). The samples were analysed at the Laboratory of Geomorphology and Geoecology, Institute of Geography, University of Heidelberg."

**R3.5:** Table 1 & 2: I am doubtful whether this is the right place to present this data since this is the section for Materials and Methods. I believe this fits more with the Results section, in which the authors are presenting the contents of the tables in the forms of figures and text. The authors may consider presenting the tables as supplementary material.

**A3.5**: Thank you for your valuable suggestion. We will consider making adjustments for the benefit of the reader. Our intention was to include this information in both the Methods and Results sections without overloading the Results chapter. However, in the revised manuscript, we will place Table 1 in the Results chapter and Table 2 in the Supplementary Materials section.

**R3.6:** Section 4.1: Also, this section seems a bit out of place to me, since I think that the "Results" section should reflect the results of the methods that the authors employed. I think this section may be a perfect follow-up for Section 2.6, scaling down from an introduction to the region to the actual sites that have been investigated in this research.

**A3.6**: Thank you for your valid comment. This section is primarily based on the cited literature, with only a few minor observations. We therefore agree to move this section to chapter 2.7 of the study site, so that the reader can concentrate on the findings of the study in the results chapter.

**References:**

Chepstow-Lusty, A. J., Frogley, M. R., Bauer, B. S., Leng, M. J., Cundy, A. B., Boessenkool, K. P., and Gioda, A.: Evaluating socio-economic change in the Andes using oribatid mite abundances as indicators of domestic animal densities, Journal of Archaeological Science, 34, 1178–1186, https://doi.org/10.1016/j.jas.2006.12.023, 2007.

Handley, J., Branch, N., Meddens, F. M., Simmonds, M., and Iriarte, J.: Pre-Hispanic terrace agricultural practices and long-distance transfer of plant taxa in the southern-central Peruvian Andes revealed by phytolith and pollen analysis, Vegetation History and Archaeobotany, https://doi.org/10.1007/s00334-023-00946-w, 2023.

Soßna, V.: Climate and settlement in southern Peru: the northern Río Grande de Nasca drainage between 1500 BCE and 1532 CE, Dr Ludwig Reichert, 2015.

---

## Author Response (AR1)

**Copernicus Publications**

**Editorial board**

Montevideo, 28.07.2024

**Preprint egusphere-2024-637**

Dear editor,

Please find attached our revised manuscript:

Fernando Leceta, Christoph Binder, Christian Mader, Bertil Mächtle, Erik Marsh, Laura Dietrich, Markus Reindel, Bernhard Eitel, and Julia Meister:

**The impact of agriculture on tropical mountain soils in the western Peruvian Andes: a pedo-geoarchaeological study of terrace agricultural systems in the Laramate region (14.5°S).**

We hope that the revised version will be accepted for publication. Below we attached detailed response to the referees' comments.

Very best wishes and many thanks

Fernando Leceta

**Universidad de la República Uruguay (UdelaR)**

**Espacio Interdisciplinario**

11200 Montevideo, Uruguay

E-Mail: leceta@gmail.com

Revision notes for the manuscript:

**The impact of agriculture on tropical mountain soils in the western Peruvian Andes: a pedo-geoarchaeological study of terrace agricultural systems in the Laramate region (14.5° S).**

Submitted 12th March 2024 to SOIL Journal, Copernicus Publications; MS No: egusphere-2024-637

**Authors statement:**

We sincerely thank the editor and the three reviewers for the thorough review of our manuscript and their constructive comments, which helped to improve the paper. We really appreciate your support and your attempt to improve the paper. The reviewer's comments are given below, complemented by our statements in blue.

We hope that this will clarify all remaining questions. Thank you again for all your time invested and efforts made.

Yours sincerely,

Fernando Leceta and co-authors

**Remarks Anonymous Referee #1**

**General comments**

**R1.1.: The authors obtained important conclusions**: Examining soil quality indicators of terrace soils reveals no significant signs of severe degradation, even with long-term use. The final abandonment of the cultivation system is not attributed to soil exhaustion or terrace structural instability.

The reviewer believes that in conclusion, the Authors could make a breakthrough if they separated relict, agrogenically determined responses, and recent postagrogenic properties.

In contrast to the numerous processes that are united by the phrase "agricultural soil degradation", and for which there is a Mont Blanc of scientific facts, progressive Agropedogenesis, called progradation, is sometimes noted.

A comparison of Agrosoils that differ in the duration of agricultural load with the supposed alternation of land use practices (including those that, to one degree or another, contained agrotechnical components of a soil-saving orientation, and not always consciously applied), makes it possible to establish inherited signs of progradation. Moreover, their contradictory nature may have features of pseudo-progress [Lisetskii F.N. Agrogenic transformation of soils in the dry steppe zone under the impact of antique and recent land management practices // Eurasian Soil Science. 2008. Vol. 41. No. 8. P. 805–817.]

A1.1: Thank you for this important comment. Throughout the manuscript, we have emphasized that the terrace soils of the Laramate region are the result of sequential genetic processes and are not comparable to soils subjected to natural pedogenesis. Their evolution reflects the following main phases: 1) natural pedogenesis on undisturbed soils, 2) soils modified by intensive agricultural practices, and 3) post-agrogenetic soils.

Two major sections of the revised manuscript version reflect these changes: sections 2.5 [L211-226] and 5.3.2 [L1010-1020] in the Study Site and Discussion chapters, respectively.

**Specific Comments**

**R1.2: Keywords**. phytolith analysis. The reviewer draws the attention of the Authors to the fact that in the corporate community of scientists of this profile, the term phytolith is considered obsolete (due to the fact that it is narrowed), and the normative term is biomorphs.

**A1.2**: Thank you for your suggestion regarding the terms "phytoliths" and "biomorphs" for laboratory analysis. We have included "Plant microfossils" in the "Keywords" section instead, as it includes both phytoliths and starch grains. However, we propose retaining the terms "phytoliths" and "phytolith analysis" in the abstract and subsequent sections.

**R1.3: Abstract.** The reviewer does not believe that the logic of this important component of the Article is ideal. For example, should I write in detail about three WRB Reference Soil Groups?

Abstract must be formatted according to international standards and include the following points.

Introductory speech about the research topic.

Purpose of scientific research.

Description of the scientific and practical significance of the work.

Description of the research methodology.

Main results, conclusions of the research work.

The value of the research conducted (what contribution this work made to the relevant field of knowledge).

The final part of the Introduction section also does not formulate the purpose of the study, but rather a list of what will be done. **[But in the Conclusion section the reader finally learns about the purpose of the study].**

**A1.3**: Thank you for your comment. You are right. The abstract [L21-54] has been structured according to international standards and suggested guidelines. Adjustments include a concise restatement of the study objectives to achieve a comprehensive understanding of the pedological history of the Laramate area through the assessment of abandoned terrace soils and their undisturbed soil context, along with the main results and conclusions.

**R1.4: 1 Introduction.**

Given the fact that terrace agriculture was widely practiced in ancient times in the foothills and mountains of various regions of the world, it would be good to provide a global context in the introductory paragraph without limiting it to the Andes. As a hint, you can indicate

specialized studies of ancient terraces lands (The Negev Highlands, Israel) or (Eastern Caucasus), etc.

Stavi, I., Eldad, S., Xu, C., Xu, Z., Gusarov, Y., Haiman, M., & Argaman, E. (2024). Ancient agricultural terrace walls control floods and regulate the distribution of Asphodelus ramosus geophytes in the Israeli arid Negev. Catena, 234, 107588.

Sapir, T., Mor-Mussery, A., Abu-Glion, H., Sariy, G., & Zaady, E. (2023). Reclamation of ancient agricultural terraces in the Negev Highlands; soil, archeological, hydrological, and topographical perspectives. Land Degradation & Development, 34(5), 1337-1351.

Borisov, A. V., Kashirskaya, N. N., El'tsov, M. V., Pinskoy, V. N., Plekhanova, L. N., & Idrisov, I. A. (2021). Soils of ancient agricultural terraces of the Eastern Caucasus. Eurasian Soil Science, 54(5), 665-679.

**A1.4**: Thank you for your comment. Including a brief description of terrace cultivation in other regions of the world provides essential context for understanding this phenomenon on a global scale. Particular attention has been given to regions with similar climatic conditions and ancient farming systems, such as central and southern Mexico, the Caucasus, the Mediterranean, and Israel.

The changes are included in the first section of the introduction [L60-65] of the revised manuscript.

**R1.5: b**) The authors of the text Article actively (14 times) use the works of Sandor, J. A. and Eash, N. S. This is scientifically correct, because the pioneering contribution of one of these scientists to the development of this topic is significant. **Jonathan A. Sandor (Department of Agronomy, Iowa State University, Ames, Iowa, USA)**

However, the Reviewer would like to point out that both works in References are from 1995, and the key author has more recent Articles.

Eash, N. S. and Sandor, J. A.: Soil chronosequence and geomorphology in a semi-arid valley in the Andes of southern Peru, Geoderma, 65, 59–79, https://doi.org/10.1016/0016-7061(94)00025-6, 1995.

Sandor, J. A. and Eash, N. S.: Ancient Agricultural Soils in the Andes of Southern Peru, Soil Science Society of America Journal, 59, 170–179, https://doi.org/10.2136/sssaj1995.03615995005900010026x, 1995.

In this regard, the Reviewer believes that in addition to priority works, one cannot ignore the most generalizing previous research of the Author (J. A. Sandor) and the large Chapter 2006, as well as the most recent Article on ancient terraces lands in the Chile region.

Sandor, J. A. (2006). Ancient agricultural terraces and soils. In (Ed.) Warkentin, B. P., Footprints in the soil: People and ideas in soil history (pp. 505–534). Elsevier

Sandor, J. A., Huckleberry, G., Hayashida, F. M., Parcero-Oubiña, C., Salazar, D., Troncoso, A., & Ferro-Vázquez, C. (2022). Soils in ancient irrigated agricultural terraces in the Atacama Desert, Chile. Geoarchaeology, 37(1), 96-119.

**A1.5**: Thank you for your comment. Our study indeed benefits from the inclusion of newly published literature. Accordingly, the introduction of the manuscript has been complemented by incorporating recent studies on terrace soils in the Peruvian Andes and the neighbouring regions of the entire Andean highlands [L79-84].

**R1.6:** L 105-106: «The altitude of the mountainous region ranges from 2000 m to 4200 m asl.». This amplitude is of a "background" informational nature, while the reader, if he looks at the three lower insets of Figure 1, then they all reflect valley-river landscapes, which obviously have a significantly lower altitude, which, if indicated, are more useful for understanding geomorphology of study sites.

It is strange that the Authors, speaking about landscapes where there were terraces, limited themselves to section 2.2 Geology. The reviewer believes that this block needs a Relief (Topography) section, where a full-fledged geomorphological analysis is very important: with a range of heights, where there are terraces, slopes, exposure, shape of slopes, etc.

**A1.6**: Thank you for your valuable suggestion. The authors agree that the inclusion of a comprehensive geomorphological analysis of the study area will provide a more detailed understanding of the morphodynamics and their effects on the structural stability of the terrace system. The following elements have been included:

1. Description of the main structural features and units of the middle and upper Rio Grande.

2. Relief and morphological analysis of the fluvial subsystems of the Ingenio, Viscas and Palpa rivers using available topographic information.

3. Analysis of the morphological situation of each terrace system within the landscape.

We present the changes in the revised manuscript version as follows:

1. Additional information on terrain uplift and morphological units of the Cordillera Occidental in the geology section [L144-147].
2. Expanded methodological descriptions for GIS-based geomorphological analysis and mapping of agricultural terraces in Chapter 3.1 [L328-333].

3. A newly included section on the geomorphology and landforms of the Laramate terrace complex in Chapter 4.1 [L424-451] of the revised manuscript version.

**R1.7:** The historical-agrarian section is missing; what agricultural technologies were used in the past? (depth of processing, crops, etc.).

**A1.7**: Ethnohistorical studies on agrarian technologies (e.g., crops, tillage tools, irrigation techniques, camelid dung application, crop rotation, surface burning) are unfortunately scarce and almost non-existent in the Laramate area.

Goodman-Elgar (2008) and Nanavati et al. (2016) highlight the advantages of non-mechanized tillage tools in their studies on pre-Hispanic terraces at Viejo Sangayaico in the nearby upper Ica and Paca valleys, respectively. Tools like the *chaquitaclla*, which have been used in the Peruvian Andes since ancient times and remain part of the current rural landscape, typically till the soil to a depth of 30 cm with their curved metal or wooden tips. In conjunction with another traditional hand-tillage tool, the *ranccana*, these tools facilitate soil cultivation with less destructiveness compared to industrial ploughing. It is reasonable to infer their use on the Laramate terraces. Future work should test this hypothesis and explore their actual effects on the Laramate terrace soils.

The potential role of organic fertilizers of animal origin (e.g., camelid dung) needs to be systematically addressed in future studies. Handley et al. (2023) found that the Chicha Soras terrace system was likely maintained by the regular addition of camelid dung. Although the addition of organic nitrogen sources (e.g., urea or compost) was difficult to detect in the Laramate terraces, their use cannot be completely excluded. Studies of domestic animal densities in an archaeological context, using oribatid mite concentrations as an indirect indicator of manuring, such as those carried out in the Patacancha Valley in Cuzco by Chepstow-Lusty et al. (2007), could be considered in future work to better understand this hypothesis.

Reliable information on crops in the Laramate area comes from a recent publication by Mader et al. (2024). The study indicates that maize was most likely an important crop on the agricultural terraces around the Cutamalla archaeological site.

With the information currently available, the picture of the agricultural history of the Laramate region is only partial, but there is significant potential for future studies.

The revised version of the manuscript includes the points discussed above in a new section 5.3.6: "Prehispanic Agricultural Practices in the Laramate Region" [L1170-1199].

**R1.8:** For example, Table 2: 50-70 cm = 1 grain Zea Mays. Why at such a depth? Is this the result of formation turnover? [L 856: «terric horizons with a total thickness of 50 cm»].

**A1.8**: Thank you for your observation. Regarding the presence of a maize grain on Pe10-31/4 of Sihuilca (Bwt 50-70 cm), we interpret its origin to be from the former soil surface (Ah 20-50 cm) during an early phase of cultivation, prior to the terrace construction. This is more likely than a later incorporation from the uppermost Ap horizons.

The mixing of the soil substrate in the 30 cm thick horizon at Ah, and its subsequent translocation into the underlying Bwt horizon, is plausible considering the combined action of the available tillage tools of the time and the influence of local burrowing fauna. It is interpreted that the Sihuilca site experienced early agricultural use, specifically the cultivation of maize, during the initial stages of the Late Formative/Early Horizon or possibly even earlier.

We discuss the points discussed above in section 5.3.5 (L1156-1162) of the revised version of the manuscript.

**R1.9:** With the indicated phase of aridization and the emergence of river valleys, did the contribution of irrigation manifest itself in the transformation of the agricultural system?

**A1.9**: Thank you for your question. Despite technological advances in terracing and irrigation techniques during the Middle Horizon, these innovations did not lead to a massive transformation of agricultural systems in the Laramate region. Although there were a small number of newly established irrigated terrace systems in the area, they represented a relatively small proportion of the cultivated area. In addition, most of these terraces were located in areas with poorer soil quality that had not previously been used for agriculture. Thus, although the agricultural utilisation of previously unused land through technological improvements represents a systemic development, it did not lead to a profound transformation of the agricultural system in the study area.

In addition to the comments in section 5.4 of our original manuscript, we have added some additional lines [L1257-1259] to reaffirm our original position.

**R1.10:** Figure 5. Between 0 and 85 cm I would like to see the depth values at the boundaries of the horizons.

**A1.10**: We appreciate your feedback. Due to space and design constraints, the boundaries of each horizon in this figure are indicated indirectly by the lateral graphic scale. The boundaries are shown in centimetres as follows: Ap1: 0-10, Ap2: 10-20, Ah: 20-50, Bwt: 50-

70, C1: 70-85. This information is also provided in Chapter 4.2.1 of the revised manuscript, as well as in the supplementary section.

**R1.11:** L 195. Anthrosols. This most important component of the study is described very sparingly. The reviewer believes that it is important to show that there is a dual nature of Soils of ancient agricultural terraces, on the one hand, as cultural soils that are formed as a result of Agropedogenesis [Kuzyakov, Y., & Zamanian, K. (2019), and on the other hand, these are postagrogenic soils with inherited characteristics from their prehistory.

Kuzyakov, Y., & Zamanian, K. (2019). Reviews and syntheses: Agropedogenesis—Humankind as the sixth soil-forming factor and attractors of agricultural soil degradation. Biogeosciences, 16(24), 4783–4803. https://doi.org/10.5194/bg-16-4783-2019.

**A1.11**: Thank you for your feedback. The authors agree that the concept of *Anthrosols* needs to be further developed in the text. The following aspects have been addressed:

1. The formation of anthropogenic soils in relation to the processes of agropedogenesis and their behaviour after the final abandonment of agricultural use.

2. Diagnostic characteristics of anthrosols in relation to local climatology and conditions of formation.

3. A brief overview of anthrosols on agricultural terraces, which are a typical feature of the Peruvian Andes landscape.

Although the recovery process of the Laramate soils has clearly begun, it remains unclear to what extent the current soil properties reflect the post-agrogenetic process and the degradation level previously reached. A valuable approach to understanding the dimensions of agrogenetic and post-agrogenetic processes described by Kuzyakov and Zamanian, (2019) is offered by a group of terraces not included in this study but investigated in the field: the Patachana system (14°15'24 "S, 74°49'37 "W, 3425 m a.s.l.). This system shares typological similarities with Sihuilca, Ayllapampa, and Santa María but features unique prehispanic irrigation characteristics. Unlike other systems in the Laramate region, largely abandoned after 1532 CE, the Patachana system has been intensively used since the colonial period. Therefore, the Patachana system is a valuable site for evaluating ancient agrogenetic soils without post-agrogenetic traits and should be considered in future research to validate Kuzyakov and Zamanian's (2019) model of agrogenetic soils.

We mention the above points in Section 2.5 [L211-226] on the description of *Anthrosols* and discuss them further in Section 5.3.2 [L1010-1020] of the revised manuscript version.

**R1.12:** L 265. Above The authors have used WRB many times and a reference to it earlier would have been more appropriate. And so this is a repetition of what has already been used.

**A1.12**: Thank you for your note. However, we believe that this specific reference to the description of the field data and soil classification system should remain in the methods section. Previous and subsequent mentions of the WRB have been removed.

**R1.13:** L 284: Mg2+), Kalium (K+), Calcium (Ca2+). All valences must be given in uppercase Mg2+…

**A1.13**: Your observation is correct, thank you. Valences should be written in uppercase letters according to the standards. This has been corrected throughout the revised manuscript.

**R1.14:** 5.2.2 Soil acidity, nutrient availability and soil quality. The application in the section on soil quality characteristics was not implemented (an integral assessment was not obtained based on the available indicators of potential fertility).

**A1.14**: Thank you for your feedback. The authors agree that the section needs to provide a comprehensive assessment of the agro-ecological potential for specific crops within the terrace systems, and to emphasize the relevance of a central theme of this research: the management of soil resources through agricultural practices. Indicators such as soil acidity and CEC/BS are well-known for assessing soil quality and are effective for evaluating soil responses to intensive agricultural practices. These indicators reflect efforts to mitigate the negative effects of intensive cultivation (e.g., nutrient depletion or soil acidification) and to optimize crop production, particularly through the incorporation of organic matter into the topsoil and soil mixing during terrace construction.

The changes include various revised passages throughout section 5.3.2 [L965-1020] of the revised manuscript.

**R1.15: References.**
When comparing 1260 and 1265 onwards: why is the Title Article given either in capital or in lowercase letters?

**A1.15**: Thank you for your comment. The inconsistencies in our references section are due to the direct import of bibliography files in different formats. Your observation is correct; references have been formatted consistently in the revised version of the manuscript.

**R1.16: Supplementary.** SuppFig1 (b). The reviewer stubbornly does not see the boundary of the transition to the AC horizon.

Look, you have an OM of 2.2% and below 2.2%, Munsell color = the same, so the photo objectively shows that there was a clear error in determining the boundary. Perhaps this is a buried humus layer. (Very bad photo? Crooked ruler, half of the profile in the shadow).

**A1.16**: Thank you for this important note. We agree with your remark. The high content of mixed anthropogenic material (animal bones, pottery fragments, and terrace building material fragments) at Pe10-30/4 led to an incorrect determination and interpretation of the horizon at this level. The terrace serves both agricultural and domestic functions, which is evident in the high level of reworked material.

On the basis of field observations, macromorphological data and available analyses we have reassigned the profile sequence as follows: Ap1, Ap2, Ah1, Ah2, C. It was explicitly stated in the manuscript that all horizons in this profile are particularly diffuse in the corresponding section 4.2.3 [L661-673]. This distinction is supported by the attached data analysis, where the variations are very subtle. The Ap1-Ap2 and Ah1-Ah2 blocks can be grouped according to their clay content and soil skeleton content as distinguishing features.

**R1.17: Supplementary Table S1: Pedochemical analysis.**
Why are commas used and not periods as separators for numbers?

**A1.17**: You are correct. Indeed, periods should be used consistently as the default separator for numerical decimal information throughout the manuscript, including figures, tables and appendices. The changes were made to all documents.

**R1.18:** [cmolc Ca2+/kg]. Hereinafter, valences must be in upper case (Excel will allow you to do this). $Ca^{2+}$

**A1.18**: You are right. Throughout the revised manuscript, the valences have been written in superscript.

**R1.19:** The authors show Munsell color (moist). This is "field" humidity, which will change color at different sites and at different Depths. In this regard, comparability can be maintained by giving Munsell color (dry). The reviewer recommends that the authors, if such data are available, provide a replacement.

**A1.19**: We appreciate your feedback and the opportunity to clarify the procedures applied. The colour determination of soil sediments was conducted under laboratory conditions following specific protocols. To ensure consistency, several measures were taken, including using the same amount of homogenized soil substrate, maintaining consistent diffuse

lighting conditions, utilizing the same edition of the Munsell Soil Colour Charts book (Munsell Color, 2000), and applying the same amount of water through a water sprayer to achieve a low level of moisture in the sediment.

Due to logistical and staffing constraints, conducting a new determination of this parameter in the laboratory is currently not feasible. However, we believe that the procedures outlined above adhere to the standards for laboratory measurements in soil science. We have enhanced the methodology section 3.3.1 in the revised manuscript with a more detailed description [L363-366].
* * *
**R1.20:** The authors create confusion with the designation of carbon: C [%]; Corg, Ctot/N (Compare to L 290: «The carbon/nitrogen (C/N) ratios)»

What is the difference between C and Corg, and where does Ctot suddenly appear? (the C and Corg data differ slightly, is this due to different determination methods?). The text indicates DIN 19684-2, 1977 and CNS analyzer vario MAX (and if the values turned out to be close, what does this add scientifically?).
* * *
**A1.20**: We appreciate your constructive comments in this section and apologize for any confusion caused by misunderstanding of terms.

Both C [%] and Ctot [%] refer to the total carbon fraction within the soil sample, measured by elemental analysis through the CNS analyzer, encompassing both organic and inorganic fractions. To avoid confusion, references in the revised manuscript and supplementary materials only use C and not Ctot. Corg specifically denotes the organic carbon fraction of the soil, determined by photometric measurement at 590 nm using a spectrophotometer via the wet combustion dichromate method. Both C and Corg are expressed as percentages.

These complementary methods (elemental analysis and wet combustion) enable a thorough interpretation of the pedogenetic process intensity. The Corg content primarily reflects pedogenetic processes, while the inorganic carbon content (Cinorg) offers insights into the nature and composition of the source rock. The inorganic carbon content is derived by subtracting the organic carbon content from the total carbon content (C - Corg = Cinorg).

Samples where the C and Corg values are almost identical, particularly in soils developed over carbonate-free parent material, underscore the intensity of the pedogenic process.
* * *
**R1.21:** Typically, the ratio Ctot/N is the same as Corg/N, and more elegant (by default, they write C:N, rounded to whole numbers (under a well-known rating scale).

A1.21: The Ctot/N ratio is written as C:N in the revised manuscript. However, as the ratios within the profiles sometimes vary only slightly, we have chosen to present the ratios to one decimal place.

**R1.22:** If in the 1st line OM = 2.1*0.579=1.216 Corg, but not 1.0. It is necessary to clarify how the transition from OM to Corg was made (and in principle, Corg alone (without OM) would be enough).

**A1.22**: Thank you for your comment. To obtain the OM value indirectly, we follow the procedures for soil analysis as cited by ISRIC (van Reeuwijk, 2002, p. 23) and Barsch et al. (2000, p. 344). These procedures recommend converting organic carbon (%) to organic matter (%) by multiplying with the empirical factor 2, especially when specific information on the organic matter is unavailable (Nelson and Sommers, 1982).

In the revised manuscript [L373-375], we have added a more detailed description of the methodology in section 3.3.1.

**References:**

Barsch, H., Billwitz, K., and Bork, H.: Arbeitsmethoden in Physiogeographie und Geoökologie, Klett-Perthes Gotha, 612 pp., 2000.

Chepstow-Lusty, A. J., Frogley, M. R., Bauer, B. S., Leng, M. J., Cundy, A. B., Boessenkool, K. P., and Gioda, A.: Evaluating socio-economic change in the Andes using oribatid mite abundances as indicators of domestic animal densities, Journal of Archaeological Science, 34, 1178–1186, https://doi.org/10.1016/j.jas.2006.12.023, 2007.

Goodman-Elgar, M.: Evaluating soil resilience in long-term cultivation: a study of pre-Columbian terraces from the Paca Valley, Peru, Journal of Archaeological Science, 35, 3072–3086, https://doi.org/10.1016/j.jas.2008.06.003, 2008.

Handley, J., Branch, N., Meddens, F. M., Simmonds, M., and Iriarte, J.: Pre-Hispanic terrace agricultural practices and long-distance transfer of plant taxa in the southern-central Peruvian Andes revealed by phytolith and pollen analysis, Vegetation History and Archaeobotany, https://doi.org/10.1007/s00334-023-00946-w, 2023.

Kuzyakov, Y. and Zamanian, K.: Reviews and syntheses: Agropedogenesis – humankind as the sixth soil-forming factor and attractors of agricultural soil degradation, Biogeosciences, 16, 4783–4803, https://doi.org/10.5194/bg-16-4783-2019, 2019.

Mader, C., Godde, P., Behl, M., Binder, C., Hägele, E., Isla, J., Leceta, F., Lyons, M., Marsh, E., Odenthal, R., Fernengel, E., Stryjski, P., Weber, A.-K., Reindel, M., and Meister, J.: An integrative approach to ancient agricultural terraces and forms of dependency: the case of Cutamalla in the prehispanic Andes, Frontiers in Environmental Archaeology, 3, https://doi.org/10.3389/fearc.2024.1328315, 2024.

Munsell Color, L.: Munsell soil color charts, Year 2000 rev. washable ed., Munsell Color, GretagMacbeth, New Windsor, NY, 2000.

Nanavati, W. P., French, C., Lane, K., Oros, O. H., and Beresford-Jones, D.: Testing soil fertility of Prehispanic terraces at Viejo Sangayaico in the upper Ica catchment of south-central highland Peru, Catena, 142, 139–152, 2016.

Nelson, D. W. and Sommers, L. E.: Total Carbon, Organic Carbon, and Organic Matter, in: Methods of Soil Analysis, 539–579, https://doi.org/10.2134/agronmonogr9.2.2ed.c29, 1982.

van Reeuwijk, L. P.: Procedures for soil analysis, SIXTH EDITION., ISRIC, Wageningen, 2002.

**Remarks Anonymous Referee #2**

**R2.1:** A lot of thanks for such interesting and important article. I have only several technical notes.

**A2.1**: Thank you for reviewing our article, for your comments and for the opportunity to clarify any ambiguities.

**R2.2:** There is no explanation for photo 7 in Figure 6.

**A2.2**: You are absolutely right, and we apologize for this mistake. Figure 6 (f) and now 6 (g) show the same starch grain under different lighting conditions: (f) under plane-polarized light and (g) under cross-polarized light. We have included this information in the manuscript.

**R2.3:** You cannot use two nomenclatures of phytoliths. You should only use the latest one - for 2019. All names of phytolith morphotypes must be given according to the nomenclature for 2019. The names of phytolith morphotypes should be written in small capital letters and not in italics.

**A2.3**: We appreciate your feedback and you are right. We have revised the nomenclature of phytolith morphotypes according to Neumann et al. (2019), using capital letters in the text and figures.

**References:**

Neumann, K., Strömberg, C. A. E., Ball, T., Albert, R. M., Vrydaghs, L., and Cummings, L. S.: International Code for Phytolith Nomenclature (ICPN) 2.0, Ann Bot, 124, 189–199, https://doi.org/10.1093/aob/mcz064, 2019.

**Remarks Anonymous Referee #3**

**R3.1:** This paper provides extensive research involving multiple methods on the anthrosols formed in the terraced agricultural systems in the Andes. The paper provides extensive data covering soil survey results, phytolith analysis, radiocarbon dating, and more. I believe that this paper would provide a firm base for future studies to understand the past human activities and soil in the region. The paper is also well-written and interesting to read, and I believe that it ultimately deserves to be published.

**A3.1**: Thank you for your kind words and for recognizing the importance of our work and suggesting it for publication. We hope our study will contribute to a better understanding of past human activities and soil conditions in the region.

**R3.2:** One thing that I would like to specifically comment on is the smart way that the authors dealt with the chronology of these soils. Dating soils can be very technically challenging, and this is especially the case when applying radiocarbon dating. The authors chose not to use the radiocarbon dates as the main source of establishing the chronology but as a supplement to existing archaeological interpretations and climatic data, which I think would be the most reasonable way with the current data. There have been some recent advances in dating the soil formations using ramped pyrolysis or luminescence dating, so the authors may consider applying these methods and see if there is any new information that can be obtained in future research.

**A3.2**: We value your positive feedback. Your words motivate us to continue our research and to maintain a close interdisciplinary collaboration with our colleagues in archaeology, palaeobotany and archaeometry. To better understand the chronological development of terrace soils in the southern Peruvian Andes, we plan to incorporate additional dating methods, such as OSL, in future work.

**R3.3:** There is also one thing that I would like to know what the authors think. The authors conclude that the anthrosols do not show indicators of severe soil degradation despite the elongated agricultural practices. The authors mention the consistent application of organic manure in the terraces (Section 5.2.3.), which I think the authors point out as the main strategy to prevent soil degradation. If so, what was the source of organic manure and how was it acquired? I am also curious about what the authors think about the impact on soils by the acquisition of manure on a landscape scale. It is a frequent case in the Eurasian context that the acquisition of organic manure, usually originating from the dung of livestock, may

cause soil degradation within a wider landscape, so I am curious about this case in the Andes.

**A3.3**: Thank you for these intriguing questions, which are also of great interest to us. We have addressed them in the revised version of the manuscript. We conclude that maintaining a consistent level of organic matter through targeted organic manuring played a crucial role in maintaining soil health. Based on the results, where the addition of organic nitrogen sources (e.g. urea) is difficult to trace, and the phytolith assemblages indicate a wide range of plant groups and parts (e.g. maize leaves or glumes, especially cobs), this suggests intensive cultivation and deliberate application of plant material to the soil surface for fertilisation and/or mulching purposes. Typically, a combined cultivation and fertilisation strategy benefits from in situ biomass, which is available at higher rates than in natural vegetation, and from allochthonous organic matter, which is applied as required by the demands of intensive cultivation.

The potential role of organic fertilizers of animal origin (e.g., camelid dung) in the Laramate region warrants systematic investigation in future studies. Handley et al. (2023) found that in the Chicha Soras terrace system in the neighbouring Apurimac region, the nutrient content of the terraces was likely maintained in prehispanic times by periodic burning of the terrace vegetation combined with the regular addition of camelid dung. In the Laramate terraces, the addition of organic nitrogen sources (e.g., urea) was difficult to detect, as evidenced by the almost negligible total nitrogen contents. However, the total phosphorus contents are elevated in the profiles of the Santa Maria complex, particularly in profile Pe11-06 (Supplementary Table 1), which is not lithologically related to the outcropping quartzites or nearby plutonites. This suggests that the use of livestock manure is plausible and cannot be completely excluded, especially since a substantial number of prehispanic circular structures (Soßna, 2015), now utilized seasonally as corrals, remain well-preserved in the study area. Quantifying these structures using GIS techniques could be a starting point for studies on domestic animal densities in an archaeological context. Moreover, using oribatid mite concentrations as an indirect indicator of manuring, similar to studies conducted in the Patacancha Valley in Cuzco Chepstow-Lusty et al. (2007), could be considered in future work to better understand this hypothesis. Notably, there is no evidence supporting the notion that nutrient levels were sustained through seasonal burning in the Laramate region.

In the revised version of the manuscript, most of these points are described in a new section 5.3.6: "Prehispanic agricultural practices in the Laramate region" [L1170-1199].

**R3.4:** Line 270: "Twenty-eight samples were collected …" to "Twenty-eight samples for physico-chemical soil analyses". I think it would be better to address what samples are they for firsthand. The following sentence can be modified accordingly.

**A3.4**: Thank you for your pertinent comment. We agree with your suggestion. We have added the following alternative sentences: "A total of twenty-eight samples for soil physico-chemical analyses were collected from seven soil profiles (Supplementary Table S1). The samples were analysed at the Laboratory of Geomorphology and Geoecology, Institute of Geography, University of Heidelberg". The revised section can be found in section 3.3.1 [L354-356] of the revised manuscript.

**R3.5:** Table 1 & 2: I am doubtful whether this is the right place to present this data since this is the section for Materials and Methods. I believe this fits more with the Results section, in which the authors are presenting the contents of the tables in the forms of figures and text. The authors may consider presenting the tables as supplementary material.

**A3.5**: Thank you for your valuable suggestion. Our initial intention was to include this information in both the Methods and Results sections without overloading the Results chapter. However, in the revised manuscript, Table 1 is placed in the Results chapter [L471-473] and Table 2 in the Supplementary Materials section (now Supplementary Table S2).

**R3.6:** Section 4.1: Also, this section seems a bit out of place to me, since I think that the "Results" section should reflect the results of the methods that the authors employed. I think this section may be a perfect follow-up for Section 2.6, scaling down from an introduction to the region to the actual sites that have been investigated in this research.

**A3.6**: Thank you for your valid comment. This section is mainly based on the literature cited, with only a few minor observations. We have therefore agreed to move this section to the study site chapter (new section 2.7), so that the reader can concentrate on the findings of the study in the results chapter. [L270-314]

**References:**

Chepstow-Lusty, A. J., Frogley, M. R., Bauer, B. S., Leng, M. J., Cundy, A. B., Boessenkool, K. P., and Gioda, A.: Evaluating socio-economic change in the Andes using oribatid mite abundances as indicators of domestic animal densities, Journal of Archaeological Science, 34, 1178–1186, https://doi.org/10.1016/j.jas.2006.12.023, 2007.

Handley, J., Branch, N., Meddens, F. M., Simmonds, M., and Iriarte, J.: Pre-Hispanic terrace agricultural practices and long-distance transfer of plant taxa in the southern-central Peruvian Andes revealed by phytolith and pollen analysis, Vegetation History and Archaeobotany, https://doi.org/10.1007/s00334-023-00946-w, 2023.

Soßna, V.: Climate and settlement in southern Peru: the northern Río Grande de Nasca drainage between 1500 BCE and 1532 CE, Dr Ludwig Reichert, 2015.